## Registered report

psychology

research practices, open science,
scientific integrity, informed consent

**Author for correspondence:**
Julia G. Bottesini
e-mail: jbottesini@ucdavis.edu

# What do participants think of our research practices? An examination of behavioural psychology participants' preferences

Julia G. Bottesini[1], Mijke Rhemtulla[1] and Simine Vazire[1,2]

[1]Department of Psychology, University of California—Davis, Davis, CA, USA
[2]Department of Psychology, University of Melbourne, Melbourne School of Psychological Sciences, Melbourne, Victoria, Australia

 JGB, 0000-0002-5340-993X; MR, 0000-0003-2572-2424;
SV, 0000-0002-3933-9752

What research practices should be considered acceptable? Historically, scientists have set the standards for what constitutes acceptable research practices. However, there is value in considering non-scientists' perspectives, including research participants'. 1873 participants from MTurk and university subject pools were surveyed after their participation in one of eight minimal-risk studies. We asked participants how they would feel if (mostly) common research practices were applied to their data: *p*-hacking/cherry-picking results, selective reporting of studies, Hypothesizing After Results are Known (HARKing), committing fraud, conducting direct replications, sharing data, sharing methods, and open access publishing. An overwhelming majority of psychology research participants think questionable research practices (e.g. *p*-hacking, HARKing) are unacceptable (68.3–81.3%), and were supportive of practices to increase transparency and replicability (71.4–80.1%). A surprising number of participants expressed positive or neutral views toward scientific fraud (18.7%), raising concerns about data quality. We grapple with this concern and interpret our results in light of the limitations of our study. Despite the ambiguity in our results, we argue that there is evidence (from our study and others') that researchers may be violating participants' expectations and should be transparent with participants about how their data will be used.

# 1. Background

What research practices should be considered acceptable, and who gets to decide? Historically, scientists—and as a group, scientific organizations—have set the standards and have been the main drivers of change in what constitutes acceptable research practices. Perhaps this is warranted. Who better to set the standards than those who know research practices best? It seems reasonable that decisions regarding those practices should be entrusted to scientists themselves. However, there may be value in considering non-scientists' perspectives and preferences, including research participants'.

The replicability crisis in psychology has demonstrated that scientists are not always good at regulating their own practices. For example, a surprisingly high proportion of researchers admit to engaging in questionable research practices (QRPs) (as described in [1]; see also [2–4]). These include things like failing to report some of the conditions or measures in a study, excluding outliers after seeing their effect on the results, and a wide range of other practices that can be justified in some instances but also inflate rates of false positives in the published literature [5]. A large sample of social and personality psychologists reported engaging in these practices less often than 'sometimes', but more often than 'never' [6].

To combat the corrupting influence of these practices on the ability to accumulate scientific knowledge, individual scientists and scientific organizations have led the push for making research practices more rigorous and open. In the case of funding agencies, the NIH's Public Access Policy dictates that all NIH-funded research papers must be made available to the public [7].[1] Some journals and publishers have also pushed in the direction of more open scientific practices. For example, 53 journals, including some of the most sought-after outlets in psychology like *Psychological Science*, now offer open science badges, which easily identify articles that have open data, open materials or include studies that have been preregistered [8]. Although simply having badges doesn't necessarily mean the research is more open or trustworthy, there's evidence of significant increases in data sharing which may be attributable to the implementation of the badge system ([9,10]; cf. [11]).

How do scientists decide which practices are consistent with their values and norms? Currently, the norms in many scientific communities are in flux and are quite permissive regarding the use of both QRPs and open science practices. This approach of letting research practices evolve freely over time, without external regulation, tends to select for practices that produce the most valued research output. In the current system, what is most valued is often the quantity of publications in top journals, regardless of the quality or replicability of the research [12]. In short, scientists operate in a system where incentives do not always align with promoting rigorous research methods or accurate research findings. Thus, if we leave the development and evolution of research practices up to scientists alone, this may not select for practices that are best for science itself. Therefore, it may be a good idea to provide checks and balances on norms about scientific research practices, and these checks and balances should be informed by feedback from those outside the guild of science.

One way to obtain such feedback is to solicit the preferences and opinions of non-scientists, who can offer another perspective on the norms and practices in science and are likely influenced by a different set of incentives than are scientists. One such group of non-scientist stakeholders are patients suffering from specific diseases, and their loved ones, who form organized communities to advocate for patients' interests. Some of these communities, called patient advocacy groups, have pushed for more efficient use of the scarce data on rare diseases, including data sharing [13]. Other independent organizations, such as AllTrials, have also influenced scientific practices in the direction of greater transparency. With the support of scientists and non-scientists alike, AllTrials has championed transparency in medical research by urging researchers to register and share the results of all clinical trials [14]. In addition, non-scientist watchdog groups (e.g. journalists, government regulatory bodies) can call out problematic norms and practices, and push for new standards.

Another group of non-scientist stakeholders is research participants. While they have not traditionally formed communities to advocate for their interests (cf., patient advocacy groups, Amazon Mechanical Turk workers' online communities), they are also a vital part of the research process and important members of the scientific community in sciences that rely on human participants. In fact, because they are the only ones who experience the research procedure directly, research participants can sometimes have information or insight that no other stakeholder in the research process has. As such, participants might have a unique, informative perspective on the research process.

---

[1]To guarantee that future readers will have access to the content referenced here and in other non-DOI materials cited, we have compiled a list of archival links for those references (https://osf.io/26ay8/).

A fresh perspective on research practices is not the only reason to care about what participants think. One practical reason to consider research participants' preferences is that ignoring their wishes risks driving them away. Most research in psychology relies on human participants, and their willingness to provide scientists with high-quality information about themselves. Motivation to be a participant in scientific studies is varied, but besides financial compensation, altruism and a desire to contribute to scientific knowledge are common reasons people mention for participating [15,16]. If participants believe researchers are not using their data in a way that maximizes the value of their participation, they might feel less inclined to participate or participate but provide lower quality data. In addition, going against participants' wishes could undermine public trust in science even among non-participants, if they feel we are mistreating participants.

There are also important considerations regarding informed consent to take into account when thinking about research practices. Although informed consent is usually thought of in terms of how participants are treated within the context of the study, their rights also extend to how their data are used thereafter. This is explicitly acknowledged in human subjects regulations, but there has not been much attention paid to what this means for the kinds of research practices that have been the target of methodological reforms, beyond data sharing. Specifically, informed consent must contain not only a description of how the confidentiality and privacy of the subjects will be maintained but also enough information in order for participants to understand the research procedures *and their purpose* [17]. There is some ambiguity in this phrase, but it could arguably encompass the types of QRPs scientists have been debating among themselves. For example, it is conceivable that participants might have preferences or assumptions about whether researchers will filedrawer (i.e. not attempt to publish or disseminate) results that do not support the researchers' hypothesis or theory. If we take informed consent to mean that participants should have an accurate understanding of the norms and practices that the researchers will follow, and should consent to how their data will be used, it is important to understand study participants' preferences and expectations.

What should we do with what we learn about participants' expectations and preferences about how we handle their data? If participants do have views about what would and would not be acceptable for researchers to do with their data, should scientists simply let those preferences dictate our research practices completely? Clearly not. Scientists are trained experts in how to conduct research, and many of our current research practices are effective and adequate. Moreover, it is probably unreasonable to expect participants to understand all of the intricacies of data analysis and presentation. However, participants' expectations and preferences should inform our debates about the ethics and consequences of scientific practices and norms. Moreover, participants' expectations should inform our decisions about what information to provide in consent forms and plain language statements, to increase the chances that participants will be aware of any potential violations of their expectations.

There are several possible outcomes of investigating research participants' views about research practices. On the one hand, participants may feel that scientists' current research practices are acceptable. This would confirm that we are respecting our participants' wishes, and obtaining appropriate informed consent by treating participants' data in a way that is expected and acceptable to them. On the other hand, if participants find common research practices unacceptable, this may help us identify participants' misconceptions about the research process, and areas where there is a mismatch between their expectations and the reality of research.

If we do find that there is an inconsistency between participants' expectations and research practices, scientists have several options. First, they may want to listen to participants. Humans—of which scientists are a subset—are prone to motivated reasoning, and tend to have blind spots about their weaknesses, especially when they are deeply invested, a problem that a fresh perspective might alleviate. As outsiders who are familiar with the research, it is possible that participants may recognize those blind spots and areas for improvement better than researchers (particularly for 'big picture' issues that do not require technical expertise). Second, researchers may decide not to change their practices completely, but to accommodate the principle behind participants' preferences. For example, if participants want all of their data to be shared publicly, in situations where this is not possible because of re-identification risk, researchers might make an effort to share as much of the data as possible. Finally, researchers may decide that a practice that is considered unacceptable by participants is still the best way to go about doing research. In that case, better communication with participants may be needed to clarify why this practice is necessary and to honour the spirit of informed consent.

Any effort to take participants' preferences into account when engaging in research assumes participants do have preferences about the fate of their data. It is possible, however, that many

participants have weak preferences or no preferences at all. This would still be useful for researchers to know because it would increase researchers' confidence that they are not violating participants' preferences or expectations.

It is likely that at least some participants do have clear preferences about what we do with their data. On the subject of data-sharing, studies with genetic research or clinical trial participants suggest that, despite some concerns about privacy and confidentiality, a majority of participants support sharing of de-identified data, and are willing to share their own data, with some restrictions [18–20].

There is also data on what participants think about selective reporting, that is, the practice of reporting only a subset of variables or studies performed when investigating a given question, and about data fabrication. In a series of studies, Pickett & Roche [21] examined attitudes toward these practices among the general public in the United States—a population similar to research participants in many psychology studies—and among Amazon Mechanical Turk workers. Across both samples, there was high agreement that data fabrication is morally reprehensible and should be punished. Furthermore, in the Amazon Mechanical Turk sample, 71% of participants found selective reporting to be morally unacceptable, with over 60% saying researchers should be fired and/or receive a funding ban if they engage in selective reporting.

In addition to this empirical evidence, it seems intuitive that many participants would be surprised and disappointed if their data were being used in extremely unethical ways (e.g. to commit fraud, or further the personal financial interests of the researchers at the expense of accurate scientific reporting). What is less clear is whether participants care, and what they think, about a wider set of QRPs and proposed open science reforms that are currently considered acceptable, and practiced by at least some researchers, in many scientific communities.

## 1.1. Study aims

To further investigate this topic, we asked a sample of actual study participants, after their participation in another study, about how they would feel if some common research practices were applied to their own data. We did this using a short add-on survey (that we will refer to as the *meta-study*) at the end of different psychological studies (that we will refer to as the *base studies*). The meta-study asked participants to consider several research practices and imagine that they would be applied to the data they had just provided in the base study.

We asked participants about eight research practices, including QRPs and their consequences, and open science or proposed best practices, referred to here as *open science practices.* We followed two guidelines when choosing which practices to include. First, we sought to include the most common open science practices and every QRP from John *et al*. [1] that is simple enough for participants to understand without technical expertise. Second, we selected those practices we judged as most directly impacting participants' contributions. For example, filedrawering could reduce participants' perceived value of their contribution because their data may never see the light of day; *p*-hacking (repeating statistical analyses several different ways but only reporting some of them) might distort the accuracy of reported findings and decrease the value of participants' contributions; posting data publicly could increase participants' concerns about privacy. Conversely, publishing the results in an open access format would enable participants to potentially access the results of research they have contributed to, which may be important to them.

The practices we asked participants about were: (1) *p*-hacking, or cherry-picking results, (2) selective reporting of studies, (3) HARKing (hypothesizing after the results are known), (4) committing fraud, (5) conducting direct replications, (6) sharing methods ('open methods'), by which we mean making the procedure of a study clear enough that others can replicate it, (7) publishing open access papers, and (8) sharing data ('open data').

What is the best way to present these research practices to participants? One option is to describe the practice (and, in some cases, its complement) without giving any explanation for why a researcher might engage in this practice. Another option is to explain the context, incentives and trade-offs that might lead a researcher to choose to engage in this practice. We carefully considered both options and decided on the former in all but one case (data sharing, see Method below). While providing participants with context for these research practices may help them understand why scientists might engage in them, and the benefits and costs of doing so, we did not feel it would be possible to provide this context in a way that was not leading, without having participants take an hours-long course in research methods and scientific integrity. In addition, we felt that participants' naive reactions to these practices would be most informative for extrapolating to what a typical research participant thinks about these practices

(i.e. without special insight or expertise into the technical, social and political aspects of scientific research). In the light of these considerations, we asked participants for their views about these practices without providing much information about the costs and benefits of each practice (with the exception of data sharing). As a result, participants' responses should be taken to reflect their spontaneous views about these practices, which might capture ideals rather than firmly held expectations.

The goal of this study was to provide accurate estimates of research participants' views about these research practices. We had two research questions (though we did not have hypotheses about the results):

**RQ1**: What are participants' views about questionable research practices (including p-hacking, selective reporting, and HARKing) and fraud?

**RQ2**: What are participants' views about open science practices (data sharing, direct replication, open methods, open access)?

## 1.2. Scope

Because we did not have the time or resources to survey the full range of psychological science research studies, we limited our scope to minimal-risk psychology studies on English-speaking convenience samples that were run entirely on a computer or online, where all the data were provided by the participant in one session.

By including only this subset of studies, we expected to have minimal to no variance in study sensitivity, effort required for data contribution, and other characteristics of the studies which might affect participants' opinions of the examined research practices. Therefore, we recognize that we cannot explore any potential effects of these variables in this study, nor generalize the obtained results beyond the types of studies we included. However, we are able to generalize the results to other minimal-risk studies of the same kind, a common design that we believe represents a large proportion of psychology studies.

## 1.3. Pilot studies

In order to help us develop the materials for the proposed study, we conducted three pilot studies. In the first study (Pilot Study A), we aimed to gauge participants' opinions about data sharing only. In the second study (Pilot Study B), we added questions about all of the practices we planned to ask about in our proposed study, and changed the language of the data sharing question based on the results from Pilot Study A. In a third study (Pilot Study C), we fine-tuned the language used in the questions, which were almost identical to the proposed study. All materials and data for these pilots can be found at https://osf.io/bgpyc/.

With the notable exception of open access publishing[2], a majority of participants seemed to support using research best practices ('open science practices'). These preliminary results suggest that participants do have consistent opinions about these matters, which they are able to articulate. Participants overwhelmingly supported data sharing—over 70% for all versions of the question—including sharing publicly, and sharing so others can verify the claims being made or re-use the data. Sharing enough detail about the procedure of the study to allow others to replicate it (i.e. open methods) was also supported by a majority of participants. Finally, most participants (over 60% for all versions of the question) favoured replication, even when it was presented as a trade-off between replicating the same study or moving on to a new study.

Furthermore, research participants seem to have strong preferences against the use of questionable research practices, with a majority of participants—over 75% for all questions and versions, with one exception[3]—disapproving of QRPs. In fact, the proportion of participants indicating that researchers should not p-hack, filedrawer studies or HARK was similar to the proportion rating fraud as unacceptable. It is reassuring, however, that the distribution of answers was more extreme for fraud (80.7% of participants in Pilot B and 92.8% in Pilot C selected the most extreme response for fraud,

---

[2]While participants still favoured open access over publishing behind a paywall, a sizeable portion of participants selected the middle answer, indicating they were indifferent (Pilot B: 34.5%; Pilot C: 21.4%)

[3]Participants tended to see selective reporting of studies (i.e., filedrawering) less negatively when it was presented without explicitly saying the researchers reported only the results that came out the way they predicted (neutral version of the question; see https://osf.io/eyfcu/). In the UK sample (Pilot C), slightly under 70% of participants saw selective reporting as unacceptable.

versus 11.4–56.1% for the three QRPs mentioned here). Detailed results and figures for all three pilots can be found in the electronic supplementary materials.

# 2. Registered report study

The present registered report study expands our pilot studies to investigate participants' opinions about fraud and QRPs (RQ1), and open science practices (RQ2), in a much larger sample. By including both research pool participants at multiple universities and Amazon Mechanical Turk ('MTurk') workers— two groups that make up a large proportion of psychology research participants—we can improve generalizability as well as explore any preference discrepancies between undergraduate participants and MTurkers. Based on the pilot results, we honed our questions to more adequately measure participants' preferences, with as little interference or bias as possible. Finally, including a larger selection of minimal-risk base studies improves the generalizability of the results to other minimal-risk studies.

## 2.1. Method

### 2.1.1. Participants

We aimed to collect data from both online platforms and undergraduate student populations. In computing our target sample size, we chose a simple target analysis—estimating proportions (e.g. proportion of participants who chose a response above the 'indifferent' midpoint on a given question). Specifically, we aimed for enough precision such that the width of our 95% confidence interval would be at least as narrow as ±3% when the proportion is equal for all categories (precision is higher for uneven proportions). To achieve this, our precision analysis suggests that our target sample size should be 1317 participants—see https://osf.io/v68hu/ for R code and the supplementary materials for details on this calculation. We aimed for a sample of (1) approximately 50% online participants and 50% undergraduate student participants (2) from at least three universities (for the student sample) and (3) at least eight different base studies. However, it was difficult to be sure we would be able to achieve this breakdown at the subgroup level because we relied on cooperation with other researchers (see below). To ensure we would be able to compare online and undergraduate participants' views, we set a maximum of 60% of participants from either population. Although the exact breakdown of online versus student participants might vary within this range, we planned to collect data from at least 1600 participants before exclusions (see supplement on precision calculation for details on how we arrived at this number). Data were collected by base study (i.e. we continued to seek out new base studies and collect the full sample size agreed upon for that study) and we stopped seeking out new base studies when, after completing data collection for a base study, these targets had been reached. After that, we finished collecting the planned sample for any base studies that were still ongoing, but did not begin any new data collection. An explanation of how participants were compensated for their time can be found in the electronic supplementary materials.

### 2.1.2. Study selection

We used two main strategies to acquire base studies for our sample. First, we asked researchers whom we know personally or heard about who had extra time in their studies to add our questions to the end of their survey, as we did for Pilot Study A. Second, we offered to run an agreed upon number of participants ourselves using other people's base studies, either on MTurk or on the UC Davis student subject pool, in exchange for adding our questions to the end of their study. These scenarios could happen alone or in combination. That is, for some base studies, it is possible only the base-study researcher collected data, only our team collected data, or both teams collected data. To find these researchers, we planned to use the Study Swap website (osf.io/meetings/studyswap/), social media and personal contacts.

We decided which studies to include in our sample on a case-by-case basis. The base studies had to meet the following criteria: (1) a minimal-risk study where all the data would be collected in a single session, either online or on a local computer; (2) the study was run in English, and, if it used an undergraduate subject pool, it was run at a college or university where English is the primary language of instruction; (3) the participants were recruited from either college/university subject pools or online platforms and meet our inclusion criteria (see below); (4) feasibility constraints—e.g.

**Table 1.** Description of survey questions. *Note.* Each participant saw only one version of each question. See materials for a full description of the question wording, versions, and response options.

| question number | question topic | number of versions |
| --- | --- | --- |
| 1 | *p*-hacking or cherry-picking results | 2 versions |
| 2 | selective reporting of studies (filedrawering) | 2 versions |
| 3 | HARKing | 1 version |
| 4 | fraud | 1 version |
| 5 | direct replication | 1 version |
| 6 | open methods | 2 versions |
| 7 | open access publication | 1 version |
| 8 | data sharing | 2 versions |

whether we had the resources to run participants on our end, whether the IRB approval would be easy to obtain, etc.; (5) progress of our sample size goals—e.g. if we had met our goal for student or online participants, we stopped collecting data from that population; (6) time constraints—we would be able to complete data collection for the study within the time frame allotted for the project; (7) sample size—the study would provide a minimum of 50 participants; and (8) base study materials would be made publicly available.

### 2.1.3. Sample selection criteria

Participants had to speak English and be at least 18 years old. They also had to qualify for and complete the base study that preceded ours, so our study inclusion criteria included the inclusion criteria used by each base study to which we appended our meta-study. For example, if one of the studies selected only first-generation college students, or only women, this was also a criterion to participate in our meta-study for that subsample.

We had funds to collect data on MTurk and resources to collect data from the UC Davis undergraduate subject pool, so data collection conducted by us came from one of these two populations. For MTurk samples recruited by us, we planned for participants to meet the following criteria: (1) be located in the United States; (2) have a Human Intelligence Task (HIT) approval rate of 90% or higher[4]; and (3) have at least 10 HITs approved. MTurk samples recruited by partner researchers running base studies would follow that team's criteria.

It was also possible that some data would be collected by the base study researchers, and these data could be collected from other colleges' or universities' subject pools, or online platforms other than MTurk, like Prolific (https://prolific.co/). In these cases, we planned for the selection criteria for participants (beyond the requirement that participants speak English and be at least 18 years old) to be decided by the base study researchers.

### 2.1.4. Meta-study

The meta-study asked participants to consider an anonymized version of the data they had just provided in the base study, and imagine a series of hypothetical situations in which researchers use different research practices on their data. Specifically, we asked them their opinions on the eight practices shown in table 1. We honed the wording of these questions using the data and feedback from our pilot studies, which we describe in detail in the supplementary materials. The full text for the questions in table 1 can be found at https://osf.io/p8n9w/.

Our goal was to ask the questions in a way that is not leading. When we could not find a way to do this while still providing a clear description of the practice (i.e. for Questions 1, 2 and 8—see table 1 for a list of which questions correspond to which research practice), we wrote two different versions of the

---

[4]A HIT, or 'human intelligence task', is a task available for Amazon Mechanical Turk workers. A workers' HIT approval rate is the proportion of tasks that have been approved by the requester. The authors consider 90% to be a reasonable cutoff to ensure high quality data.

question reflecting the trade-off between providing a fuller but potentially leading description of the practice, and providing a vaguer but less valenced description of the practice. For Question 6, we also created two versions: the 'positive' version of the question, asking participants how they would feel if the researchers shared enough details about their materials and procedures for others to conduct a replication study, or the 'negative' version, which asks participants how they would feel if researchers did not share enough details. If the answers differ by version, which the pilot studies suggested might happen, we would have estimates of the distribution of responses to these practices for two different, but hopefully reasonable, ways to ask the same question. In other words, these two versions provide a kind of robustness check across variations that we hope capture similar phenomena. For questions 1, 2, 3 and 4, the research practices were described in simple terms, and participants were asked to rate each practice on a 5-point scale with anchors at −2 ('definitely not acceptable') through 0 ('indifferent') to +2 ('definitely acceptable').

Question 5 asked participants their opinion about whether researchers should attempt to replicate a finding before publishing it or simply move on to a new project. With this question, we hoped to make the trade-offs involved in conducting a direct replication (versus not conducting one) clear, without leading participants toward one answer or the other. Participants answered on a 5-point scale with anchors being 'strongly prefer that the researchers move on to their next project', 'slightly prefer that the researchers move on to their next project', 'indifferent', 'slightly prefer that the researchers replicate the study', and 'strongly prefer that the researchers replicate the study'.

Questions 6, 7 and 8 asked participants to consider situations where researchers can choose to use open science practices. For Question 6 (open methods), which is about whether researchers should share their materials and procedures, participants were asked to rate this practice on a 5-point scale with anchors at −2 ('feel strongly that the researchers should *not* do this') through 0 ('indifferent') to +2 ('feel strongly that the researchers *should* do this'). There were two versions of this question. The positive version describes researchers providing all necessary information for replication, while the negative version (reverse scored) describes not providing enough information.

For Question 7, participants were asked whether they have a preference for where the article reporting the results of the base study should be published: an open access journal versus a pay-walled journal. Participants answered on a 5-point scale with anchors being 'strongly prefer that it cost about $30 to read the article', 'slightly prefer that it cost about $30 to read the article', 'indifferent', 'slightly prefer that the article be free to read', and 'strongly prefer that the article be free to read'. The value of $30 is typical of several top journals in psychology. Based on feedback on our pilot materials, we also added a clarification statement so respondents understand that the value paid for the article does not go to the authors of the article—a reasonable but false assumption—but to the publisher.

For Question 8, we asked two versions of the question: one that explicitly stated reasons why a researcher may or may not want to share their data ('reasons provided'), and one that did not ('neutral'). The reasons-provided version spells out the main reasons for and against data sharing. We developed this list of reasons by consulting published work on researchers' stated reasons for sharing or not sharing data [22]. The neutral version of this question asks participants to consider potential reasons before answering, and makes it clear that valid reasons exist both for and against data sharing. Participants answered on a 5-point scale with anchors at −2 ('feel strongly that the researchers should *not* do this') through 0 ('indifferent') to +2 ('feel strongly that the researchers *should* do this').

For all of the questions, we used a 5-point response scale. This was changed from a 7-point scale in Pilots B and C, as we believe this better reflects the granularity of judgement that is reasonable to expect from research participants. Moreover, having fewer response options gives us more precision when estimating the proportion of people who choose each response option. We also changed the order and anchors for some of the questions. For questions where it makes sense to have a negative and positive end of the scale (e.g. 'researchers should *not* do this' versus 'researchers *should* do this') we kept the numbering (−2 to +2) with anchors at the ends and midpoint. However, some of the questions represent trade-offs (e.g. whether to publish open access versus behind a paywall) which have no clear 'positive' or 'negative' end. Therefore, we labelled all 5 points for Questions 5 ('direct replication') and 7 ('open access publishing') with words rather than numbers, to avoid inadvertently conveying that one end of the scale is more desirable than the other (e.g. higher numbers, or positive numbers).

For questions which have two versions, participants were randomly assigned to answer one or the other. Random assignment was independent between questions; e.g. a participant who was assigned

to the neutral version of the data sharing question could be assigned to either the neutral or the reasons-provided version of the selective reporting question. Furthermore, the order of the eight questions was randomized.

Finally, we asked additional questions for potential exploratory analyses. First, we asked about demographics, including gender, race and ethnicity, year of birth, education, proximity to science and the number of psychology studies the participant participated in during the previous two weeks. We also measured trust in psychological science with three statements ('Findings from psychology research are trustworthy', 'I have very little confidence in research findings from psychology' (reverse-scored), and 'I trust psychology researchers to do good science') which participants were asked to rate on a 7-point scale from (1) 'strongly disagree' to (7) 'strongly agree'. We also asked participants 'Have you heard of the replication crisis in psychology?' (Yes/No). If participants answered yes, we then asked them 'Please describe what you have heard about the replication crisis:' and provided an open-ended text box for their response. Although we did not have planned analyses that use these additional questions, they were collected to allow for exploratory analyses both by the authors and others who wish to re-use the data.

### 2.1.5. Data exclusion criteria

The survey included one open-ended attention/comprehension check. We planned for these answers to be coded by an independent coder, who would be blind to how they related to the rest of the data, as 'appropriate', 'inappropriate' and 'unclear'. Only 'inappropriate' answers would be excluded.

## 2.2. Results

### 2.2.1. Sample

The data were collected between January and October of 2021, yielding a total of 1990 observations before exclusions from eight different base studies. After performing the preregistered exclusions, we obtained a final sample of 1873 participants—40% from participants in Amazon Mechanical Turk studies (five studies) and 60% from university subject pool study participants (three studies across four subject pools)—with the breakdown described in table 2.

57.9% of participants described themselves as female, 40.4% described themselves as male, 0.8% self-identified as non-binary or a third gender, 0.3% preferred to self-describe and 0.5% preferred not to report their gender. The median year of birth for participants was 1999 (IQR = 14; range = 1946–2003). Participants could select multiple race and ethnicity categories; 51.8% identified as white, 29.2% as Asian, 13.5% as Hispanic or Latino or Chicano or Puerto Rican, 8.0% as Black or African American, 1.6% Middle Eastern or North African, 0.8% American Indian or Alaska Native, 0.7% Native Hawaiian or Pacific Islander and 1.4% said they had another identity; 7.5% of participants declined to self-identify on race and ethnicity.

#### 2.2.1.1. Deviations from Planned Recruiting Strategy
In the base-study recruiting phase, we communicated with several potential base studies, and received responses from many researchers willing to collaborate. Those not mentioned here did not meet our inclusion criteria, or the collaborating researcher later decided against following through for a variety of reasons, and we never reached the data collection stage with these studies. The one exception to this was a study for which we did not have enough information to realize it did not meet our inclusion criteria until after data collection was completed, so although we do have the participants' data for this other study, we are not including it or its data here. One other slight deviation from our recruiting plan was that BS03 did not have an initial agreed-upon sample size, but an end date (31 October) when the collaborating researchers had preregistered to check whether they had enough data to perform their analyses; this served as the stopping rule for our part of the study. Finally, some MTurk studies ended up with a few more observations than we aimed to collect, and BS02 had 10 fewer observations than agreed due to reaching the end of their semester.

### 2.2.2. Analyses

Our primary analyses examine the distribution of participants' responses, which we examined using descriptive statistics presented in tables 4 and 5. We also present the corresponding visualizations

**Table 2.** Sample size, population, and short description of each base study.

| base study | sample size | | | population | study description |
|---|---|---|---|---|---|
| | before exclusions | after preregistered exclusions only | after non-preregistered (strict) exclusions | | |
| BS01 | 500 | 499 | 437 | Sacramento State University and University of California, Davis Subject Pools | a study about individuals' reactions to marginalized individuals in positions of power |
| BS02 | 390 | 389 | 363 | University of Pennsylvania and University of California, Davis Subject Pools | a study exploring the reasons that people overassess experts' abilities |
| BS03 | 237 | 237 | 227 | Princeton University Subject Pool | a study about friendship formation and related attitudes |
| BS04 | 162 | 145 | 93 | Amazon Mechanical Turk Workers | a study about interviews in false confessions documentaries and how they influence laypeople's perceptions of a confession |
| BS05 | 252 | 201 | 115 | Amazon Mechanical Turk Workers | a study about perspective taking of climate refugees |
| BS06 | 106 | 100 | 86 | Amazon Mechanical Turk Workers | a study about the relationships between dark personality, self-control and aggression |
| BS07 | 130 | 123 | 99 | Amazon Mechanical Turk Workers | a study testing interindividual variability in free- and cued-recall memory performance |
| BS08 | 213 | 179 | 117 | Amazon Mechanical Turk Workers | a study about people's perceptions of the most moral, least moral, and morally average people they personally know |
| Total | 1990 | 1873 | 1537 | — | — |

of the distributions of responses in figures 1 and 2. We first report the descriptives using the preregistered exclusion criteria, starting with results for the combined samples (table 4 and figure 1), then results for each question version separately (for questions that had more than one version; table 5 and figure 2). We also report the exploratory analyses we outlined in the stage 1 manuscript.

Despite our relatively strict preregistered exclusion criteria, we nevertheless found certain patterns of results suspicious, especially in the percentage of participants who expressed neutral or positive views of scientific fraud. Because of this, in our 'exploratory analyses not described in the preregistration' section, we repeat most of the analyses with non-preregistered but stricter exclusion criteria, which we fully describe at the beginning of the section. With these stricter criteria, we aimed to provide an alternative test of our research questions, and we suspect that some readers might feel these are more appropriate results to interpret, given our possible data quality issues. We have clearly marked these results as exploratory. All the code used in the analyses and figures presented here can be found at https://osf.io/34gbv/; this follows and expands the stage 1 preregistered analyses which can be found at https://osf.io/ytdek/.

### 2.2.3. Preregistered main analyses

For each question, we were interested in the proportion of participants that selected a negative, neutral, and positive response. Table 3 details the response scales for each question and its labels.

**Table 3.** Response scale anchors for each question. *Note.* Anchor numbers were not shown for questions 5 and 7.

| question number | question topics | response scale anchors |
|---|---|---|
| 1, 2, 3, 4 | *P*-hacking, filedrawering, HARKing, fraud | −2: definitely not acceptable |
| | | 0: indifferent |
| | | +2: definitely acceptable |
| 5 | direct replication | −2, −1: [strongly/slightly] prefer that the researchers move on to their next project |
| | | 0: indifferent |
| | | +2, +1: [strongly/slightly] prefer that the researchers replicate their study |
| 6, 8 | open methods, data sharing | −2: feel strongly that the researchers should **not** do this |
| | | 0: indifferent |
| | | +2: feel strongly that the researchers **should** do this |
| 7 | open access publication | −2, −1: [strongly/slightly] prefer that it cost about $30 to read the article |
| | | 0: indifferent |
| | | +2, +1: [strongly/slightly] prefer that the article be free to read |

### 2.2.3.1. Overall results: Preregistered exclusion criteria

As can be seen in table 4 and figure 1, for all eight questions, a clear majority of participants selected a response on one side of the neutral point. That is, between 68% and 81% of participants reported that *p*-hacking, filedrawering, HARKing, and fraud are not acceptable, that they prefer that researchers share their methods and data, that replication is preferable to moving on without replicating, and that publishing open access is preferable to publishing behind a paywall. Fewer than 15% of participants selected the neutral option (indifferent) for each question, except for the open access publishing question, for which 25% of participants selected the neutral option. Participants' preferences/opinions were most pronounced for replication and fraud, though a troubling percentage of participants (19%) expressed indifferent or positive attitudes about fraud. We return to this unexpected pattern of results below, in the non-preregistered section.

**Table 4.** Descriptive statistics for each question, with preregistered exclusions, collapsing across question version. *Note*. N = 1873 for all questions. Multinomial 95% confidence intervals [LL, UL] using the Sison-Glaz method. 'Rs' in questions 6 and 8 refers to 'researchers'. Each response category except 'indifferent' collapses across two response options on the 5-point scales.

| question | median (IQR) | category | % [LL, UL] |
|---|---|---|---|
| question 1: *p*-hacking/cherry-picking results | −1 (1) | not acceptable | 68.3 [66.2, 70.5] |
| | | indifferent | 7.42 [5.29, 9.56] |
| | | acceptable | 24.2 [22.1, 26.4] |
| question 2: selective reporting of studies/ filedrawering | −1 (2) | not acceptable | 69.2 [67.1, 71.3] |
| | | indifferent | 7.79 [5.71, 9.94] |
| | | acceptable | 23.0 [20.9, 25.2] |
| question 3: HARKing | −1 (2) | not acceptable | 68.7 [66.6, 70.9] |
| | | indifferent | 8.38 [6.30, 10.5] |
| | | acceptable | 22.9 [20.8, 25.1] |
| question 4: fraud | −2 (0) | not acceptable | 81.3 [79.6, 83.1] |
| | | indifferent | 4.00 [2.30, 5.75] |
| | | acceptable | 14.7 [13.0, 16.4] |
| question 5: direct replication | 1 (1) | move on | 7.69 [5.98, 9.49] |
| | | indifferent | 12.2 [10.5, 14.0] |
| | | replicate | 80.1 [78.4, 81.9] |
| question 6: open methods | 1 (2) | Rs should not do this | 14.9 [13.0, 17.0] |
| | | indifferent | 12.0 [10.0, 14.0] |
| | | Rs should do this | 73.0 [71.1, 75.0] |
| question 7: open access publication | 1 (2) | paywall | 4.11 [2.08, 6.22] |
| | | indifferent | 24.5 [22.4, 26.6] |
| | | free | 71.4 [69.4, 73.5] |
| question 8: data sharing | 1 (1) | Rs should not do this | 10.9 [9.08, 12.9] |
| | | indifferent | 13.2 [11.3, 15.1] |
| | | Rs should do this | 75.9 [74.0, 77.8] |

### 2.2.3.2. Results for question versions: Preregistered exclusion criteria

For the four questions with multiple versions, we examined the descriptive statistics and distribution of responses separately for each version (table 5 and figure 2). As preregistered, we did not conduct inferential tests comparing the two versions of each question, as we did not have hypotheses regarding the effect of version. Instead, we provide the results separately for each version to provide a sense of the robustness of results across question formats.

As shown in table 5 and figure 2, participants reported more extreme views about *p*-hacking and filedrawering when these practices were described as motivated (i.e. 'only reported the results/studies that came out the way they predicted'), compared to participants who saw these practices described in a neutral manner (i.e. 'only reported some of the results/studies'). However, in both versions, the majority of participants rated these practices as not acceptable. For open methods, the question framing ('provided a lot of details […] other researchers could easily conduct a replication' versus 'did not provide a lot of details […] other researchers could not easily conduct a replication'; responses to the second version were reverse-scored) led to slightly different distributions, with more participants supporting closed methods in the second version. Finally, for the data sharing question, one version of the question did not present any reasons why researchers might choose to share or not to share their data, while the other version presented reasons for both choices. Across both versions, most participants selected responses in favour of data sharing, but participants who read about reasons for and against data sharing had slightly less extreme views in favour of data sharing than did participants who did not read reasons.

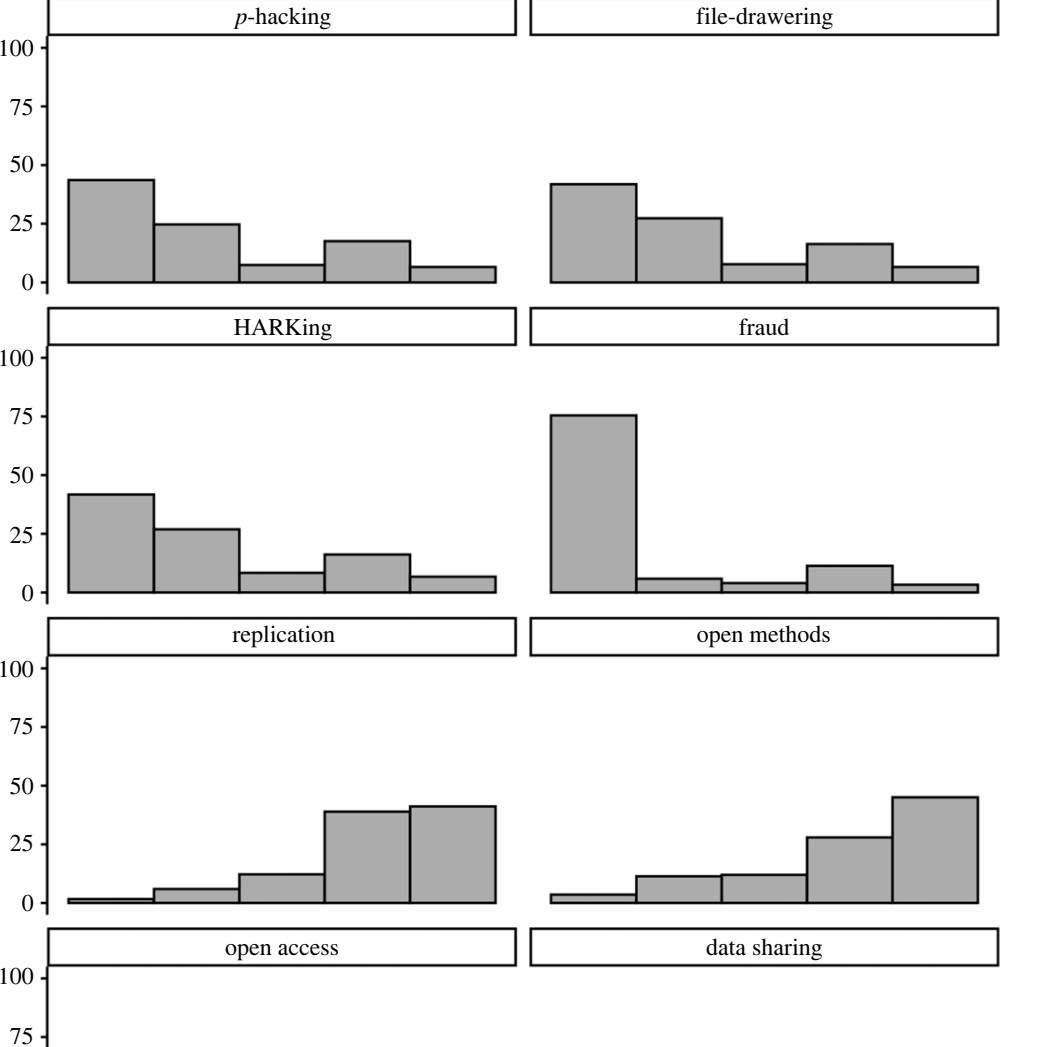

**Figure 1.** Distribution of participants' answers for each question. For the top four panels, negative numbers indicate that participants found the practice unacceptable while positive numbers indicate they found the practice acceptable. For the bottom four panels, higher numbers indicate more support for the practice. $N = 1873$. See also table 4.

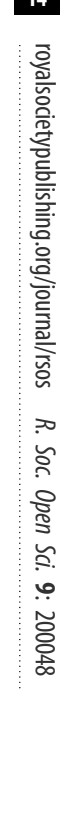

**Figure 2.** Distribution of participants' answers for each question, by question version, for the four questions with two versions. For the top two panels, negative numbers indicate that participants found the practice unacceptable while positive numbers indicate they found the practice acceptable. For the bottom two panels, higher numbers indicate more support for the practice. For $p$-hacking and filedrawering, the neutral version described the behaviour only (i.e. researchers 'only reported some of the results' or 'did not report all of the studies they ran'), while the motive version implied motivated reasons behind the selective reporting of results or studies (e.g. 'only reported the results/studies that came out the way they predicted'). The positive and negative versions of the open methods question were phrased as '[researchers] provided (versus did not provide) a lot of details about how they did the study. Therefore, other researchers could (versus could not) easily conduct a replication...'. The neutral data sharing question asked whether participants thought 'researchers should share the dataset when they publish their results' while the reasons version asked the same but provided some reasons why researchers may or may not want to share their data (e.g. concerns about scooping or making it possible for others to verify their work). See table 5 for more detailed results and sample sizes.

**Table 5.** Descriptive statistics for questions with two versions, with preregistered exclusions only. *Note*. The median and interquartile range reported for Question 6, closed methods version, is after the question was reverse scored. Multinomial 95% confidence intervals [LL, UL] using the Sison-Glaz method. 'Rs' in questions 6 and 8 refers *to 'researchers'*. Each response category except 'indifferent' collapses across two response options on the 5-point scales.

| question | category | % [LL, UL] | |
|---|---|---|---|
| question 1: *p*-hacking/ | | *neutral (%)* | *motive (%)* |
| cherry-picking results | | (*Mdn* = −1, *IQR* = 3) | (*Mdn* = 2, *IQR* = 2) |
| | | *n* = 934 | *n* = 939 |
| | not acceptable | 69.5 [66.6, 72.7] | 73.5 [70.7, 76.4] |
| | indifferent | 8.18 [5.23, 11.3] | 7.05 [4.25, 9.99] |
| | acceptable | 22.3 [19.3, 25.4] | 19.5 [16.7, 22.4] |
| question 2: selective | | *neutral (%)* | *motive (%)* |
| reporting of studies/ | | (*Mdn* = −1, *IQR* = 2) | (*Mdn* = −2, *IQR* = 2) |
| filedrawering | | *n* = 950 | *n* = 923 |
| | not acceptable | 66.8 [63.9, 69.9] | 71.6 [68.8, 74.6] |
| | indifferent | 9.37 [6.42, 12.5] | 6.18 [3.36, 9.18] |
| | acceptable | 23.8 [20.8, 26.9] | 22.2 [19.4, 25.2] |
| question 6: open | | *Rs use open methods (%)* | *Rs use 'closed' methods (%)* |
| methods | | (*Mdn* = 2, *IQR* = 1) | (*Mdn* = 1, *IQR* = 2) |
| | | *n* = 907 | *n* = 966 |
| | Rs should not do this | 7.83 [5.29, 10.5] | 68.1 [65.2, 71.1] |
| | indifferent | 13.9 [11.4, 16.6] | 10.2 [7.35, 13.3] |
| | Rs should do this | 78.3 [75.7, 81.0] | 21.6 [18.7, 24.6] |
| question 8: data sharing | | *neutral (%)* | *reasons provided (%)* |
| | | (*Mdn* = 1, *IQR* = 1) | (*Mdn* = 1, *IQR* = 1) |
| | | *n* = 926 | *n* = 947 |
| | Rs should not do this | 8.96 [6.48, 11.5] | 12.9 [10.1, 15.8] |
| | indifferent | 11.3 [8.86, 13.8] | 15.0 [12.2, 17.9] |
| | Rs should do this | 79.7 [77.2, 82.2] | 72.1 [69.4, 75.0] |

### 2.2.4. Exploratory analyses

#### 2.2.4.1. Exploratory analyses described in preregistration

In the preregistration, we described that we would also conduct exploratory analyses examining: (1) variance partitioning of responses to identify the proportion of variance that was between-studies and (2) differences between the MTurk and subject pool participants' responses.

*2.2.4.1.1. Variance partitioning.* As shown in table 6, the variance in each outcome that can be attributed to between-study variability ranged between 0.2 and 15.7% with preregistered exclusions. For questions 1–4, related to QRPs, between-study variance seemed to be higher, ranging from 11.1% to 15.7%, while it was lower for positive research behaviors like replication, open access publishing and data sharing. The open methods question is a notable exception, although that might be related to the fact that, while the positive version of this question asks about a positive research behaviour—making one's methods open—the negative version asks about a questionable behaviour—making one's methods 'closed'. This pattern is less pronounced but still clearly visible after performing non-preregistered strict exclusions (table 6, right column), potentially indicating more consensus between studies for positive practices than questionable practices. We should note, however, that this between-study variance includes variance due to the different platforms (Amazon MTurk versus university subject pools), as each study was conducted on only one of these platforms. Thus, between-study variance could be

**Table 6.** Variance partitioning. Proportion of between-study variance in each outcome variable.

| question | between-study variance (after preregistered exclusions only) (%) | between-study variance (after stricter, non-preregistered exclusions) (%) |
| --- | --- | --- |
| question 1: *p*-hacking/ cherry-picking results | 11.1 | 4.7 |
| question 2: selective reporting of studies/filedrawering | 11.9 | 3.5 |
| question 3: HARKing | 12.7 | 4.1 |
| question 4: fraud | 15.7 | 3.6 |
| question 5: direct replication | 1.6 | 1.5 |
| question 6: open methods | 8.4 | 3.7 |
| question 7: open access publication | 1.3 | 2.1 |
| question 8: data sharing | 0.2 | 0.7 |

driven by between-platform variance. We examine differences between the responses from MTurk and subject pool participants in the next section.

*2.2.4.1.2. MTurk* versus *subject pool.* For all questions except data sharing (Question 8), there were statistically significant differences between the MTurk and subject pool samples (all *ps* < 0.001; see electronic supplementary material, table S4 for detailed results of Pearson's Chi-squared tests). These results suggest that students may hold more extreme views than MTurk participants when it comes to research practices, more strongly disapproving of questionable practices and supporting open practices. The same pattern holds after implementing the non-preregistered, strict exclusions (which are described in detail below). The distribution of responses (after strict exclusions) is shown in figure 3 and reported in table 7. See electronic supplementary material, figure S17 for an analogous visualization to figure 3 but with preregistered exclusions only.

### 2.2.4.2. Exploratory analyses not in the preregistration

As mentioned above, the finding that a surprisingly high proportion of participants reported having neutral (4%) or positive (15%) views about fraud concerned us. To be clear, the item wording did not leave much room for misinterpretation. Specifically, we asked participants to 'Imagine that the researchers changed the data to make the results come out the way they predicted (in other words, committed scientific fraud). Do you think this would be acceptable?' In our pilot studies, we found that only around 2 to 10% of participants expressed positive or neutral attitudes toward fraud. Similarly, Pickett & Roche [21] found that 96% of a sample of MTurk participants believe data fabrication and fraud are morally unacceptable, and 91% of a representative US sample believe data fabrication and fraud should be a crime. Based on these previous results and on common sense, we were skeptical that 19% of participants truly have a neutral or positive view of research fraud.

Even more alarming, when we look at the distribution of responses for the MTurk and subject pool participants separately, we find that a whopping 32.8% of MTurk participants expressed neutral or positive attitudes toward fraud. In the subject pool subsample, only 9% of participants expressed neutral or positive attitudes toward fraud, in line with our pilot studies and expectations. Thus, we suspect that the exclusion criteria we preregistered were not sufficiently strict, and that many non-serious responses remained in the MTurk samples, even after the preregistered exclusions.

Of course, estimating the proportion of participants who believe fraud is acceptable was one of the aims of the current study, so deciding to change our exclusion criteria after seeing the results seriously increases the risk of bias of any subsequent analyses. We must seriously consider the possibility that a non-trivial fraction of participants believe it is acceptable for researchers to commit scientific fraud, at least in the context of simple, low-risk psychology studies, and we address this possibility in the discussion. Nevertheless, if we are correct that many of the participants expressing neutral or, especially, positive views of fraud are not responding seriously, including them in our analyses affects

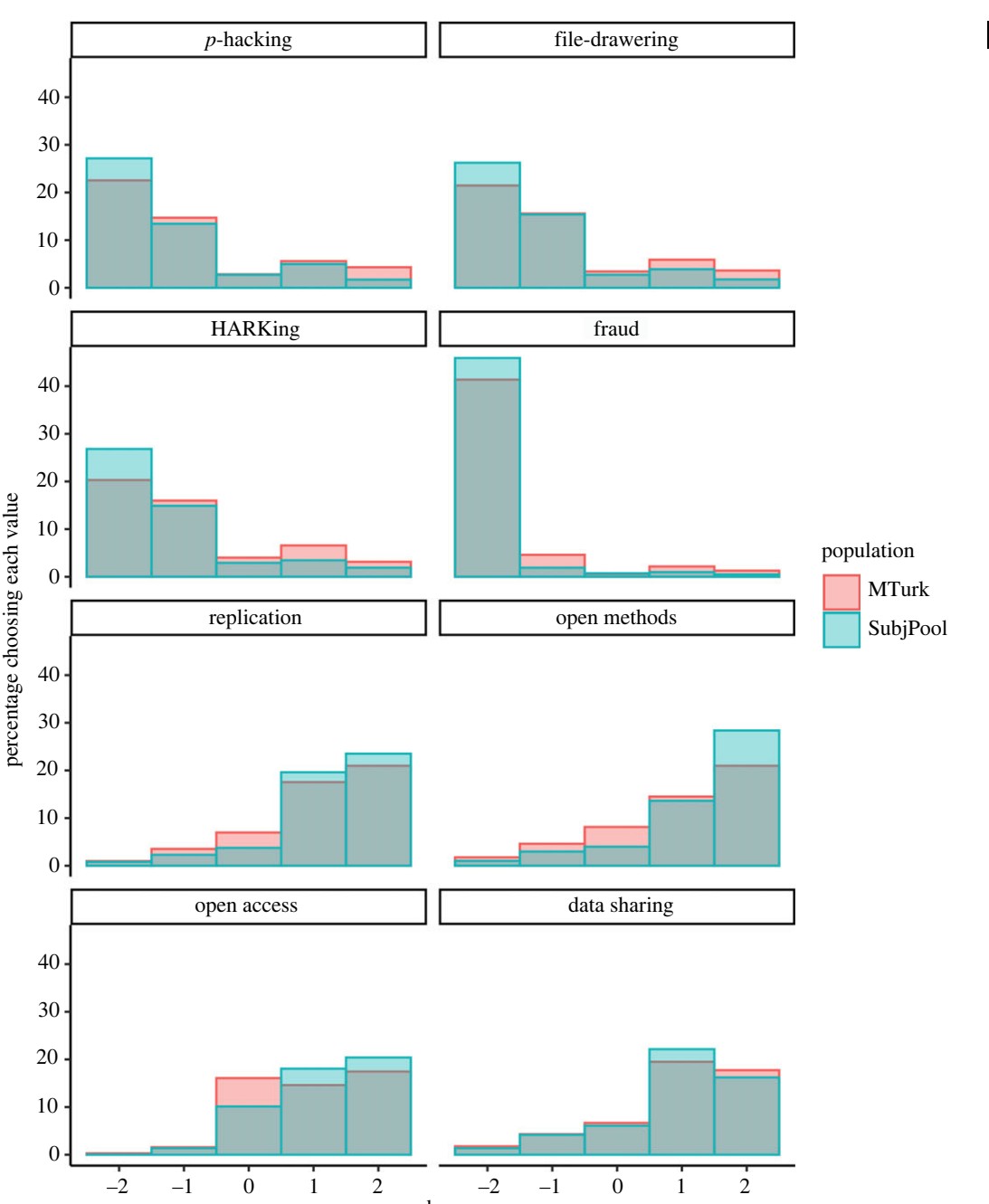

**Figure 3.** Distribution of university subject pool and MTurk participants' responses for all eight questions, after non-preregistered, strict exclusions. For the top four panels, negative numbers indicate that participants found the practice unacceptable while positive numbers indicate they found the practice acceptable. For the bottom four panels, higher numbers indicate more support for the practice. $N = 1{,}537$. See also table 7.

the accuracy of all other estimates reported here. Thus, we believe it is prudent to also explore stricter exclusion criteria and examine the results for all eight questions in this smaller subsample.

In exploring other possible ways to identify low quality or non-serious responders, in addition to the preregistered exclusion criteria, we examined our data from many different angles. We considered how to use the open-ended responses to filter out non-serious responders; we examined the distributions to identify unexpected bumps (e.g. we found that, before reverse-scoring, all questions and versions showed an unexpected bump in the distribution of responses for the response option corresponding to '+1' on the −2 to +2 scale, but only in the MTurk samples); and we examined the effects of various exclusion criteria on the number of participants excluded, and the proportion of remaining participants reporting neutral

**Table 7.** Descriptive statistics for all eight questions, with non-preregistered (strict) exclusions, by population (university subject pool students versus Amazon Mechanical Turk workers). *Note*. All differences between MTurk & subject pool are significant (Pearson's Chi-squared test) at $p < 0.001$ for all questions except question 8 (not significant). See electronic supplementary material, Table S4 for details. Multinomial 95% confidence intervals [LL, UL] using the Sison-Glaz method. 'Rs' in questions 6 and 8 refers to 'researchers'. Each response category except 'indifferent' collapses across two response options on the 5-point scales.

| question | category | % [LL, UL] | |
|---|---|---|---|
| | | subject pool (%) $n = 1027$ | MTurk (%) $n = 510$ |
| question 1: *p*-hacking/ cherry-picking results | not acceptable | 81.2 [79.0, 83.6] | 74.5 [71.0, 78.4] |
| | indifferent | 5.45 [3.21, 7.83] | 5.69 [2.16, 9.60] |
| | acceptable | 13.3 [11.1, 15.7] | 19.8 [16.3, 23.7] |
| question 2: selective reporting of studies/ filedrawering | not acceptable | 83.3 [81.1, 85.5] | 74.1 [70.4, 77.9] |
| | indifferent | 5.45 [3.31, 7.68] | 6.86 [3.14, 10.6] |
| | acceptable | 11.3 [9.15, 13.5] | 19.0 [15.3, 22.8] |
| question 3: HARKing | not acceptable | 83.4 [81.3, 85.6] | 72.5 [68.8, 76.5] |
| | indifferent | 5.84 [3.70, 8.04] | 8.04 [4.31, 12.0] |
| | acceptable | 10.7 [8.57, 12.9] | 19.4 [15.7, 23.3] |
| question 4: fraud | not acceptable | 95.6 [94.5, 96.8] | 92.0 [89.8, 94.2] |
| | indifferent | 1.46 [0.399, 2.66] | 1.18 [0.00, 3.39] |
| | acceptable | 2.92 [1.85, 4.12] | 6.86 [4.71, 9.08] |
| question 5: direct replication | move on | 6.23 [4.28, 8.25] | 9.02 [5.69, 12.7] |
| | indifferent | 7.50 [5.55, 9.52] | 13.9 [10.6, 17.6] |
| | replicate | 86.3 [84.3, 88.3] | 77.1 [73.7, 80.8] |
| question 6: open methods | Rs should not do this | 7.98 [5.94, 10.2] | 12.7 [9.02, 16.8] |
| | indifferent | 7.98 [5.94, 10.2] | 16.3 [12.5, 20.3] |
| | Rs should do this | 84.0 [82.0, 86.2] | 71.0 [67.3, 75.0] |
| question 7: open access publication | paywall | 2.82 [0.29, 5.41] | 3.73 [0.00, 8.12] |
| | indifferent | 20.3 [17.7, 22.8] | 32.2 [28.0, 36.5] |
| | free | 76.9 [74.4, 79.5] | 64.1 [60.0, 68.5] |
| question 8: data sharing | Rs should not do this | 11.1 [8.67, 13.7] | 12.2 [8.63, 16.0] |
| | indifferent | 12.2 [9.74, 14.8] | 13.3 [9.80, 17.2] |
| | Rs should do this | 76.7 [74.3, 79.3] | 74.5 [71.0, 78.3] |

**Table 8.** Percentage of participants expressing neutral or positive attitudes toward scientific fraud, divided by population (MTurk versus university subject pool), after preregistered exclusions only and after non-preregistered (strict) exclusions.

| population | % reporting neutral or positive attitudes toward scientific fraud | |
|---|---|---|
| | after prereg exclusions only | after non-prereg, strict exclusions |
| subject pool | 9.3 | 4.4 |
| MTurk | 32.8 | 8.0 |

and positive attitudes toward fraud. That is, our process was iterative and very much data-dependent, and thus the results of all analyses after applying these non-preregistered exclusion criteria should be taken as susceptible to our biases. Limitations related to these decisions are discussed below.

**Table 9.** Descriptive statistics for each question, with non-preregistered (strict) exclusions, collapsing across question version. *Note.* N = 1537 for all questions. Multinomial 95% confidence intervals [LL, UL] using the Sison-Glaz method. 'Rs' in questions 6 and 8 refers to 'researchers'. Each response category except 'indifferent' collapses across two response options on the 5-point scales.

| question | median (IQR) | category | % [LL, UL] |
|---|---|---|---|
| question 1: *p*-hacking/cherry-picking results | −2 (1) | not acceptable | 79.0 [77.0, 81.0] |
| | | indifferent | 5.53 [3.58, 7.57] |
| | | acceptable | 15.5 [13.5, 17.5] |
| question 2: selective reporting of studies | −1 (1) | not acceptable | 80.2 [78.3, 82.2] |
| | | indifferent | 5.92 [4.03 7.91] |
| | | acceptable | 13.9 [12.0, 15.8] |
| question 3: HARKing | −1 (1) | not acceptable | 79.8 [77.9, 81.8] |
| | | indifferent | 6.57 [4.68, 8.58] |
| | | acceptable | 13.6 [11.7, 15.6] |
| question 4: fraud | −2 (0) | not acceptable | 94.4 [93.4, 95.5] |
| | | indifferent | 1.37 [0.33, 2.45] |
| | | acceptable | 4.23 [3.19, 5.32] |
| question 5: direct replication | 1 (1) | move on | 7.16 [5.40, 8.98] |
| | | indifferent | 9.63 [7.87, 11.4] |
| | | replicate | 83.2 [81.5, 85.0] |
| question 6: open methods | 2 (1) | Rs should not do this | 9.56 [7.68, 11.6] |
| | | indifferent | 10.7 [8.85, 12.7] |
| | | Rs should do this | 79.7 [77.8, 81.7] |
| question 7: open access publication | 1 (2) | paywall | 3.12 [0.91, 5.41] |
| | | indifferent | 24.2 [22.0, 26.5] |
| | | free | 72.7 [70.5, 75.0] |
| question 8: data sharing | 1 (1) | Rs should not do this | 11.5 [9.37, 13.6] |
| | | indifferent | 12.6 [10.5, 14.7] |
| | | Rs should do this | 76.0 [73.9, 78.1] |

We settled on the following strict exclusion criteria. First, the first author went through each participant's three open-ended questions (see materials for question wording). The first author used her judgement to mark participants as 'suspicious' or not. Importantly, the first author only had access to these three answers and none of the participants' other responses. Examples of 'suspicious' answers (beyond the preregistered criteria) include those using ungrammatical English, answered in all capital letters, nonsensical comments (e.g. 'GOOD'), or answers that were copy-pasted from Internet search results. This criterion excluded 181 participants (9.7%). Second, we calculated the standard deviation (SD) for each person's answers to the 8 main questions (before reverse-scoring). A person who gave the same answer to each question would have a SD of 0, while the most extreme response possible would have a SD of 2.14. The median SD for observations in our dataset was 1.51. We chose a cut-off of 0.8 and excluded participants with smaller SDs, which excluded 247 participants (13.2%). Finally, we also excluded participants who took less than 2 min to complete the whole study, our subjective judgement of the minimum amount of time it would take someone to skim and click through the survey while still possibly providing meaningful answers. The time criterion mostly applied to MTurk participants, as we did not have the time participants spent on just the meta-study portion of their study for subject pool participants (and for one MTurk study). This criterion excluded 25 participants (1.3%).

When combined, all three criteria excluded a total of 336 participants, or 17.9% of the total sample after our preregistered exclusions, with 12.7% being from MTurk and 5.2% being from university

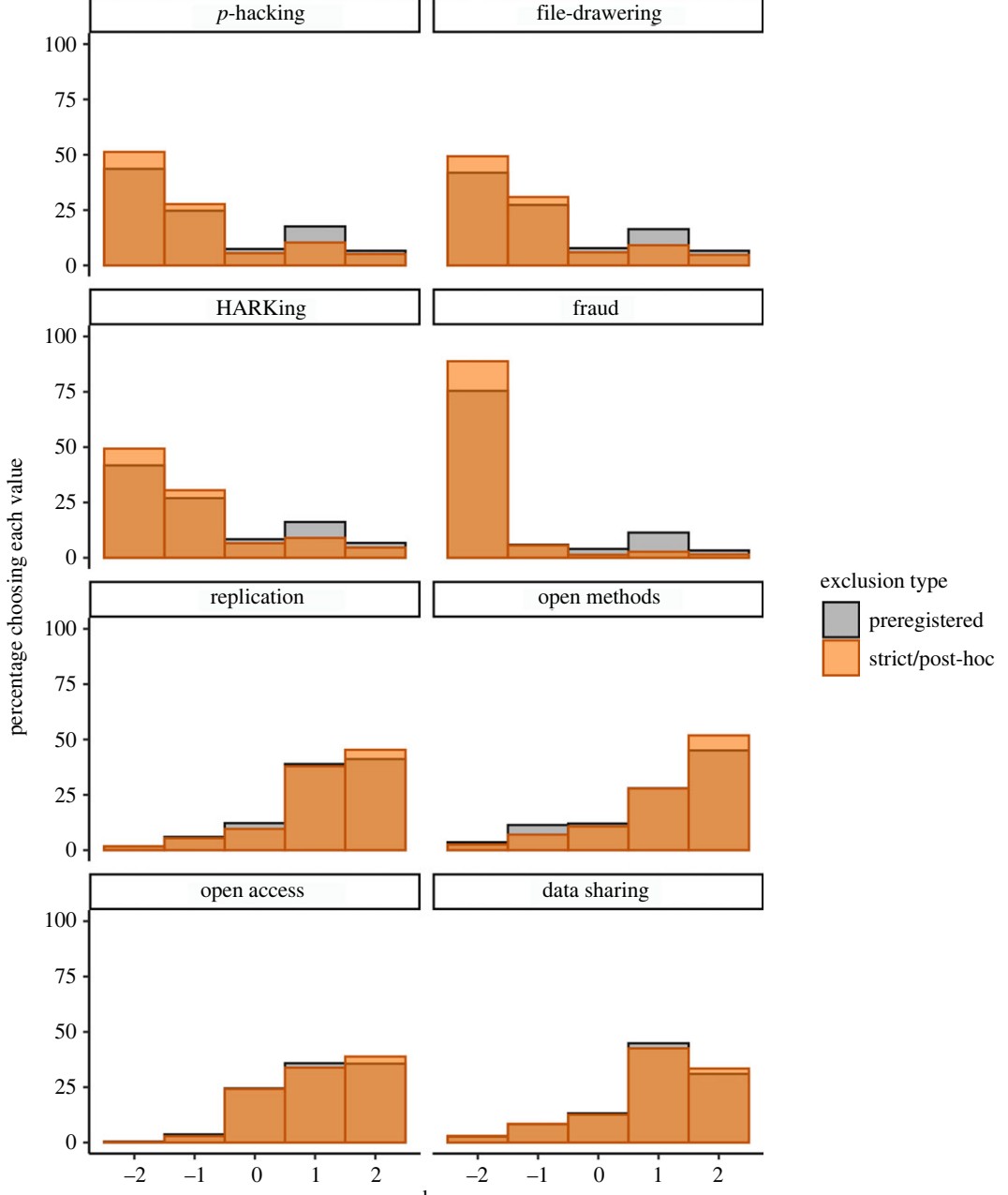

**Figure 4.** Distribution of participants' answers for each question with non-preregistered, strict exclusions (orange), overlayed on the same distribution with preregistered exclusions only (gray), presented in figure 1. For the top four panels, negative numbers indicate that participants found the practice unacceptable while positive numbers indicate they found the practice acceptable. For the bottom four panels, higher numbers indicate more support for the practice. Table 9 for additional results and sample sizes.

subject pools. The resulting sample sizes for each base study after applying these strict exclusion criteria can be found in table 2, and the proportion of participants reporting neutral or positive opinions of fraud before and after exclusions can be seen in table 8.

The results for our main research question, exploring the distribution of responses for each of the eight questions after these strict exclusions, can be found in table 9 and figure 4. As these results show, the distributions look similar to those found with the less strict, preregistered exclusion criteria, but they tend to be more extreme (greater consensus). Interestingly, even after applying these strict exclusion criteria (which excluded mostly MTurk participants), the responses from the subject pool participants were still more extreme than those of MTurk participants.

## 2.3. Discussion

Do people who participate in research have preferences about what scientists do with the data they have provided, and if so, what are those preferences? We attempted to provide an answer to these questions by directly asking participants. Specifically, people who had just participated in a variety of minimal-risk psychology studies self-reported their views about how researchers should treat their data in relation to eight research practices.

Our results show that an overwhelming majority of psychology research participants in these types of studies think the QRPs presented here are unacceptable (though, surprisingly, participants did not have much more extreme views about fraud than about QRPs). Additionally, they were very supportive of practices to increase transparency and replicability, such as conducting direct replications of studies, openly sharing methods (e.g. materials, code, etc.) and data, and publishing in an open access format. For most questions, 5–30% of participants had a different view from the majority. Although an 'indifferent' option was offered for every question (and labelled as such), not many people were indifferent, with values ranging from 1–15% for all questions but one; the open access versus paywalled publishing question was an exception, with about a quarter of participants reporting being indifferent. The similarity in response distributions for different versions of the same question indicates that, although responses can be pushed around by changes in wording or differences in framing, the overall pattern of results seems robust to such variations.

These results, other than participants' views about fraud, are consistent with Pickett & Roche [21], who found that 71% of MTurk participants surveyed report that selective reporting of research findings is morally unacceptable. Indeed, Pickett and Roche found that most participants reported that researchers should be punished (fired and/or banned from receiving funding) for engaging in selective reporting. Given the consistent consensus about QRPs in our study and in Pickett and Roche's study, we first discuss what our results would mean if taken at face value and assuming they are accurate estimates of the views of participants in minimal-risk psychology studies. Then, we discuss reasons why our results may be inaccurate or why such conclusions may be premature.

What should psychologists running minimal-risk research studies do with these findings? First, researchers may want to listen to participants' preferences more. Despite being provided with an opportunity to report being indifferent to what researchers did with their data, participants used this option relatively rarely, suggesting that most participants have opinions about what is acceptable to do with their data. These opinions may reflect not just what they wish would be done with their data, but also how they expect researchers to act. Going directly against participants' expectations might result in less cooperation or in unwillingness to provide high quality data.

At the extreme, it could become an ethical issue; if we continue to engage in practices that we know participants consider unacceptable—and therefore likely expect us not to engage in—we cannot say that participants are providing informed consent to participate in research. Clearly, participants should not be the only ones deciding what research practices are acceptable—highly trained researchers have more information and knowledge to make these decisions. However, if we decide to continue to engage in practices that most research participants consider unacceptable, we should make that explicit in the consent process. For example, in the same way that we warn participants that their anonymized data may be shared with other researchers, we should also let them know that their data may not be shared or published at all, if we continue to selectively report studies or results.

What would it mean if we took the results of our preregistered analyses regarding fraud—namely, that 19% of participants have neutral or positive attitudes toward fraud—at face value? First, this would be very inconsistent with Pickett & Roche's [21] findings from their Study 1, which was also conducted on MTurk and found that 96% of participants expressed the view that fraud is morally unacceptable (even though the word 'fraud' was not used in their questions). Indeed, in their study, 96% of participants also believed that researchers who commit fraud should be fired, and 66% believed fraud should be a crime (in a later study with a representative sample, this view was even more prevalent). Thus, if we are to believe the results of our own preregistered analyses regarding fraud, this would suggest that there are important moderators of participants' views about scientific fraud. There are a number of plausible differences between ours and Pickett and Roche's study that could suggest moderator hypotheses. For example, our participants were asked about a scenario where researchers committed fraud on the data that the same participants had just provided in the base study, whereas participants in Pickett and Roche's study were asked about the abstract idea of fraudulent practices, and fraudulent practices in two hypothetical scenarios. Perhaps participants are less bothered by potential fraud when they have participated in the study themselves and can judge how (in)consequential fraudulent practices would be. However, as we explain below, we

should also seriously consider the possibility that the results of our preregistered analyses regarding fraud are inaccurate and should not be taken at face value, particularly given the inconsistencies with Pickett and Roche's results.

## 2.4. Limitations

There are several reasons to be cautious in interpreting our results. One important limitation of this study is the potential for data quality issues, most obvious in the non-trivial proportion of people expressing positive or neutral views about scientific fraud. Notably, this proportion is much higher for MTurk than subject pool participants when using only our preregistered exclusion criteria (32.8% versus 9.3%; table 8). While the proportion in the subject pool data is consistent with what we saw in our pilot data (around 2 to 10%), the results in the MTurk population are quite alarming, and at odds with another recent MTurk study [21]. We believe this may indicate data quality problems that need to be taken into account when interpreting our results. One implication of low data quality is that our results may be inaccurate. If non-serious responders were responding randomly, or frequently selecting the midpoint, this would add noise to our results and suggest that participants' true attitudes are even more extreme than our results reflect. However, we cannot rule out the possibility that non-serious responders responded in ways that exaggerated the consensus or extremity in our sample's responses.

We attempted to use non-preregistered strict exclusions to reduce the influence of non-serious responders, and although this serves as a robustness check, these exploratory estimates have their own limitations. First, our decisions were data-driven and we explored several ways of excluding participants, many of which we do not report here. This was a subjective process and one indicator we used to decide when we had reached a good set of exclusion criteria was the lower rate of participants reporting that fraud was acceptable. There are two important consequences of this process. First, the fraud estimates from these exploratory analyses are uninformative as our decisions about exclusions were driven in part by our preconceptions about what these levels should be. Second, the results with strict exclusions for all other questions probably underestimate the proportion of truly indifferent participants, because someone who was indifferent to most things would likely have been excluded when we applied our strict exclusion criteria.

Another limitation relates to how we worded the questions. Although we spent a considerable amount of time writing and rewriting them to be as clear and unbiased as possible, our own opinions about these research practices are certainly reflected in the final wording, and likely had some influence on how participants responded to the questions. In fact, we see evidence that participants' opinions can be moved around by question wording: participants reported more extreme opinions when they read the version of the $p$-hacking or filedrawering questions that implied a motive for not reporting every result or study than when they read a neutral version of the same question. Similarly, for the data sharing question, participants reported less extreme views about data sharing after reading about the pros and cons of data sharing, and some of the reasons researchers may or may not want to share their data, compared to participants who were presented with the same question but without the explicit pros and cons. However, those same results provide some constraint around the plausible effects of question wording. Although changing the question wording affected how extreme the responses were, the proportion of participants who approve versus disapprove of each practice remained relatively stable (compare tables 4 and 9). It would be difficult to imagine a way in which we could ask the same question that would sway participants enough to change the general consensus we see across participants for most of the questions.

Another limitation of our study is that it is not clear what importance participants place on the views they have expressed here. Do participants have pre-existing views about the acceptability of these practices, or did they formulate these views on the spot in response to our questions? Either way, how important is it to participants that researchers behave in accordance with participants' expectations and views of what is acceptable? Here again, the findings of Pickett & Roche [21] are relevant, as participants in Study 1 reported their views on several potential punishments for researchers who engage in selective reporting. Their findings suggest that most participants believe selective reporting (similar to the $p$-hacking and filedrawering questions in our study) is quite serious, and should be punished. 63% of MTurk participants in Pickett and Roche's Study 1 reported that researchers who engage in selective reporting should be fired. However, participants in that study were given two scenarios as examples, only one of which was a minimal-risk psychology study (the other was a study about blood pressure medication). We suspect that participants view QRPs in the context of minimal-risk psychology research as less serious than in the context of medical research. Thus, it is an open question how serious participants believe QRPs to be in the context of minimal-risk psychology research.

In our opinion, the most important follow-up questions regarding the importance that participants place on these practices are: Would participants still choose to participate if they were aware of the (questionable and open) practices that researchers routinely engage in with their data? Would knowing how researchers are planning to use their data affect the quality of participants' responses? Would it affect their views of the credibility and importance of minimal-risk psychology research, and their support for public funding of such research? Our findings suggest that these questions are urgent and worth studying, but we do not yet know the answers.

Finally, another important limitation of our study is that there are serious constraints on the generality of our findings. We believe our findings can be generalized beyond the current sample to some extent. Specifically, although we only had eight base studies, we believe these base studies are fairly representative of other minimal-risk, online, cross-sectional psychology studies. Therefore, we believe the results of this study accurately represent the reported opinions of the typical research participant in minimal-risk, online, cross-sectional psychology studies and may apply to similarly simple online studies in other social and behavioural sciences. However, these results cannot be generalized further than that.

Specifically, we do not believe that our results would generalize to participants' views of how their data should be treated in studies with more intensive designs (e.g. longitudinal designs, field studies), higher-risk studies (e.g. studies collecting personal health information, recordings of private behaviour) or studies on more obviously consequential topics (e.g. clinical trials). Elements of these studies may affect how much participants are invested in the research process, and could produce very different results. We can imagine these features shifting attitudes in various directions. Participants may feel even more strongly that their data should be handled with as little bias (less tolerance for questionable practices) and as much transparency (stronger endorsement of open practices) as possible when the study asked more of them or when the topic is perceived as more important. On the other hand, participants may be less enthusiastic about data sharing when the data they provided are more personal, and they may be more tolerant of publishing without replication when the topic is considered urgent and important. However, as mentioned earlier, studies in higher-risk contexts suggest that, despite some concerns about privacy and confidentiality, a majority of participants support sharing of de-identified data and are willing to share their own data, with some restrictions [18–20].

It is also unclear whether these results would generalize to other types of participants. First, differences between the general population and the typical research participant in opt-in samples have been well documented [23]. Second, our participants were living (as far as we know) exclusively in the United States. It is possible that other countries or cultures may differ in their opinions of research practices, even for minimal-risk studies.

# 3. Conclusion

Our findings are more ambiguous than we would have hoped, due to data quality concerns raised by the surprising distribution of responses to our question about fraud. Nevertheless, we believe the findings paint a fairly clear picture of participants' views about questionable and open research practices: most participants in online, minimal-risk, simple, cross-sectional psychology studies would not approve of their data being used to $p$-hack, filedrawer, or HARK, and would prefer that the research findings be subjected to replication attempts and shared transparently and openly. These findings are in line with those in the literature.

Our findings add to a growing body of evidence suggesting that researchers may routinely violate participants' expectations about how their data will be used, assuming that participants do not expect researchers to act in ways that they (the participants) find unacceptable. If we want to honour participants' expectations, we have several choices. We can: (1) align our practices with participants' expectations, (2) change participants' expectations by educating participants and the public about why practices that they initially disapprove of may be necessary or beneficial for science, (3) do more research to understand the reasons and principles behind participants' expectations and look for ways to simultaneously honour participants' and researchers' values, or (4) transparently inform participants about how we will handle their data and accept that some may drop out or provide low quality data. While further research is necessary to understand the breadth of this problem, and what the consequences might be, in the meantime we should, at a minimum, communicate our plans more transparently to participants, so that they can make a more informed decision about participating in our research.

Ethics. Permission to perform this study was granted by the University of California Institutional Review Board (IRB), IRB IDs1423371-2, 1787646-1 and 1744965-1. Permission to perform this study (and accompanying base studies) at

other universities was granted by the Sacramento State Institutional Review Board (IRB), IRB ID Cayuse-20-21-240; the Princeton University Institutional Review Board (IRB), IRB ID 13508-04; and the University of Pennsylvania Institutional Review Board (IRB), IRB IDs 844186 and 844870.

Data accessibility. All data for the pilots is available at the OSF page for this project (https://osf.io/bgpyc/). Data for the main study can be found at https://osf.io/zr29g/. The registration for the stage 1 manuscript for this report can be found at https://osf.io/8anxu, and the corresponding stage 1 manuscript can be directly accessed at https://osf.io/re5uf/.

Authors' contributions. J.G.B.: Conceptualization, data curation, formal analysis, investigation, methodology, project administration, resources, software, visualization, writing—original draft, writing—review and editing; M.R.: Methodology, supervision, writing—review and editing; S.V.: Conceptualization, investigation, methodology, supervision, writing—original draft, writing—review and editing.

All authors gave final approval for publication and agreed to be held accountable for the work performed therein.

Conflict of interest declaration. We declare we have no competing interests.

Funding. Funding for this study is provided by university research funds to Simine Vazire and Mijke Rhemtulla.

Acknowledgements. We thank Hale Forster, Oliver Clark, Jessie Sun, Gerit Pfuhl, Eric Y. Mah, D. Stephen Lindsay, Yeji Park, Kate M. Turetsky, Kevin Reinert, Samuel H. Borislow, Jasmin Fernandez Castillo, Greg M. Kim-Ju, Jeremy R. Becker, Kate Hussey, and Fabiana Alceste for agreeing to provide us with base studies. We also thank Hale Forster and Oliver Clark for running data collection for Pilots A and C; Jessie Sun, Yeji Park, Gerit Pfuhl, Jasmin Fernandez Castillo, Samuel H. Borislow, and Jack Friedrich for running data collection for parts of the main study; and Beth Clarke for comments on the manuscript.

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
