## [Peer Review File · Royal Society Open Science]

Review History

RSOS-200048.R0 (Original submission)

Review form: Reviewer 1

Do you have any ethical concerns with this paper?

No

Recommendation?

Accept with minor revision

Comments to the Author(s)

1.) This Stage 1 manuscript seeks to examine research participants' perspectives of research practices. I read the article with great interest because I agree with the authors assessment that we should at least poll the stakeholders involved in our research enterprise to get their opinions. In a very real sense, I view them as partial owners of the data that they produce (whether it be via

their own efforts, their tax payer dollars, or tuition dollars). The same goes with MTurk participants. In this review system, we're asked to comment on a variety of different domains, which I will do below. I was positively disposed to the manuscript and I think the results would be informative to the field (so an in principle acceptance would be a reasonable decision from my perspective).

The analysis was also really reasonable because it's a descriptive account of participants' evaluations. From my perspective, there might not be that much ambiguity about how to analyze the data. Plus, what ambiguity that did exist (particularly related to question wording and selection) was discussed at length and even partially tested in these pilot studies. The question holds much scientific validity because it speaks to the reflections that stakeholders have on the research process and is of natural interest to modern researchers.

2.) The logic and rationale of the project were laid out very well and carefully. The inclusion of pilot studies was particularly beneficial because it settled some of the ambiguity in how participants might respond to questions depending on how they were worded.

3.) The soundness and feasibility were major strengths. The sample size for the proposed registered study was healthy and well-justified. Again, the study is sound and reasonable partially because the pilot studies approximate what will be done on a larger scale.

4.) The clarity/degree of detail is sufficient to replicate the study (and likely the analysis pipeline). I say this because the materials were presented very clearly in the manuscript and online supplementary materials. Regarding some things to improve for the manuscript, the authors could clarify how any comparisons between MTurkers and undergrads are made. Even if it's just "eye-balling" and reporting it separately by sample and through a formal comparison (e.g., t-tests or chi-square tests). Also, in their analysis plan (I think this went without saying), I got the sense that the positive or negative items would be reverse scored so that t-tests would be interpretable (e.g., to compare if the responses differed based on the question). If this is the case, it might be good to say that. If not, just clarify that no recoding of the variables will be necessary. Reporting of effect sizes is also preferred (again, likely goes without saying). The proposed study is really good, so these minor suggestions are the best I can come up with!

5.) The likelihood of undisclosed flexibility is low. I say this because of the study's simplicity – the questions will be tacked onto some existing studies and the analysis/reporting is pretty straightforward.

6.) We're asked if the authors considered a sufficient number of outcome-neutral conditions such that the results indeed test the hypotheses. The design here is a bit different (and there aren't hypotheses; although, honestly, it wouldn't be hard to make some based on the pilot studies...). Nevertheless, the authors managed to do even this given the design. They selectively varied the way various questions were worded (e.g., positive/negative, motives/neutral) so that the results will be informative. A superficial read of this final paper might lay the challenge that the items were worded in suggestive ways or that the authors were "in the bag" for convincing students/Turkers that these are good things. Their neutral approach and varying of the stimuli help defend against the criticism, too.

Review form: Reviewer 2 (Loukia Tzavella)

Do you have any ethical concerns with this paper?

No

Recommendation?

Accept with minor revision

Comments to the Author(s)

Please see attached pdf (Appendix A).

Review form: Reviewer 3 (Robert Thibault)

Do you have any ethical concerns with this paper?

No

Recommendation?

Accept with minor revision

Comments to the Author(s)

The proposed experiment is very well-documented, has been refined through pilot studies, highlights several open practices the authors have taken, and would make a clear, reproducible, and important contribution to the scientific literature. It is among the most well-documented and clear protocols I have read. I recommend that it receives in-principle acceptance after minor revisions.

I have one substantial comment regarding the research questions and one substantial comment regarding the wording of a few questions. The remainder of my comments are minor and scattered throughout the attached pdf in comment boxes. I leave it to the authors' judgement of whether or not to respond to some of the minor comments. Below are the specific questions I was asked to explicitly respond to.

Please comment explicitly on each of the following points in your comments to the authors:

The scientific validity of the research question(s)

The two itemized research questions come across as more general than the questions the authors appear to be asking based on a reading of the proposal in its entirety. The itemized research questions ask: What are participants' opinions about (1) QRPs and fraud, and (2) open science practices. The specific questions that participants will be asked, however, center on whether participants support or oppose QRPs, fraud, and open practices. Specifically, related to their own data and the data from studies they've participated in. If the RQs were mainly interested in 'opinions', then open text boxes would be useful. However, it appears to me that this isn't the main purpose of the study.

Refining the RQs to be more specific could help refine some of the questions or help justify the current wording of the questions. For example, Question 6 asks if participants feel that researchers should or should not share their materials and procedures. This question doesn't ask to weigh the costs of sharing (which some might argue is a lot of time). It's unclear whether the RQs aim to discover what participants want to be done with their data (ideals) or what they think is reasonable to do with their data (practicality), or both. More specificity could clear this up. There are a few comment boxes in the pdf touching on these points.

I think the overarching research question is valid and important.

The logic, rationale, and plausibility of the proposed hypotheses

The authors do not propose hypotheses. They do propose two research questions (see previous answer).

The soundness and feasibility of the methodology and analysis pipeline (including statistical power analysis where applicable)

The methodology is sound and feasible. The authors refined it over multiple pilot studies. The main analyses are sound and feasible and supported by a reasonable precision analysis.

I recommend the authors include more information about the t-tests they will perform, including their inferential criteria, power, smallest effect size of interest, multiple comparisons, and conclusions they expect to be able to draw based on potential outcomes. Are they testing for differences, equivalence, or both?

Whether the clarity and degree of methodological detail would be sufficient to replicate exactly the proposed experimental procedures and analysis pipeline

The authors provide all the materials and data from their pilot studies on the osf, as well as appropriate links throughout their proposal. I could not identify where the questionnaire used for pilot C is located. Presumably this file could be shared at this point in time. I also did not locate the questionnaire that will be used for the RR. Although the authors lucidly explain how they changed the questions, it would be useful to see the exact questions as they will be presented to participants.

Whether the authors provide a sufficiently clear and detailed description of the methods to prevent undisclosed flexibility in the experimental procedures or analysis pipeline

The manuscript is written with good specificity as to prevent undisclosed flexibility. They clearly identify that some collected data will only be used in exploratory analyses. In the pdf comments, I identify a few minor tweaks that could be made to increase specificity (reduce the potential for flexibility).

Whether the authors have considered sufficient outcome-neutral conditions (e.g. positive controls) for ensuring that the results obtained are able to test the stated hypotheses

The authors do not propose hypotheses. The questionnaire includes a question on fraud which the authors propose could be used as a positive control. I recommend the authors provide more detail regarding this point. They also use an attention/comprehension check, which I recommend the authors explain in more detail. The methods are sufficient to inform the overarching research question in its current form. However, as stated above, I recommend the authors refine the research question to more closely match what they (appear) to be proposing to answer.

Thank you for the opportunity to review this manuscript. I look forward to reading the results when they come in.

Robert Thibault

Review form: Reviewer 4 (Farid Anvari)

Do you have any ethical concerns with this paper?

No

Recommendation?

Accept with minor revision

Comments to the Author(s)

I like the proposed project and think it tackles an interesting and important question. I think the authors make a strong case for why we should know what participants think about our research practices and how this might fall within the requirements of informed consent, broadly defined. And I also believe that the justification for focusing on minimal risk studies with convenience samples is quite sound, with constraints on generalizability. (Surprising results from pilot study A. So many MTurk workers want their data shared for verification and other analyses. Amazing!) My suggestions for improvement/clarification follow:

Methods & Analyses

Questions 3, 5, 6, 7, and 8, which have had at least minor changes made to their wordings weren't presented for me to adequately examine them. My comments below are based on assumptions I made (and which I make clear) regarding the wording of the questions.

Questions 5, Page 21, lines 26-28: I assume the question is "researchers should attempt to replicate a finding before publishing it or trust that the finding is solid and move on to a new project". I think it has the same problem as in Pilot B (in which some participants gave what seemed to be contradictory answers to the two versions of the question) but now the problem is presented within the question. I think the question can distort responses since both positions can be held at the same time. E.g., I don't think every finding needs to be replicated before being published but I also don't think researchers should trust it is solid and move on. You can publish a finding without trusting that it is solid. And sometimes this can make science move forward, as long as independent replications (both negative and positive) are published. I agree that giving participants context (e.g., that there is a tradeoff) in this case is necessary. My main concern is that in trying to give context within the question the question has become very difficult to answer precisely and thus very difficult to interpret its responses, particularly because the tradeoff isn't clear (at least to me). The question is a kind of double-barreled question (the second part not only asks whether authors should publish without replicating but also about whether they should move on after trusting it is solid). I think the authors should make clearer what the tradeoff is, be clear that it is a tradeoff rather than two positions that can be both held simultaneously, and then reword the question. Perhaps even providing the context in a sentence before asking the question. I'm really not sure about this one and it very much depends on what the authors want an answer to. What precisely is the tradeoff that the authors are getting at and is it actually a tradeoff? Maybe the authors want to know, "should researchers make sure their finding is solid before publishing it, or is it ok to publish something you aren't sure is solid yet"? Or is the tradeoff they want to get at better described with a statement such as, "part of establishing whether a finding is solid is by replicating it in another study to see if you get the same results", and then asking "should researchers replicate their finding in a new study before publishing or is it ok to publish the finding of one study without replicating"?

Question 7, Pages 21-22: I assume the wording of the question is given in the first sentence of the paragraph (i.e., "...where the article reporting the results of the base study should be published: an open access journal vs a paywalled journal"). If the intention is to give a clearer picture of what happens to the funds and who will be accessing the article, then perhaps more explicit

wording is required. E.g., “the article reporting the results should be published in an open access journal that gives free access to anyone who wants to read the article vs a for-profit paywalled journal that charges a fee” (note that this is only a suggestion). The concern with the question as presented in Pilot B, that participants might have thought the authors are compensated and that it isn’t the publisher that makes the money, isn’t quite addressed in this case (i.e., where do the funds at the paywalled journal go? Do they pay the author or does it go to funding more research?). There’s still a lot left to assumptions of the reader.

Analyses: Given that some of the data are likely to be highly skewed (e.g., Pilot B, QRPs), the means and standard deviations seem less informative to me (although this is mostly offset by presenting the results in more detail, as the authors did for Pilot B and they propose to do for the main study, by presenting percentage of participants finding the practices (un)acceptable). I recommend also providing the median. For the main study, the authors propose comparing the two versions of each question (where relevant) by using t-tests but if the data are highly skewed then perhaps a non-parametric test is better. I also wondered why they aren’t proposing a chi-square test comparing percentage of participants finding the practices (un)acceptable, between the two methods of asking each question.

Page 21, lines 18-21, questions 5 and 7. If the worry is that participants might think one end of the numbering scale might be less desirable vs more desirable, then perhaps presenting these labelled options in a nonlinear way might be best. E.g., rather than from left to right (which participants might inadvertently perceive as going from less desirable to more desirable (given that the questions preceding it are presumably going to be presented in this way)), the options for questions 5 and 7 might be presented vertically or via a dropdown box, or in whatever way the authors prefer. This is just a suggestion.

Page 22, question 8: I can’t see the wording of the question nor the reasons presented to participants.

Page 22: data exclusion attention/comprehension check: what is the question? Also, (I may have missed it but) the authors should be clear if excluding participants who took part in the pilot studies a and b from taking part in main study.

Participants: Is there a reason the authors are not including any Prolific participants? If the worry is that MTurk workers are different from prolific workers, then generalizability beyond MTurk will be an issue (and should be stated). In the same way that the authors propose to collect data from student participants at different universities, I would suggest that online participants from different platforms should not be a problem. Prolific allows prespecifying that participants be from the US. Of course, I’m not suggesting that there needs to be equal number of MTurk and Prolific participants, but merely that the authors may want to consider also including studies being run on Prolific (just a suggestion).

Issues of clarity/argumentation in the introduction:

Page 4, lines 29-31, “going against participants’ wishes could undermine public trust in science even among non-participants, if they feel we are mistreating participants”: I don’t follow the logic of the argument that leads to the final sentence. The paragraph is about participants potentially being driven away from participating due to their preferences being ignored. So how does it follow that the public trust in science would be diminished if they perceived that we mistreat participants? Is ignoring their preferences for data sharing, or using their data to p-hack, going to be perceived as mistreatment? I think the final sentence of the paragraph requires more reasoning but it would be a digression from the point of the paper without adding much added value.

Page 5 lines 20-25: The arguments leading up to this paragraph have been (1) researchers are not necessarily incentivized to produce good replicable work and so this may have created norms and practices that are not optimal for science, and getting participants' views may be one way for checks and balances against this, and (2) participants should be informed about current research practices, potentially as part of informed consent, and thus their views are important because then we can see what their perceptions are. When the paragraph speaks of blind spots, I found it confusing. On my second reading I realized the blind spots are with regards to point (1). So maybe something to make this clearer in the paragraph will benefit readers.

Minor points

Page 5 line 19, the paragraph starts, "to solve this inconsistency". This statement is ambiguous. Is it referring to "mismatch between their expectations and the reality of research"? If so, then the ambiguity can be cleared up with something like, "If we do find that there is an inconsistency between participants' expectations and research practices, scientists have several options".

Page 4, lines 26-29: not only might participants stop from participating, but they may continue to participate but lose the motivation to provide high quality data.

Page 17 lines 32-36: although 89% is quite low for finding scientific fraud unacceptable, and may be due to nonserious reporting, it is not too different from some of the results from Picket and Roche who found 91% reported fraud as unacceptable (Study 2), though their MTurk sample had 96% finding it unacceptable.

Interesting note (Page 18, lines 31-32): Picket and Roche's findings on selective reporting are very consistent with Pilot B results on all QRPs. The finding from Pilot B suggests that over 75% found all QRPs unacceptable; 71% of Picket and Roche's large MTurk sample considered selective reporting unacceptable, though they used the binary yes/no response options.

Study selection, page 20, lines 31-33: what is the time-frame of the study? Perhaps this should be prespecified.

I always sign my reviews,
Farid Anvari

Review form: Reviewer 5 (Hannah Fraser)

Do you have any ethical concerns with this paper?

No

Recommendation?

Accept with minor revision

Comments to the Author(s)

Full comments are included in the attached document (see Appendix B) but we respond briefly to the specific RSOS questions here

The scientific validity of the research question(s):

This is Stage 1 Registered Report for a study proposing to characterise English-speaking, psychology research participants' views on 1) select questionable research practices (taken from John et al 2012) and 2) open science practices (data sharing, replication, open methods and open access) online. We feel that the proposed study would be a timely and meaningful contribution to the literature

The logic, rationale, and plausibility of the proposed hypotheses:

The study is primarily descriptive and doesn't offer any specific hypotheses. However, we ask that the authors add hypotheses regarding the t-tests included in their analysis plan.

The soundness and feasibility of the methodology and analysis pipeline (including statistical power analysis where applicable)

The overall plan appears sound however we would like to see more description about how the t-tests will be conducted. The authors should provide a power analysis for these t-tests in addition to the power analysis currently included. We are also concerned that the authors' choice of confidence intervals may not be suitable for multinomial proportions and have suggested an alternative R package that might work better.

Whether the clarity and degree of methodological detail would be sufficient to replicate exactly the proposed experimental procedures and analysis pipeline

At this point the authors appear to have provided materials for their pilot studies but not for the Registered Report study. These should be made available and more information should be provided about the statistical analyses.

Whether the authors provide a sufficiently clear and detailed description of the methods to prevent undisclosed flexibility in the experimental procedures or analysis pipeline

If the authors provide the requested information about the materials and t-test and confidence interval calculations there will be no room for flexibility in experimental procedures or analyses

Whether the authors have considered sufficient outcome-neutral conditions (e.g. positive controls) for ensuring that the results obtained are able to test the stated hypotheses:

NA

Review Conducted by Hannah Fraser and Daniel Hamilton

Decision letter (RSOS-200048.R0)

12-Feb-2020

Dear Ms Bottesini

On behalf of the Editors, I am pleased to inform you that your Manuscript RSOS-200048 entitled "What Do Participants Think of Our Research Practices? An Examination of Behavioral Psychology Participants' Preferences" deemed suitable for in-principle acceptance in Royal Society Open Science subject to revision in accordance with the referee and editor suggestions. Please find their comments at the end of this email.

The reviewers and handling editors have recommended publication, but also suggest some minor revisions to your manuscript. Therefore, I invite you to respond to the comments and revise your manuscript.

Your due date for this revised manuscript will be 11 March 2020. Please do let us know as soon as possible if you have any queries regarding this.

To revise your manuscript, log into <https://mc.manuscriptcentral.com/rsos> and enter your Author Centre, where you will find your manuscript title listed under "Manuscripts with

Decisions". Under "Actions," click on "Create a Revision." You will be unable to make your revisions on the originally submitted version of the manuscript. Instead, revise your manuscript and upload a new version through your Author Centre.

Full author guidelines can be found here <https://royalsocietypublishing.org/rsos/registered-reports#ReviewerGuideRegRep>.

Kind regards
Professor Chris Chambers
Royal Society Open Science
openscience@royalsociety.org

on behalf of Professor Chris Chris Chambers
(Subject Editor, Royal Society Open Science)
openscience@royalsociety.org

Associate Editor Comments to Author (Professor Chris Chambers):

Associate Editor: 1

Comments to the Author:

Five reviewers have now assessed the Stage 1 manuscript. The reviews are very positive overall and also very detailed and constructive. Key issues to address in revision include clarification of the survey questions (reviewers 3 and 4), suitability of the attention check (reviewers 2 and 3), more details about the planned analyses including t-tests (reviewers 2, 3 and 5), clarity and structure of the introduction (reviewers 4 and 5), as well as a range other concerns including sample validity (reviewers 4), suggestions to minimise jargon (an excellent point given the broad remit of RSOS; reviewer 5), clarification of various terms and definitions (reviewer 2) and well as clarification of methodological details (raised by all reviewers). Concerning the recommendation of reviewer 3 to shift details of the pilot studies to the supplementary information, I will leave this to the discretion of the authors since either approach can work for a RR.

Reviewer comments to Author:

Reviewer: 1

Comments to the Author(s)

1.) This Stage 1 manuscript seeks to examine research participants' perspectives of research practices. I read the article with great interest because I agree with the authors assessment that we should at least poll the stakeholders involved in our research enterprise to get their opinions. In a very real sense, I view them as partial owners of the data that they produce (whether it be via their own efforts, their tax payer dollars, or tuition dollars). The same goes with MTurk participants. In this review system, we're asked to comment on a variety of different domains, which I will do below. I was positively disposed to the manuscript and I think the results would be informative to the field (so an in principle acceptance would be a reasonable decision from my perspective).

The analysis was also really reasonable because it's a descriptive account of participants' evaluations. From my perspective, there might not be that much ambiguity about how to analyze the data. Plus, what ambiguity that did exist (particularly related to question wording and selection) was discussed at length and even partially tested in these pilot studies. The question holds much scientific validity because it speaks to the reflections that stakeholders have on the research process and is of natural interest to modern researchers.

2.) The logic and rationale of the project were laid out very well and carefully. The inclusion of pilot studies was particularly beneficial because it settled some of the ambiguity in how participants might respond to questions depending on how they were worded.

3.) The soundness and feasibility were major strengths. The sample size for the proposed registered study was healthy and well-justified. Again, the study is sound and reasonable partially because the pilot studies approximate what will be done on a larger scale.

4.) The clarity/degree of detail is sufficient to replicate the study (and likely the analysis pipeline). I say this because the materials were presented very clearly in the manuscript and online supplementary materials. Regarding some things to improve for the manuscript, the authors could clarify how any comparisons between MTurkers and undergrads are made. Even if it's just "eye-balling" and reporting it separately by sample and through a formal comparison (e.g., t-tests or chi-square tests). Also, in their analysis plan (I think this went without saying), I got the sense that the positive or negative items would be reverse scored so that t-tests would be interpretable (e.g., to compare if the responses differed based on the question). If this is the case, it might be good to say that. If not, just clarify that no recoding of the variables will be necessary. Reporting of effect sizes is also preferred (again, likely goes without saying). The proposed study is really good, so these minor suggestions are the best I can come up with!

5.) The likelihood of undisclosed flexibility is low. I say this because of the study's simplicity – the questions will be tacked onto some existing studies and the analysis/reporting is pretty straightforward.

6.) We're asked if the authors considered a sufficient number of outcome-neutral conditions such that the results indeed test the hypotheses. The design here is a bit different (and there aren't hypotheses; although, honestly, it wouldn't be hard to make some based on the pilot studies...). Nevertheless, the authors managed to do even this given the design. They selectively varied the way various questions were worded (e.g., positive/negative, motives/neutral) so that the results will be informative. A superficial read of this final paper might lay the challenge that the items were worded in suggestive ways or that the authors were "in the bag" for convincing students/Turkers that these are good things. Their neutral approach and varying of the stimuli help defend against the criticism, too.

Reviewer: 2

Comments to the Author(s)

Please see attached pdf (Review_RSOS-200048_LT).

Reviewer: 3

Comments to the Author(s)

THE TEXT BELOW IS INCLUDED IN AN ATTACHED FILE WHICH IS FORMATTED TO BE EASIER TO READ.

The proposed experiment is very well-documented, has been refined through pilot studies, highlights several open practices the authors have taken, and would make a clear, reproducible, and important contribution to the scientific literature. It is among the most well-documented and clear protocols I have read. I recommend that it receives in-principle acceptance after minor revisions.

I have one substantial comment regarding the research questions and one substantial comment regarding the wording of a few questions. The remainder of my comments are minor and scattered throughout the attached pdf in comment boxes. I leave it to the authors' judgement of whether or not to respond to some of the minor comments. Below are the specific questions I was asked to explicitly respond to.

Please comment explicitly on each of the following points in your comments to the authors:

The scientific validity of the research question(s)

The two itemized research questions come across as more general than the questions the authors appear to be asking based on a reading of the proposal in its entirety. The itemized research questions ask: What are participants' opinions about (1) QRPs and fraud, and (2) open science practices. The specific questions that participants will be asked, however, center on whether participants support or oppose QRPs, fraud, and open practices. Specifically, related to their own data and the data from studies they've participated in. If the RQs were mainly interested in 'opinions', then open text boxes would be useful. However, it appears to me that this isn't the main purpose of the study.

Refining the RQs to be more specific could help refine some of the questions or help justify the current wording of the questions. For example, Question 6 asks if participants feel that researchers should or should not share their materials and procedures. This question doesn't ask to weigh the costs of sharing (which some might argue is a lot of time). It's unclear whether the RQs aim to discover what participants want to be done with their data (ideals) or what they think is reasonable to do with their data (practicality), or both. More specificity could clear this up. There are a few comment boxes in the pdf touching on these points.

I think the overarching research question is valid and important.

The logic, rationale, and plausibility of the proposed hypotheses

The authors do not propose hypotheses. They do propose two research questions (see previous answer).

The soundness and feasibility of the methodology and analysis pipeline (including statistical power analysis where applicable)

The methodology is sound and feasible. The authors refined it over multiple pilot studies. The main analyses are sound and feasible and supported by a reasonable precision analysis.

I recommend the authors include more information about the t-tests they will perform, including their inferential criteria, power, smallest effect size of interest, multiple comparisons, and conclusions they expect to be able to draw based on potential outcomes. Are they testing for differences, equivalence, or both?

Whether the clarity and degree of methodological detail would be sufficient to replicate exactly the proposed experimental procedures and analysis pipeline

The authors provide all the materials and data from their pilot studies on the osf, as well as appropriate links throughout their proposal. I could not identify where the questionnaire used for pilot C is located. Presumably this file could be shared at this point in time. I also did not locate the questionnaire that will be used for the RR. Although the authors lucidly explain how they changed the questions, it would be useful to see the exact questions as they will be presented to participants.

Whether the authors provide a sufficiently clear and detailed description of the methods to prevent undisclosed flexibility in the experimental procedures or analysis pipeline

The manuscript is written with good specificity as to prevent undisclosed flexibility. They clearly identify that some collected data will only be used in exploratory analyses. In the pdf comments, I identify a few minor tweaks that could be made to increase specificity (reduce the potential for flexibility).

Whether the authors have considered sufficient outcome-neutral conditions (e.g. positive controls) for ensuring that the results obtained are able to test the stated hypotheses

The authors do not propose hypotheses. The questionnaire includes a question on fraud which the authors propose could be used as a positive control. I recommend the authors provide more detail regarding this point. They also use an attention/comprehension check, which I recommend the authors explain in more detail. The methods are sufficient to inform the overarching research question in its current form. However, as stated above, I recommend the authors refine the research question to more closely match what they (appear) to be proposing to answer.

Thank you for the opportunity to review this manuscript. I look forward to reading the results when they come in.

Robert Thibault

Reviewer: 4

Comments to the Author(s)

I like the proposed project and think it tackles an interesting and important question. I think the authors make a strong case for why we should know what participants think about our research practices and how this might fall within the requirements of informed consent, broadly defined. And I also believe that the justification for focusing on minimal risk studies with convenience samples is quite sound, with constraints on generalizability. (Surprising results from pilot study A. So many MTurk workers want their data shared for verification and other analyses. Amazing!) My suggestions for improvement/clarification follow:

Methods & Analyses

Questions 3, 5, 6, 7, and 8, which have had at least minor changes made to their wordings weren't presented for me to adequately examine them. My comments below are based on assumptions I made (and which I make clear) regarding the wording of the questions.

Questions 5, Page 21, lines 26-28: I assume the question is "researchers should attempt to replicate a finding before publishing it or trust that the finding is solid and move on to a new project". I think it has the same problem as in Pilot B (in which some participants gave what seemed to be contradictory answers to the two versions of the question) but now the problem is presented within the question. I think the question can distort responses since both positions can be held at

the same time. E.g., I don't think every finding needs to be replicated before being published but I also don't think researchers should trust it is solid and move on. You can publish a finding without trusting that it is solid. And sometimes this can make science move forward, as long as independent replications (both negative and positive) are published. I agree that giving participants context (e.g., that there is a tradeoff) in this case is necessary. My main concern is that in trying to give context within the question the question has become very difficult to answer precisely and thus very difficult to interpret its responses, particularly because the tradeoff isn't clear (at least to me). The question is a kind of double-barreled question (the second part not only asks whether authors should publish without replicating but also about whether they should move on after trusting it is solid). I think the authors should make clearer what the tradeoff is, be clear that it is a tradeoff rather than two positions that can be both held simultaneously, and then reword the question. Perhaps even providing the context in a sentence before asking the question. I'm really not sure about this one and it very much depends on what the authors want an answer to. What precisely is the tradeoff that the authors are getting at and is it actually a tradeoff? Maybe the authors want to know, "should researchers make sure their finding is solid before publishing it, or is it ok to publish something you aren't sure is solid yet"? Or is the tradeoff they want to get at better described with a statement such as, "part of establishing whether a finding is solid is by replicating it in another study to see if you get the same results", and then asking "should researchers replicate their finding in a new study before publishing or is it ok to publish the finding of one study without replicating"?

Question 7, Pages 21-22: I assume the wording of the question is given in the first sentence of the paragraph (i.e., "...where the article reporting the results of the base study should be published: an open access journal vs a paywalled journal"). If the intention is to give a clearer picture of what happens to the funds and who will be accessing the article, then perhaps more explicit wording is required. E.g., "the article reporting the results should be published in an open access journal that gives free access to anyone who wants to read the article vs a for-profit paywalled journal that charges a fee" (note that this is only a suggestion). The concern with the question as presented in Pilot B, that participants might have thought the authors are compensated and that it isn't the publisher that makes the money, isn't quite addressed in this case (i.e., where do the funds at the paywalled journal go? Do they pay the author or does it go to funding more research?). There's still a lot left to assumptions of the reader.

Analyses: Given that some of the data are likely to be highly skewed (e.g., Pilot B, QRPs), the means and standard deviations seem less informative to me (although this is mostly offset by presenting the results in more detail, as the authors did for Pilot B and they propose to do for the main study, by presenting percentage of participants finding the practices (un)acceptable). I recommend also providing the median. For the main study, the authors propose comparing the two versions of each question (where relevant) by using t-tests but if the data are highly skewed then perhaps a non-parametric test is better. I also wondered why they aren't proposing a chi-square test comparing percentage of participants finding the practices (un)acceptable, between the two methods of asking each question.

Page 21, lines 18-21, questions 5 and 7. If the worry is that participants might think one end of the numbering scale might be less desirable vs more desirable, then perhaps presenting these labelled options in a nonlinear way might be best. E.g., rather than from left to right (which participants might inadvertently perceive as going from less desirable to more desirable (given that the questions preceding it are presumably going to be presented in this way)), the options for questions 5 and 7 might be presented vertically or via a dropdown box, or in whatever way the authors prefer. This is just a suggestion.

Page 22, question 8: I can't see the wording of the question nor the reasons presented to participants.

Page 22: data exclusion attention/comprehension check: what is the question? Also, (I may have missed it but) the authors should be clear if excluding participants who took part in the pilot studies a and b from taking part in main study.

Participants: Is there a reason the authors are not including any Prolific participants? If the worry is that MTurk workers are different from prolific workers, then generalizability beyond MTurk will be an issue (and should be stated). In the same way that the authors propose to collect data from student participants at different universities, I would suggest that online participants from different platforms should not be a problem. Prolific allows prespecifying that participants be from the US. Of course, I'm not suggesting that there needs to be equal number of MTurk and Prolific participants, but merely that the authors may want to consider also including studies being run on Prolific (just a suggestion).

Issues of clarity/argumentation in the introduction:

Page 4, lines 29-31, "going against participants' wishes could undermine public trust in science even among non-participants, if they feel we are mistreating participants": I don't follow the logic of the argument that leads to the final sentence. The paragraph is about participants potentially being driven away from participating due to their preferences being ignored. So how does it follow that the public trust in science would be diminished if they perceived that we mistreat participants? Is ignoring their preferences for data sharing, or using their data to p-hack, going to be perceived as mistreatment? I think the final sentence of the paragraph requires more reasoning but it would be a digression from the point of the paper without adding much added value.

Page 5 lines 20-25: The arguments leading up to this paragraph have been (1) researchers are not necessarily incentivized to produce good replicable work and so this may have created norms and practices that are not optimal for science, and getting participants' views may be one way for checks and balances against this, and (2) participants should be informed about current research practices, potentially as part of informed consent, and thus their views are important because then we can see what their perceptions are. When the paragraph speaks of blind spots, I found it confusing. On my second reading I realized the blind spots are with regards to point (1). So maybe something to make this clearer in the paragraph will benefit readers.

Minor points

Page 5 line 19, the paragraph starts, "to solve this inconsistency". This statement is ambiguous. Is it referring to "mismatch between their expectations and the reality of research"? If so, then the ambiguity can be cleared up with something like, "If we do find that there is an inconsistency between participants' expectations and research practices, scientists have several options".

Page 4, lines 26-29: not only might participants stop from participating, but they may continue to participate but lose the motivation to provide high quality data.

Page 17 lines 32-36: although 89% is quite low for finding scientific fraud unacceptable, and may be due to nonserious reporting, it is not too different from some of the results from Picket and Roche who found 91% reported fraud as unacceptable (Study 2), though their MTurk sample had 96% finding it unacceptable.

Interesting note (Page 18, lines 31-32): Picket and Roche's findings on selective reporting are very consistent with Pilot B results on all QRPs. The finding from Pilot B suggests that over 75% found all QRPs unacceptable; 71% of Picket and Roche's large MTurk sample considered selective reporting unacceptable, though they used the binary yes/no response options.

Study selection, page 20, lines 31-33: what is the time-frame of the study? Perhaps this should be prespecified.

I always sign my reviews,
Farid Anvari

Reviewer: 5

Comments to the Author(s)

Full comments are included in the attached document but we respond briefly to the specific RSOS questions here

The scientific validity of the research question(s):

This is Stage 1 Registered Report for a study proposing to characterise English-speaking, psychology research participants' views on 1) select questionable research practices (taken from John et al 2012) and 2) open science practices (data sharing, replication, open methods and open access) online. We feel that the proposed study would be a timely and meaningful contribution to the literature

The logic, rationale, and plausibility of the proposed hypotheses:

The study is primarily descriptive and doesn't offer any specific hypotheses. However, we ask that the authors add hypotheses regarding the t-tests included in their analysis plan.

The soundness and feasibility of the methodology and analysis pipeline (including statistical power analysis where applicable)

The overall plan appears sound however we would like to see more description about how the t-tests will be conducted. The authors should provide a power analysis for these t-tests in addition to the power analysis currently included. We are also concerned that the authors' choice of confidence intervals may not be suitable for multinomial proportions and have suggested an alternative R package that might work better.

Whether the clarity and degree of methodological detail would be sufficient to replicate exactly the proposed experimental procedures and analysis pipeline

At this point the authors appear to have provided materials for their pilot studies but not for the Registered Report study. These should be made available and more information should be provided about the statistical analyses.

Whether the authors provide a sufficiently clear and detailed description of the methods to prevent undisclosed flexibility in the experimental procedures or analysis pipeline

If the authors provide the requested information about the materials and t-test and confidence interval calculations there will be no room for flexibility in experimental procedures or analyses

Whether the authors have considered sufficient outcome-neutral conditions (e.g. positive controls) for ensuring that the results obtained are able to test the stated hypotheses:

NA

Review Conducted by Hannah Fraser and Daniel Hamilton

Author's Response to Decision Letter for (RSOS-200048.R0)

See Appendix C.

RSOS-200048.R1 (Revision)

Review form: Reviewer 1

Do you have any ethical concerns with this paper?

No

Recommendation?

Accept in principle

Comments to the Author(s)

I'd like to thank the authors for their careful attention to the reviews and think that the information from these studies will be very informative.

Review form: Reviewer 2 (Loukia Tzavella)

Do you have any ethical concerns with this paper?

No

Recommendation?

Accept in principle

Comments to the Author(s)

The scientific validity of the research question(s):

Consistent with the first review, the research questions are scientifically valid.

The logic, rationale, and plausibility of the proposed hypotheses:

The revised introduction provides a very clear rationale for the study. The authors have addressed previous comments and concepts that are important for understanding the rationale and research questions are now defined early on (e.g. QRPs).

The soundness and feasibility of the methodology and analysis pipeline (including statistical power analysis where applicable):

Previous comments on statistical analysis have been addressed. The soundness and feasibility of the methodology have further been established through the pilot studies.

Whether the clarity and degree of methodological detail would be sufficient to replicate exactly the proposed experimental procedures and analysis pipeline:

The authors have adopted a high standard of transparency and methods reproducibility throughout the manuscript. Study materials are also available on OSF (full survey).

Whether the authors provide a sufficiently clear and detailed description of the methods to prevent undisclosed flexibility in the experimental procedures or analysis pipeline:

The authors have provided enough details regarding the methods and analyses (including R code) to prevent undisclosed flexibility at Stage 2.

Whether the authors have considered sufficient outcome-neutral conditions (e.g. positive controls) for ensuring that the results obtained are able to test the stated hypotheses:

The authors have included a proposed robustness check (several Qs have two versions) and an attention check. The only minor comment/suggestion I have regarding the attention check is to make it easier for the reader to figure out what that involves, without having to go through the additional materials on OSF. This can easily be done (possibly at Stage 2) by simply adding a few words [Page 19, Line 52]: "participants have to report what they had for dinner."; or verbatim "(i.e., What did you have for dinner last night?)".

I have no further comments and I look forward to reading the Stage 2 manuscript!

Loukia Tzavella

Review form: Reviewer 3 (Robert Thibault)

Do you have any ethical concerns with this paper?

No

Recommendation?

Accept in principle

Comments to the Author(s)

The authors have addressed all my comments.

Review form: Reviewer 4 (Farid Anvari)

Do you have any ethical concerns with this paper?

No

Recommendation?

Accept in principle

Comments to the Author(s)

This is round 2 of reviews, following revisions and resubmission, of a Stage 1 registered report (RR). I am happy with the revisions made in the paper specifically addressing my previous concerns. I'm also particularly happy about the decision to remove hypothesis testing and report the results descriptively, with the inclusion of confidence bounds. I think the study is ready to go and I'm excited about the results (I think they'll be very interesting and informative for the scientific community). I have no further suggestions and look forward to the Stage 2 review.

My explicit comments on each of the journal review criteria:

The scientific validity of the research question(s)

I believe the research questions are interesting and important for the scientific community and have scientific validity.

The logic, rationale, and plausibility of the proposed hypotheses

There are no hypotheses, though the logic and rationale for the importance of the research questions and the methods are sound.

The soundness and feasibility of the methodology and analysis pipeline (including statistical power analysis where applicable)

The methods and analysis pipeline are sound for addressing the research questions.

Whether the clarity and degree of methodological detail would be sufficient to replicate exactly the proposed experimental procedures and analysis pipeline

Together with the supplement there is enough clarity and methodological detail for an exact replication.

Whether the authors provide a sufficiently clear and detailed description of the methods to prevent undisclosed flexibility in the experimental procedures or analysis pipeline

I see little to no room for undisclosed flexibility.

Whether the authors have considered sufficient outcome-neutral conditions (e.g. positive controls) for ensuring that the results obtained are able to test the stated hypotheses

The authors have taken sufficient steps to obtain high quality data.

Review form: Reviewer 5 (Hannah Fraser)

Do you have any ethical concerns with this paper?

No

Recommendation?

Accept in principle

Comments to the Author(s)

The authors have done a great job of addressing the reviewer comments. The article flows well now and is more readable for an interdisciplinary audience.

We apologize for not including the references we had asked the authors to include for questionable research practices. We recommend that the authors include reference to:

Agnoli F, Wicherts JM, Veldkamp CLS, Albiero P, Cubelli R. Questionable research practices among Italian research psychologists. *PLoS One*. 2017;12: 1-17. doi:10.1371/journal.pone.0172792

Makel MC, Hodges J, Cook BG, Plucker JA. Questionable and open research practices in education research. *EdarXiv*. 2019; doi:https://doi.org/10.35542/osf.io/f7srb

Fox NW, Honeycutt N, Jussim L. How Many Psychologists Use Questionable Research Practices? Estimating the Population Size of Current QRP Users. 2018; 1-10.

In our previous review of this article we noted that referring to websites is potentially problematic but the authors rebutted this point. We still feel strongly that referencing websites that do not have DOIs is not good practice. We are particularly concerned about the references to company website pages (e.g. FAQ, or 'about' page) which are regularly updated as policies change - If someone looked at these references in the future they might find completely contradictory information and they would have no idea what the sites said when you referenced them. We would prefer it if the authors either a) found reports or articles with DOIs that speak to these issues or b) did not provide references for these statements at all (this is less important for the Bastian blog which may not be available in the future but is unlikely to change in message).

Overall this is a really well written and interesting article and we cant wait to see the results.

Review by Hannah Fraser and Daniel Hamilton

Decision letter (RSOS-200048.R1)

Dear Ms Bottesini,

On behalf of the Editors, I am pleased to inform you that your Manuscript RSOS-200048.R1 entitled "What Do Participants Think of Our Research Practices? An Examination of Behavioral Psychology Participants' Preferences" has been accepted in principle for publication in Royal Society Open Science subject to minor revision in accordance with the referee and editor suggestions. Please find their comments at the end of this email.

The reviewers and handling editors have recommended publication, but also suggest some minor revisions to your manuscript. Therefore, I invite you to respond to the comments and revise your manuscript.

Please can you submit the revised version of your manuscript within 7 days (i.e. by the 29-May-2020). If you do not think you will be able to meet this date please let me know immediately.

When submitting your revised manuscript, you will be able to respond to the comments made by the referees and you should upload a file "Response to Referees". You can use this to document any changes you make to the original manuscript. In order to expedite the processing of the revised manuscript, please be as specific as possible in your response to the referees.

Full author guidelines can be found here <https://royalsocietypublishing.org/rsos/registered-reports>.

on behalf of Professor Chris Chambers (Subject Editor, Royal Society Open Science)
openscience@royalsociety.org

Associate Editor Comments to Author (Professor Chris Chambers):

Associate Editor: 1

Comments to the Author:

All five reviewers have now assessed the revised the manuscript and are broadly satisfied. A few minor issues remain to be addressed concerning the attention check (Review 2) and referencing

(Review 5). Concerning the point about referencing of websites, the authors note in their previous response that separately archiving the content (e.g. through screenshots) could conflict with copyright law. I wonder if a compromise solution could be to register each of the websites with <http://web.archive.org/> and then link to these URLs in the references? Although entries in <http://web.archive.org/> are not necessarily permanent, and can be removed on request from site owners, they are considerably more stable than the regular internet. I will leave this for the authors to consider. Provided the authors are able to respond thoroughly to these remaining points in a minor revision, IPA should be forthcoming without requiring further in-depth Stage 1 review.

Reviewer comments to Author:

Reviewer: 1

Comments to the Author(s)

I'd like to thank the authors for their careful attention to the reviews and think that the information from these studies will be very informative.

Reviewer: 2

Comments to the Author(s)

The scientific validity of the research question(s):

Consistent with the first review, the research questions are scientifically valid.

The logic, rationale, and plausibility of the proposed hypotheses:

The revised introduction provides a very clear rationale for the study. The authors have addressed previous comments and concepts that are important for understanding the rationale and research questions are now defined early on (e.g. QRPs).

The soundness and feasibility of the methodology and analysis pipeline (including statistical power analysis where applicable):

Previous comments on statistical analysis have been addressed. The soundness and feasibility of the methodology have further been established through the pilot studies.

Whether the clarity and degree of methodological detail would be sufficient to replicate exactly the proposed experimental procedures and analysis pipeline:

The authors have adopted a high standard of transparency and methods reproducibility throughout the manuscript. Study materials are also available on OSF (full survey).

Whether the authors provide a sufficiently clear and detailed description of the methods to prevent undisclosed flexibility in the experimental procedures or analysis pipeline:

The authors have provided enough details regarding the methods and analyses (including R code) to prevent undisclosed flexibility at Stage 2.

Whether the authors have considered sufficient outcome-neutral conditions (e.g. positive controls) for ensuring that the results obtained are able to test the stated hypotheses:

The authors have included a proposed robustness check (several Qs have two versions) and an attention check. The only minor comment/suggestion I have regarding the attention check is to

make it easier for the reader to figure out what that involves, without having to go through the additional materials on OSF. This can easily be done (possibly at Stage 2) by simply adding a few words [Page 19, Line 52]: "participants have to report what they had for dinner."; or verbatim "(i.e., What did you have for dinner last night?)".

I have no further comments and I look forward to reading the Stage 2 manuscript!

Loukia Tzavella

Reviewer: 3

Comments to the Author(s)

The authors have addressed all my comments.

Reviewer: 4

Comments to the Author(s)

This is round 2 of reviews, following revisions and resubmission, of a Stage 1 registered report (RR). I am happy with the revisions made in the paper specifically addressing my previous concerns. I'm also particularly happy about the decision to remove hypothesis testing and report the results descriptively, with the inclusion of confidence bounds. I think the study is ready to go and I'm excited about the results (I think they'll be very interesting and informative for the scientific community). I have no further suggestions and look forward to the Stage 2 review.

My explicit comments on each of the journal review criteria:

The scientific validity of the research question(s)

I believe the research questions are interesting and important for the scientific community and have scientific validity.

The logic, rationale, and plausibility of the proposed hypotheses

There are no hypotheses, though the logic and rationale for the importance of the research questions and the methods are sound.

The soundness and feasibility of the methodology and analysis pipeline (including statistical power analysis where applicable)

The methods and analysis pipeline are sound for addressing the research questions.

Whether the clarity and degree of methodological detail would be sufficient to replicate exactly the proposed experimental procedures and analysis pipeline

Together with the supplement there is enough clarity and methodological detail for an exact replication.

Whether the authors provide a sufficiently clear and detailed description of the methods to prevent undisclosed flexibility in the experimental procedures or analysis pipeline

I see little to no room for undisclosed flexibility.

Whether the authors have considered sufficient outcome-neutral conditions (e.g. positive controls) for ensuring that the results obtained are able to test the stated hypotheses

The authors have taken sufficient steps to obtain high quality data.

Reviewer: 5

Comments to the Author(s)

The authors have done a great job of addressing the reviewer comments. The article flows well now and is more readable for an interdisciplinary audience.

We apologize for not including the references we had asked the authors to include for questionable research practices. We recommend that the authors include reference to:

Agnoli F, Wicherts JM, Veldkamp CLS, Albiero P, Cubelli R. Questionable research practices among Italian research psychologists. *PLoS One*. 2017;12: 1–17. doi:10.1371/journal.pone.0172792
 Makel MC, Hodges J, Cook BG, Plucker JA. Questionable and open research practices in education research. *EdarXiv*. 2019; doi:https://doi.org/10.35542/osf.io/f7srb
 Fox NW, Honeycutt N, Jussim L. How Many Psychologists Use Questionable Research Practices ? Estimating the Population Size of Current QRP Users. 2018; 1–10.

In our previous review of this article we noted that referring to websites is potentially problematic but the authors rebutted this point. We still feel strongly that referencing websites that do not have DOIs is not good practice. We are particularly concerned about the references to company website pages (e.g. FAQ, or 'about' page) which are regularly updated as policies change - If someone looked at these references in the future they might find completely contradictory information and they would have no idea what the sites said when you referenced them. We would prefer it if the authors either a) found reports or articles with DOIs that speak to these issues or b) did not provide references for these statements at all (this is less important for the Bastian blog which may not be available in the future but is unlikely to change in message).

Overall this is a really well written and interesting article and we cant wait to see the results.

Review by Hannah Fraser and Daniel Hamilton

Author's Response to Decision Letter for (RSOS-200048.R1)

See Appendix D.

Decision letter (RSOS-200048.R2)

Dear Ms Bottesini,

On behalf of the Editor, I am pleased to inform you that your Manuscript RSOS-200048.R2 entitled "What Do Participants Think of Our Research Practices? An Examination of Behavioral Psychology Participants' Preferences" has been accepted in principle for publication in Royal Society Open Science.

You may now progress to Stage 2 and complete the study as approved. Before commencing data collection we ask that you:

- 1) Update the journal office as to the anticipated completion date of your study.
- 2) Register your approved protocol on the Open Science Framework (<https://osf.io/>) or other recognised repository, either publicly or privately under embargo until submission of the Stage 2 manuscript. Please note that a time-stamped, independent registration of the protocol is mandatory under journal policy, and manuscripts that do not conform to this requirement cannot

be considered at Stage 2. The protocol should be registered unchanged from its current approved state, with the time-stamp preceding implementation of the approved study design.

Following completion of your study, we invite you to resubmit your paper for peer review as a Stage 2 Registered Report. Please note that your manuscript can still be rejected for publication at Stage 2 if the Editors consider any of the following conditions to be met:

- The results were unable to test the authors' proposed hypotheses by failing to meet the approved outcome-neutral criteria.
- The authors altered the Introduction, rationale, or hypotheses, as approved in the Stage 1 submission.
- The authors failed to adhere closely to the registered experimental procedures. Please note that any deviations from the approved experimental procedures must be communicated to the editor immediately for approval, and prior to the completion of data collection. Failure to do so can result in revocation of in-principle acceptance and rejection at Stage 2 (see complete guidelines for further information).
- Any post-hoc (unregistered) analyses were either unjustified, insufficiently caveated, or overly dominant in shaping the authors' conclusions.
- The authors' conclusions were not justified given the data obtained.

We encourage you to read the complete guidelines for authors concerning Stage 2 submissions at <https://royalsocietypublishing.org/rsos/registered-reports#ReviewerGuideRegRep>. Please especially note the requirements for data sharing, reporting the URL of the independently registered protocol, and that withdrawing your manuscript will result in publication of a Withdrawn Registration.

Please note that Royal Society Open Science will introduce article processing charges for all new submissions received from 1 January 2018. Registered Reports submitted and accepted after this date will ONLY be subject to a charge if they subsequently progress to and are accepted as Stage 2 Registered Reports. If your manuscript is submitted and accepted for publication after 1 January 2018 (i.e. as a full Stage 2 Registered Report), you will be asked to pay the article processing charge, unless you request a waiver and this is approved by Royal Society Publishing. You can find out more about the charges at <https://royalsocietypublishing.org/rsos/charges>. Should you have any queries, please contact openscience@royalsociety.org.

Once again, thank you for submitting your manuscript to Royal Society Open Science and we look forward to receiving your Stage 2 submission. If you have any questions at all, please do not hesitate to get in touch. We look forward to hearing from you shortly with the anticipated submission date for your stage two manuscript.

on behalf of Professor Chris Chambers (Registered Reports Editor, Royal Society Open Science)
openscience@royalsociety.org

Author's Response to Decision Letter for (RSOS-200048.R2)

See Appendix E.

RSOS-200048.R3

Review form: Reviewer 1

Is the manuscript scientifically sound in its present form?

Yes

Are the interpretations and conclusions justified by the results?

Yes

Is the language acceptable?

Yes

Do you have any ethical concerns with this paper?

No

Have you any concerns about statistical analyses in this paper?

No

Recommendation?

Accept with minor revision

Comments to the Author(s)

I'm reading this manuscript as one of the original reviewers on the previous iterations of the Stage 1 manuscript. I applaud the authors on executing an interesting study mostly in line to what was proposed. The fact that the wording/framing didn't matter much (from my interpretation, although it shifts things a little) was interesting and informative too. They handled the data quality issue well and clearly distinguished between what was planned v. not. I had few recommendations because the authors did everything they said they would.

One recommendation (which the authors are free to decline) would be to combine Tables 4 and 9 (with and without the strict exclusions) to resemble Table 6. They could also expand Table 7 to include the pre-registered exclusions. I recommend this because it presents the clearest point-by-point comparison for how the results change across exclusion methods. It presents info in a more accessible way by enabling people to say, "oh, here's the range of people that find it acceptable, depending on the source of data and how stringently you screen the data." Might be useful for future readers.

It might be good to put the additional exclusion criteria earlier in the manuscript. I think the exclusions sound mostly reasonable and everything is transparently presented. But doing so in a data exclusions section or fully describing them in the Analysis section would be nice.

In the Intro, there's a few places where the language is still future oriented (from the Stage 1 manuscript). They could maybe revise it to be tense-consistent or "we planned to do xx" if they feel conflicted about that.

Excellent study – really interesting.

Review form: Reviewer 2 (Loukia Tzavella)

Is the manuscript scientifically sound in its present form?

Yes

Are the interpretations and conclusions justified by the results?

Yes

Is the language acceptable?

Yes

Do you have any ethical concerns with this paper?

No

Have you any concerns about statistical analyses in this paper?

No

Recommendation?

Accept as is

Comments to the Author(s)

I am recommending that the Stage 2 Registered Report is accepted without revisions as, in my opinion, the manuscript meets all the criteria for publication at this stage.

First, the data were sufficient to answer the proposed research questions from Stage 1 and any limitations or issues with interpretation are adequately covered in the Discussion. Second, the Introduction, rationale and research questions have not been altered in the Stage 2 submission. The authors have not deviated from their proposed methods and procedures and any such cases are transparently reported in the main text with adequate justification. Authors have provided the reader with the opportunity to interpret findings from both perspectives as descriptive statistics are presented for both preregistered and non-preregistered data exclusion criteria. Importantly, the authors warrant caution in the interpretation of findings and openly acknowledge the potential introduction of bias in the latter case. Any analyses that were labelled as exploratory in the Stage 1 manuscript have been appropriately described in this manuscript and their inclusion was justified.

Overall, this Registered Report was very interesting to read and the discussion raises important points for consideration with regards to informed consent and participants' views of open research practices. I also believe that this manuscript can pave the way for more research on this topic (e.g. exploring other disciplines and types of research such as clinical trials, interviews etc.).

Review form: Reviewer 3 (Robert Thibault)

Is the manuscript scientifically sound in its present form?

Yes

Are the interpretations and conclusions justified by the results?

Yes

Is the language acceptable?

Yes

Do you have any ethical concerns with this paper?

No

Have you any concerns about statistical analyses in this paper?

No

Recommendation?

Accept with minor revision

Comments to the Author(s)

The online review system asks that I “comment explicitly on each of the following points in your comments to the authors” (see Appendix F).

1. Whether the data are able to test the authors’ proposed hypotheses by passing the approved outcome-neutral criteria (such as absence of floor and ceiling effects or success of positive controls)
2. Whether the Introduction, rationale and stated hypotheses are the same as the approved Stage 1 submission
3. Whether the authors adhered precisely to the registered experimental procedures
4. Where applicable, whether any unregistered exploratory statistical analyses are justified, methodologically sound, and informative
5. Whether the authors’ conclusions are justified given the data

I haven’t re-read the Stage 1 submission (which I reviewed 2 years ago). Based on a reading of the Stage 2 submission, it appears that the Stage 1 plan was followed and that a few deviations were identified as such and properly justified.

Yes, the authors take care to explain the robustness of their conclusions based on the data. Conclusions are appropriate.

Below, I provide additional comments. I also attach a pdf of the manuscript with several inline comment boxes.

1. The abstract could contain much more information about the results by adding only a few sentences. The numbers for each of the 8 questions could be given, and the numbers for scientific fraud should also be given. The abstract finishes with a statement about violating participants’ expectations, but I don’t think the study asks about their expectations, but rather only about what is acceptable. Perhaps that could be reworded.
2. The section “Pilot Studies” seems non-systematic. It presents a few numbers, but it’s unclear why these numbers were selected to present. Perhaps summarize all the main pilot results in a table. At the moment, some people may see this as selective reporting of the pilot studies. It’s also unclear whether the pilot studies were given after participation in a ‘base’ study.
3. Percentages are given at several points in the manuscript without a confidence interval or a denominator. With only the percentage the reader can’t know whether 70% is 7/10 people or 7000/10000 people.

4. The manuscript is very detailed. Although this makes it quite long. I would not recommend reducing length at the cost of precision or clarity. However, some material is redundant and the authors may want to reduce the length (of course, this is not necessary, it would just render the paper more digestible).

5. Almost all the tables in the results section could be collapsed into a single table on a landscape page. This would facilitate comparing the results across all the different sensitivity tests and simplify the results section.

6. I recommend giving the numbers related to fraud *before* presenting the main results. This is important information that the reader should know when interpreting the main results, not after interpreting them.

The manuscript is very detailed and transparent. It appears all the materials and data are shared on the OSF. The authors are cautious and appropriate with their conclusions.

I always sign my reviews,
Robert Thibault

Review form: Reviewer 4 (Farid Anvari)

Is the manuscript scientifically sound in its present form?

Yes

Are the interpretations and conclusions justified by the results?

Yes

Is the language acceptable?

Yes

Do you have any ethical concerns with this paper?

Yes

Have you any concerns about statistical analyses in this paper?

No

Recommendation?

Accept as is

Comments to the Author(s)

After reading the abstract I was quite excited to get into the paper. This indicates that, at least to some researchers such as myself, the research question is very interesting indeed.

The figures do a great job of illustrating the descriptive statistics. Some very interesting results.

For example, almost a quarter of participants (almost 20% after more strict exclusions) found p-hacking, filedrawing, and HARKing acceptable or were indifferent to it.

I liked the paper, and in particular how the analyses were dealt with and interpreted. I only have some very minor points that the authors can feel free to ignore.

When comparing the MTurk vs subject pool, why are the results presented in Table 7 (and figure 3) based on nonpreregistered exclusions rather than the preregistered exclusions? After reading on, I saw that the study-partitioned results for the fraud item with preregistered exclusions was

reported later on pp. 34-35. Would it make sense to provide the partitioned-results from preregistered exclusions first and then use what's found for responses to the fraud questions as justification for incorporating the strict exclusions? I think this would also provide the stronger justification for why it's ok that most of the exclusions were from MTurk participants (e.g., the time criterion).

Regarding the strict, nonpreregistered exclusion criteria, I can see the authors note in the discussion that they explored various ways of excluding participants that they don't report. And I really like how the authors describe all of this, justify their decisions, and interpret the results. So, again, this is quite a minor point. The authors note that "We chose a cut-off of 0.8 and excluded participants with smaller SDs, which excluded 247 participants (13.2%)." (p. 36). I know it's a somewhat arbitrary cutoff, and the authors are clear about this. As robustness checks and to provide stronger justification for their choices the authors could do a couple of other things. 1. They could briefly report results from using other cutoffs (e.g., do the results change much?) 2. what do the results look like without this specific criterion for exclusion? The results of this too can be very briefly described.

I really liked the limitation in the first paragraph on p. 44. One thing to note, though, is the last sentence, "It would be difficult to imagine a way in which we could ask the same question that would sway participants enough to change the general consensus we see across participants for most of the questions." I don't think the important point is that the questions could be worded so that people might be generally "for" any of these practices, but instead that giving a less extreme response might indicate more nuanced views, such as the idea that in some situations the practice may be more or less acceptable, even though it generally might not be.

(and here follows my responses to the points required by the journal)

1. Whether the data are able to test the authors' proposed hypotheses by passing the approved outcome-neutral criteria (such as absence of floor and ceiling effects or success of positive controls)

Yes (though really it's N/A since there were no hypotheses).

2. Whether the Introduction, rationale and stated hypotheses are the same as the approved Stage 1 submission

Yes, though I only did a rough check.

3. Whether the authors adhered precisely to the registered experimental procedures

Yes.

4. Where applicable, whether any unregistered exploratory statistical analyses are justified, methodologically sound, and informative

Yes

5. Whether the authors' conclusions are justified given the data

Yes.

I sign my reviews,

Farid Anvari

Decision letter (RSOS-200048.R3)

Dear Ms Bottesini:

On behalf of the Editor, I am pleased to inform you that your Stage 2 Registered Report RSOS-200048.R3 entitled "What Do Participants Think of Our Research Practices? An Examination of Behavioral Psychology Participants' Preferences" has been deemed suitable for publication in Royal Society Open Science subject to minor revision in accordance with the referee suggestions. Please find the referees' comments at the end of this email.

The reviewers and Subject Editor have recommended publication, but also suggest some minor revisions to your manuscript. We invite you to respond to the comments and revise your manuscript. Below the referees' and Editors' comments (where applicable) we provide additional requirements. Final acceptance of your manuscript is dependent on these requirements being met. We provide guidance below to help you prepare your revision.

Please submit your revised manuscript and required files (see below) no later than 7 days from today's (ie 28-Jan-2022) date. Note: the ScholarOne system will 'lock' if submission of the revision is attempted 7 or more days after the deadline. If you do not think you will be able to meet this deadline please contact the editorial office immediately.

on behalf of Professor Chris Chambers
(Registered Reports Editor, Royal Society Open Science)
openscience@royalsociety.org

Associate Editor Comments to Author (Professor Chris Chambers):

Four of the original reviewers kindly returned to evaluate the completed Stage 2 manuscript. As you will, their evaluations are very positive, pointing at a few relatively minor issues to address in revision, primarily concerning the presentation of results (including reporting of robustness checks). Provided you are able to address these comments in a revision, full acceptance should be forthcoming without further in-depth review.

Reviewer Comments to Author:

Reviewer: 1

Comments to the Author(s)

I'm reading this manuscript as one of the original reviewers on the previous iterations of the Stage 1 manuscript. I applaud the authors on executing an interesting study mostly in line to what was proposed. The fact that the wording/framing didn't matter much (from my interpretation, although it shifts things a little) was interesting and informative too. They handled the data quality issue well and clearly distinguished between what was planned v. not. I had few recommendations because the authors did everything they said they would.

One recommendation (which the authors are free to decline) would be to combine Tables 4 and 9 (with and without the strict exclusions) to resemble Table 6. They could also expand Table 7 to include the pre-registered exclusions. I recommend this because it presents the clearest point-by-point comparison for how the results change across exclusion methods. It presents info in a more accessible way by enabling people to say, “oh, here’s the range of people that find it acceptable, depending on the source of data and how stringently you screen the data.” Might be useful for future readers.

It might be good to put the additional exclusion criteria earlier in the manuscript. I think the exclusions sound mostly reasonable and everything is transparently presented. But doing so in a data exclusions section or fully describing them in the Analysis section would be nice.

In the Intro, there’s a few places where the language is still future oriented (from the Stage 1 manuscript). They could maybe revise it to be tense-consistent or “we planned to do xx” if they feel conflicted about that.

Excellent study – really interesting.

Reviewer: 2

Comments to the Author(s)

I am recommending that the Stage 2 Registered Report is accepted without revisions as, in my opinion, the manuscript meets all the criteria for publication at this stage.

First, the data were sufficient to answer the proposed research questions from Stage 1 and any limitations or issues with interpretation are adequately covered in the Discussion. Second, the Introduction, rationale and research questions have not been altered in the Stage 2 submission. The authors have not deviated from their proposed methods and procedures and any such cases are transparently reported in the main text with adequate justification. Authors have provided the reader with the opportunity to interpret findings from both perspectives as descriptive statistics are presented for both preregistered and non-preregistered data exclusion criteria. Importantly, the authors warrant caution in the interpretation of findings and openly acknowledge the potential introduction of bias in the latter case. Any analyses that were labelled as exploratory in the Stage 1 manuscript have been appropriately described in this manuscript and their inclusion was justified.

Overall, this Registered Report was very interesting to read and the discussion raises important points for consideration with regards to informed consent and participants’ views of open research practices. I also believe that this manuscript can pave the way for more research on this topic (e.g. exploring other disciplines and types of research such as clinical trials, interviews etc.).

Reviewer: 3

Comments to the Author(s)

Attached file: 2201_RSOS_Stage2RR_commented.pdf

The online review system asks that I “comment explicitly on each of the following points in your comments to the authors”

1. Whether the data are able to test the authors’ proposed hypotheses by passing the approved outcome-neutral criteria (such as absence of floor and ceiling effects or success of positive controls)

2. Whether the Introduction, rationale and stated hypotheses are the same as the approved Stage 1 submission
3. Whether the authors adhered precisely to the registered experimental procedures
4. Where applicable, whether any unregistered exploratory statistical analyses are justified, methodologically sound, and informative
5. Whether the authors' conclusions are justified given the data

I haven't re-read the Stage 1 submission (which I reviewed 2 years ago). Based on a reading of the Stage 2 submission, it appears that the Stage 1 plan was followed and that a few deviations were identified as such and properly justified.

Yes, the authors take care to explain the robustness of their conclusions based on the data. Conclusions are appropriate.

Below, I provide additional comments. I also attach a pdf of the manuscript with several inline comment boxes. (2201_RSOS_Stage2RR_commented.pdf, attached)

1. The abstract could contain much more information about the results by adding only a few sentences. The numbers for each of the 8 questions could be given, and the numbers for scientific fraud should also be given. The abstract finishes with a statement about violating participants' expectations, but I don't think the study asks about their expectations, but rather only about what is acceptable. Perhaps that could be reworded.
2. The section "Pilot Studies" seems non-systematic. It presents a few numbers, but it's unclear why these numbers were selected to present. Perhaps summarize all the main pilot results in a table. At the moment, some people may see this as selective reporting of the pilot studies. It's also unclear whether the pilot studies were given after participation in a 'base' study.
3. Percentages are given at several points in the manuscript without a confidence interval or a denominator. With only the percentage the reader can't know whether 70% is 7/10 people or 7000/10000 people.
4. The manuscript is very detailed. Although this makes it quite long. I would not recommend reducing length at the cost of precision or clarity. However, some material is redundant and the authors may want to reduce the length (of course, this is not necessary, it would just render the paper more digestible).
5. Almost all the tables in the results section could be collapsed into a single table on a landscape page. This would facilitate comparing the results across all the different sensitivity tests and simplify the results section.
6. I recommend giving the numbers related to fraud *before* presenting the main results. This is important information that the reader should know when interpreting the main results, not after interpreting them.

The manuscript is very detailed and transparent. It appears all the materials and data are shared on the OSF. The authors are cautious and appropriate with their conclusions.

I always sign my reviews,
Robert Thibault

Reviewer: 4

Comments to the Author(s)

After reading the abstract I was quite excited to get into the paper. This indicates that, at least to some researchers such as myself, the research question is very interesting indeed.

The figures do a great job of illustrating the descriptive statistics. Some very interesting results.

For example, almost a quarter of participants (almost 20% after more strict exclusions) found p-hacking, filedrawing, and HARKing acceptable or were indifferent to it.

I liked the paper, and in particular how the analyses were dealt with and interpreted. I only have some very minor points that the authors can feel free to ignore.

When comparing the MTurk vs subject pool, why are the results presented in Table 7 (and figure 3) based on nonpreregistered exclusions rather than the preregistered exclusions? After reading on, I saw that the study-partitioned results for the fraud item with preregistered exclusions was reported later on pp. 34-35. Would it make sense to provide the partitioned-results from preregistered exclusions first and then use what's found for responses to the fraud questions as justification for incorporating the strict exclusions? I think this would also provide the stronger justification for why it's ok that most of the exclusions were from MTurk participants (e.g., the time criterion).

Regarding the strict, nonpreregistered exclusion criteria, I can see the authors note in the discussion that they explored various ways of excluding participants that they don't report. And I really like how the authors describe all of this, justify their decisions, and interpret the results. So, again, this is quite a minor point. The authors note that "We chose a cut-off of 0.8 and excluded participants with smaller SDs, which excluded 247 participants (13.2%)." (p. 36). I know it's a somewhat arbitrary cutoff, and the authors are clear about this. As robustness checks and to provide stronger justification for their choices the authors could do a couple of other things. 1. They could briefly report results from using other cutoffs (e.g., do the results change much?) 2. what do the results look like without this specific criterion for exclusion? The results of this too can be very briefly described.

I really liked the limitation in the first paragraph on p. 44. One thing to note, though, is the last sentence, "It would be difficult to imagine a way in which we could ask the same question that would sway participants enough to change the general consensus we see across participants for most of the questions." I don't think the important point is that the questions could be worded so that people might be generally "for" any of these practices, but instead that giving a less extreme response might indicate more nuanced views, such as the idea that in some situations the practice may be more or less acceptable, even though it generally might not be.

(and here follows my responses to the points required by the journal)

1. Whether the data are able to test the authors' proposed hypotheses by passing the approved outcome-neutral criteria (such as absence of floor and ceiling effects or success of positive controls)

Yes (though really it's N/A since there were no hypotheses).

2. Whether the Introduction, rationale and stated hypotheses are the same as the approved Stage 1 submission

Yes, though I only did a rough check.

3. Whether the authors adhered precisely to the registered experimental procedures

Yes.

4. Where applicable, whether any unregistered exploratory statistical analyses are justified, methodologically sound, and informative

Yes

5. Whether the authors' conclusions are justified given the data

Yes.

I sign my reviews,
Farid Anvari

===PREPARING YOUR MANUSCRIPT===

one version should clearly identify all the changes that have been made (for instance, in coloured highlight, in bold text, or tracked changes);

===PREPARING YOUR REVISION IN SCHOLARONE===

-- If you are requesting an article processing charge waiver, you must select the relevant waiver option (if requesting a discretionary waiver, the form should have been uploaded, see 'File upload' above).

-- If you have uploaded any electronic supplementary (ESM) files, please ensure you follow the guidance at <https://royalsociety.org/journals/authors/author-guidelines/#supplementary-material> to include a suitable title and informative caption. An example of appropriate titling and captioning may be found at https://figshare.com/articles/Table_S2_from_Is_there_a_trade-off_between_peak_performance_and_performance_breadth_across_temperatures_for_aerobic_scope_in_teleost_fishes_/3843624.

Author's Response to Decision Letter for (RSOS-200048.R3)

See Appendix G.

Decision letter (RSOS-200048.R4)

Dear Ms Bottesini:

It is a pleasure to accept your manuscript entitled "What Do Participants Think of Our Research Practices? An Examination of Behavioral Psychology Participants' Preferences" in its current form for publication in Royal Society Open Science. Please note that there is a one remaining comment below from the action editor for your consideration.

Please ensure that you send to the editorial office an editable version of your accepted manuscript, and individual files for each figure and table included in your manuscript. You can send these in a zip folder if more convenient. Failure to provide these files may delay the processing of your proof.

Thank you for your fine contribution. On behalf of the Editors of Royal Society Open Science, we look forward to your continued contributions to the journal.

on behalf of Professor Chris Chambers (Subject Editor)
openscience@royalsociety.org

Associate Editor Comments to Author (Professor Chris Chambers):

Thank you for the very thorough revision, which I'm happy to now accept without further review. Concerning your query about changing the future tense in the Introduction and Method sections to past tense, I would recommend shifting to past tense as it will read more intuitively for readers. Doing so is within the bounds of the acceptable changes to the Introduction and Methods at Stage 2, and is typically what happens in RRs (at least for submissions that I edit). However, I see no reason to hold up full acceptance on this basis, and I am happy to leave this decision to you. If you would like to change the tenses, feel free to do so at the proof stage.

General evaluation of the S1 RR criteria:

1. Scientific validity of the research question(s).

The research questions of the proposed S1 protocol are not only scientifically valid but also of great importance. Research in psychological science (and other disciplines) is supposed to benefit the public (i.e., the collective good) and yet a lot of the research practices that have been identified as 'questionable' may actually have the opposite effect. This study may provide much-needed evidence to support the claim that these practices are not considered acceptable by participants, without whom the majority of research would not be possible. I believe that this study has both ethical and practical implications (e.g., policy change).

2. Logic, rationale, and plausibility of the proposed hypotheses.

Although the proposed study does not have specific confirmatory hypotheses, the rationale for the research questions is clearly explained. Any degrees of freedom regarding the researchers' predictions and analyses at Stage 2 have been prevented with the high level of transparency adopted throughout the manuscript (but see minor comment regarding the analyses).

3. Soundness and feasibility of the methodology and analysis pipeline.

The soundness and feasibility of the methodology have been demonstrated in the pilot studies described in the S1 protocol and the proposed methods. A suggestion would be for the 'participants' and 'study selection' sections to be accompanied by a schematic (figure) to make it easier for the reader to follow all the described criteria.

4. Clarity and degree of methodological detail for replication of the proposed experimental procedures and analysis pipeline.

All procedures are described in sufficient detail to allow for direct replication (only a few minor comments).

5. Sufficiently clear and detailed description of the methods to prevent undisclosed flexibility in the experimental procedures or analysis pipeline

The authors have provided sufficiently clear and detailed descriptions of the methods and analyses, but a few minor comments could be addressed for increased specificity.

6. Sufficient outcome-neutral conditions for ensuring that the results obtained are able to test the stated hypotheses.

An attention check has been included, but I am recommending that more details regarding this survey question are provided in the S1 protocol. The authors also use an inclusion criterion for the recruitment of mTurk participants that should minimise the possibility of dishonest/invalid survey responses.

Minor comments:

- There appears to be an issue with some references (e.g., line 19 p. 3) linking to a Zotero page with this message ("You probably got here by mistake").
- *Line 20, p. 3:* Please provide an example or definition of 'top' psychology journals (e.g., high impact factor).
- *Line 33, p. 3:* Rephrase to avoid repetition of 'research practices' in the same sentence and present abbreviation last not first: e.g. "engaging in questionable research practices, or QRPS (as described in John et al. 2012) that can inflate..".
- *Lines 40-54, p. 6:* This paragraph could be integrated into the Methods section.
- **Pilot B figures:** The titles are not very self-explanatory in the current format - as I am guessing similar figures on QRPs may be produced after data collection (e.g., "Figure 3. HARKing."). Increased specificity may also be needed for the

abbreviations used in the figures- e.g., in Figure 4 you could mention in the note what 'Rs' refers to even though it may seem obvious. Correct "fell strongly" in Fig. 4.

- **Sections:** The subsections of the pilots should be consistent (e.g., see "Fraud (n = 114)"). Sample size is not indicated in other subsections while pilot A does not have any subheadings for methods/results. Following the 'fraud' subsection the summary is presented with no subheading which makes it seem like they are part of the same section.
- *Line 38, p. 20:* Include a footnote to explain what HIT stands for.
- *Lines 48-53, p. 22:* "To err on the side of caution" is not very clear. I might have missed this if it is included in OSF materials, but what is the actual open ended attention/comprehension check question? More details should be provided here for increased reproducibility.
- **Missing data [question]:** Is the attention check question enough to catch invalid responses? If the scales default at the middle, can you tell if a response is missed or valid (i.e., do participants get a warning if they haven't interacted with the question at all)? If missed responses are possible, how will you handle missing data (e.g., excluding participants)?
- **Analyses:** Please explain what the planned t-tests are exactly for increased specificity (e.g., two-tailed between-subjects or independent samples t-test).

Recommended revision for background / study aims sections:

The background and study aims paragraphs are very detailed on reasoning/rationale but the core terms presented in the research questions are not explained before-hand: p-hacking, HARKing (RQ1 - *Lines 6-12, p. 7*).

QRPs are briefly mentioned and this is a key concept (central to your RQs *and* methods) that should be appropriately defined. Why do these QRPs matter (apart from the ones you explain in the Study Aims)? For example, how does HARKing relate to the ethical use of participants' data? The importance of replications (RQ2) could also be mentioned.

I believe this study can have a very high impact which means that it can attract readers from various backgrounds and this is the reason I am recommending this revision.

General comments on S1 protocol content:

Some sentences read as a 'cover letter' and my recommendation would be not to include such observations/statements in the final S1 protocol, or rephrase. Such statements could also be moved to Supplementary Material (e.g. see lines 13-19 on p. 19: "We embarked on a third pilot study that we have since decided not to wait on the results of, as the data collection is out of our control..").

This also applies to lines 35-45, p. 7: a lot of information is presented here that should probably not be part of the final S1 protocol (e.g., "The data collection for Pilot Study C is out of our control.. we have chosen to submit this Stage 1 report..").

The description of the pilot studies is very detailed and it would be great if some of the content could be moved to Supplementary Material (e.g., pilot study A details) but this is only a suggestion, considering the length of the manuscript at Stage 2. If the third pilot study is added at Stage 2 the other pilots could be briefly summarised (key findings and methods development).

Appendix B

Review of RSOS-200048

Our understanding of the paper:

This is Stage 1 Registered Report for a study proposing to characterise English-speaking, psychology research participants' views on 1) select questionable research practices (taken from John et al 2012) and 2) open science practices (data sharing, replication, open methods and open access) online. We feel that the proposed study would be a timely and meaningful contribution to the literature, however, we recommend the following changes which we believe will assist with readability and clarify the details of the Registered Report.

Comments on the structure of the article

Given the non-standard headings in this article we are unsure about which sections of text will be made final in the Stage 1 Registered Report and which sections would be able to be restructured and rewritten. Given this uncertainty we provide comments under the assumption that the majority of what is approved at Stage 1 will need to remain the same at Stage 2.

Major Comments

Background:

We would like to see the authors make a stronger argument for this approach here. Specifically, this would mean restructuring the background section and including more references to cases where non-experts are involved in the planning of research (e.g. as stakeholders or as lay people on ethics boards) as well as examples where non-expert opinions can influence policymaking/practices/cultures. Currently, we feel that this section jumps around a bit and was quite difficult to follow. We suggest the following changes to highlight the areas we found disjointed:

We recommend moving the section about non-scientist stakeholders with examples from medicine (starting page 2, line 53) below the section about how this would look in psychology (page 3, lines 10-30) it feels like it breaks up the argument the authors are making in its current location.

Page 4, line 8 – and perhaps consider whether additional details of how the study will be undertaken should be included in the plan language statement and consent forms of psychological research.

Consider moving the paragraph starting page 4 line 19 to the end of the introduction, just before the concluding sentence, page 4 line 11.

Given that RSOS is a multidisciplinary journal, we would like to see the article written more inclusively for the other fields of science which work with human subjects. While we can see health-related research has been already been used to contextualise some arguments (Rowhani-Farid 2017, Mello NEJM paper), we would like to see continued drawing on the research from more disciplines for references, reducing the use of psychology-specific jargon (e.g. mTurk, HIT rate etc) and extending the arguments about the utility of this research to include other disciplines. This is not to say that the authors should overstate their results, we would like the authors to retain their statement about the specific population being tested.

We also feel that this is an article that participants might want to read, and that the authors should consider making it as accessible as possible to a non-expert audience and try not to assume any important knowledge (e.g. what the replicability crisis is and how open science interventions combat it, what the NIH is, what questionable research practices are).

References: We are concerned that several of the references in this article are to unstable websites. For example, we were not able to access the Bastian blog post with the link that was provided. We would like to ensure that someone in 10 years time could access the references exactly as the authors read them which will not be the case for NIH, Open Science badges, patient groups, and AllTrials references. We suggest that the authors either find stable references for these things or (if possible given copyright permissions) take a copy of these resources and add them to the supplementary materials/post them on to OSF.

We would also like to see the authors consider referencing the other social science articles about questionable research practices for completeness [1,2] and bear in mind that other fields are less familiar with this than psychology and given this is a multidisciplinary journal give some basic multidisciplinary references for the replicability crisis e.g. [3–5]

Study Aims:

These were clear and well written. However, we feel that the study implicitly also aims to understand the impact of how these questions were phrased and we would like to see that acknowledged in the study aims. Based on the use of t-tests and the text of the background it seems that there is an implied hypothesis that posing the questions motively will result in a more extreme response from participants. We would like to see it included in the study aims section.

Scope:

This section is really well done

Pilot Studies:

The use of pilot studies to develop these methods is very thorough and impressive, however we feel that they are currently difficult to digest. We propose that the authors write them up fully as studies within the manuscript, complete with full methods (including for each a table of the questions included similar to Table 1) and the results. We recommend that the description of how and why things were changed between pilot studies is accord its own separate section. As it stands, the description of the methods of these sections is confused by the description of what has changed between the different pilot studies.

We would like the authors to include uncertainty in their figures, ideally using confidence intervals

Proposed Registered Report Study method:

We would like to see more clarity about what incentives are being offered to the participants. At first we assumed that there would be none, but presumably the mTurk participants will be paid a per question rate. (A friendly sidenote here that one of the reviewers is not familiar with mTurk.)

We would like to see a table of the final questions – similar to what is included in Table 1 but without results.

We wonder how the authors will prevent repeat sampling of the same participants who might be involved in multiple base studies.

Proposed Registered Report Study Analysis:

More detail is required in the analysis section of this registered report, specifically:

What type of t tests are these, and exactly what data do they take (we presume that it will not be broken into the three categories that percentages will be calculated for but this should be clear). What alpha level are they going to use and are they going to correct for multiple comparisons?

Which confidence interval calculation will be conducted. It appears from the authors code that they plan to use Wald confidence intervals which are known to provide problematic confidence intervals when proportions are close to 0 or 1, which pilot data indicates will be the case (negative open methods and fraud questions). Further, the data described is multinomial so the confidence intervals should be constructed accordingly. The authors should consider the R package DescTools, MultinomCI function.

Power analysis: The authors should also re-do the power analysis using the new multinomial confidence interval method. They should also conduct a power analysis for results from the t-test to ensure that the sample size is sufficient for these comparisons.

Minor Comments

The authors refer to the NIH guidelines but presumably most psychology research isn't funded by the NIH. The authors should make it clear that other agencies don't have a policy on this if that is the case, or mention their policies.

In the Stage 2 report we would like to see details of all base studies (i.e. base-study purpose and design, study names, IRB approvals, any published papers/pre-prints, pre-registrations etcetera) and a discussion of how the different elements of these articles might influence participants' responses to the questionnaire.

The target sample size is very specific and it's unclear whether they will aim to get this exact number of participants or aim to get a larger number knowing that a reasonably high proportion of their data will be removed subject to their exclusion criteria. We would like clarity on the authors' approach to this stopping rule.

It appears that the authors haven't provided access to the final materials for the Registered Report study, only for the pilot studies. The authors should add these

Could the authors clarify what the attention/comprehension check is in this instance?

We don't think that getting participants' views constitutes checks and balances unless it comes with some kind of regulatory mechanism, the authors should consider reframing this. (lines 47-52)

Page 22, line 2 – does this mean variance between respondents of different base studies?

On sample selection criteria please clarify “and be able to answer our meta-study survey” and describe what a HIT rate is

Have the authors considered capturing other demographics variables previously captured by prior literature? (e.g. political leaning ala Pickett & Roche)

28th April 2020

Editor, *Royal Society Open Science*

Dear Editor,

Enclosed is a revision of our Stage I registered report manuscript entitled “What Do Participants Think of Our Research Practices? An Examination of Behavioral Psychology Participants’ Preferences.” We would like to thank you and the 5 reviewers for their thoughtful and detailed feedback. We have revised the paper to address this feedback as completely as possible.

Broadly, our changes consist of (1) moving the pilot studies to the supplement and adding a description of results for our third pilot; (2) changing our planned analyses to be simpler and more appropriate to the descriptive question we are pursuing; (3) revising the introduction to make it clearer and remove or explain field-specific words; (4) making available on OSF all pilot and proposed study materials, and code for planned and pilot analyses. We have also added an author (Mijke Rhemtulla) who is helping us with the statistical analyses, among other things.

You can find a point-by-point summary of our changes to the manuscript in response to each item raised by you and by each reviewer below.

Best Regards,

Julia G. Bottesini, Mijke Rhemtulla, & Simine Vazire

Associate Editor Comments to Author (Professor Chris Chambers):

Comments to the Author:

Five reviewers have now assessed the Stage 1 manuscript. The reviews are very positive overall and also very detailed and constructive.

Key issues to address in revision include:

[E.1] clarification of the survey questions (reviewers 3 and 4),

We have now provided full materials for our proposed study (<https://osf.io/p8n9w/>). We apologize for not having included these materials with our submission, and hope that this will clarify what questions we plan to ask of participants.

[E.2] suitability of the attention check (reviewers 2 and 3),

Similarly, we hope that any misgivings about the suitability of the data quality check can be cleared up with the addition of full materials for the proposed study. We have used simple, open-ended data quality checks in the past, including in Pilot A, and believe this to be enough to exclude those who don't understand the questions being asked.

[E.3] more details about the planned analyses including t-tests (reviewers 2, 3 and 5),

After considering the reviewers' comments and consulting with a statistical expert (who is now a co-author on the manuscript), we have decided that t-tests and means were indeed not appropriate to answer the questions we are asking. Our research questions are descriptive, and therefore, we feel that it would not be appropriate to use inferential statistics for our planned analyses. We have simplified our analyses to be descriptive (providing a percentage or proportion with a corresponding confidence interval). For questions that have two versions, we have decided not to conduct a statistical test comparing the two versions, but instead to provide readers with proportions and confidence intervals for each of the versions separately as well as combined. In other words, the two versions are not meant to test a theoretical question, but to provide a robustness check to see how consistent our results are across these variations.

[E.4] clarity and structure of the introduction (reviewers 4 and 5),

We have restructured the introduction and cleared up the meaning of some field-specific words. We hope this will make it clearer.

[E.5] as well as a range other concerns including sample validity (reviewers 4),

Although we have practical constraints that only allow us to directly recruit paid participants from Amazon Mechanical Turk (i.e., we have research funds parked there), we have updated our study inclusion criteria to allow for the incidental inclusion of participants from other online participant pools, if partner researchers are running studies on these platforms. However, given the large proportion of past and current research that still takes place on MTurk, we believe this is the most important sample of online research participants and will allow our findings to generalize to many other samples of psychology study participants.

[E.6] suggestions to minimise jargon (an excellent point given the broad remit of RSOS; reviewer 5), clarification of various terms and definitions (reviewer 2) as well as

We have made edits and added footnotes where appropriate to address these issues. We hope this will make the manuscript more readable and accessible to researchers of all areas and backgrounds.

[E.7] clarification of methodological details (raised by all reviewers).

We have expanded our method section to be more specific regarding how the study will be conducted. We have also added some information to the supplementary materials, and uploaded all the materials, code, dummy data, and proposed analyses, which we hope will clarify the method further.

[E.8] Concerning the recommendation of reviewer 3 to shift details of the pilot studies to the supplementary information, I will leave this to the discretion of the authors since either approach can work for a RR.

We thank the editor for this flexibility and reviewer 3 for this suggestion. We have now moved all pilot studies to the supplement, and include instead a brief description of the overall results in the main manuscript. We feel that this makes the manuscript more streamlined without compromising transparency, as all details that were previously in the main manuscript have been moved to the supplement. This also allowed us to describe the results of Pilot C using partial data, which we hope will be useful to readers in understanding the changes we made to the survey questions for the proposed study. We do not plan on updating these results even if more data is collected on this study (we are not in control of data collection for this project, so we do not know if more data will be collected). Moving details of the pilot studies to the supplement also addresses Reviewer 2's very reasonable suggestion of removing some of the sentences describing the reasons behind our decision not to wait for Pilot C data before submitting this

Stage I manuscript. Finally, this change requires us to more fully describe the meta-study questions instead of describing how they changed compared to the pilots, which we believe makes the manuscript clearer and easier to follow. Overall, we hope you will agree that the manuscript is much improved thanks to these changes.

Reviewer comments to Author:

Reviewer 1

[R1.1.] This Stage 1 manuscript seeks to examine research participants' perspectives of research practices. I read the article with great interest because I agree with the authors' assessment that we should at least poll the stakeholders involved in our research enterprise to get their opinions. In a very real sense, I view them as partial owners of the data that they produce (whether it be via their own efforts, their tax payer dollars, or tuition dollars). The same goes with MTurk participants. In this review system, we're asked to comment on a variety of different domains, which I will do below. I was positively disposed to the manuscript and I think the results would be informative to the field (so an in principle acceptance would be a reasonable decision from my perspective).

The analysis was also really reasonable because it's a descriptive account of participants' evaluations. From my perspective, there might not be that much ambiguity about how to analyze the data. Plus, what ambiguity that did exist (particularly related to question wording and selection) was discussed at length and even partially tested in these pilot studies. The question holds much scientific validity because it speaks to the reflections that stakeholders have on the research process and is of natural interest to modern researchers.

[R1.2.] The logic and rationale of the project were laid out very well and carefully. The inclusion of pilot studies was particularly beneficial because it settled some of the ambiguity in how participants might respond to questions depending on how they were worded.

[R1.3] The soundness and feasibility were major strengths. The sample size for the proposed registered study was healthy and well-justified. Again, the study is sound and reasonable partially because the pilot studies approximate what will be done on a larger scale.

We thank the reviewer for this positive feedback.

[R1.4] The clarity/degree of detail is sufficient to replicate the study (and likely the analysis pipeline). I say this because the materials were presented very clearly in the manuscript and online supplementary materials. **Regarding some things to improve for the manuscript, the authors could clarify how any comparisons between MTurkers and undergrads are made.** Even if it's just "eye-balling" and reporting it separately by sample and through a formal comparison (e.g., t-tests or chi-square tests). Also, in their analysis plan (I think this went without saying), I got the sense that the **positive or negative items would be reverse scored so that t-tests would be interpretable** (e.g., to compare if the responses differed based on the question). If this is the case, **it might be good to say that.** If not, just clarify that no recoding of the variables will be necessary. **Reporting of effect sizes is also preferred** (again, likely goes without saying). The proposed study is really good, so these minor suggestions are the best I can come up with!

We have now addressed all three of these issues in the manuscript: (1) we have removed t-tests from our planned analyses and clarified that the exploratory analysis between MTurkers and undergrads will be conducted using a chi-square test, (2) we have explained which items will be reverse-scored, and (3) our planned analyses are also available as an R script (see <https://osf.io/ytdek/>)

[R1.5] The likelihood of undisclosed flexibility is low. I say this because of the study's simplicity—the questions will be tacked onto some existing studies and the analysis/reporting is pretty straightforward.

We thank the reviewer for this positive feedback.

[R1.6] We're asked if the authors considered a sufficient number of outcome-neutral conditions such that the results indeed test the hypotheses. The design here is a bit different (and there aren't hypotheses; although, honestly, it wouldn't be hard to make some based on the pilot studies...). Nevertheless, the authors managed to do even this given the design. They selectively varied the way various questions were worded (e.g., positive/negative, motives/neutral) so that the results will be informative. A superficial read of this final paper might lay the challenge that the items were worded in suggestive ways or that the authors were "in the bag" for convincing students/Turkers that these are good things. Their neutral approach and varying of the stimuli help defend against the criticism, too.

We thank the reviewer for this positive feedback.

Reviewer 2 [attached pdf Review_RSOS-200048_LT]

General evaluation of the S1 RR criteria:

[R2.1] Scientific validity of the research question(s).

The research questions of the proposed S1 protocol are not only scientifically valid but also of great importance. Research in psychological science (and other disciplines) is supposed to benefit the public (i.e., the collective good) and yet a lot of the research practices that have been identified as 'questionable' may actually have the opposite effect. This study may provide much-needed evidence to support the claim that these practices are not considered acceptable by participants, without whom the majority of research would not be possible. I believe that this study has both ethical and practical implications (e.g., policy change).

[R2.2] Logic, rationale, and plausibility of the proposed hypotheses.

Although the proposed study does not have specific confirmatory hypotheses, the rationale for the research questions is clearly explained. Any degrees of freedom regarding the researchers' predictions and analyses at Stage 2 have been prevented with the high level of transparency adopted throughout the manuscript (but see minor comment regarding the analyses).

We thank the reviewer for this positive feedback.

[R2.3] Soundness and feasibility of the methodology and analysis pipeline.

The soundness and feasibility of the methodology have been demonstrated in the pilot studies described in the S1 protocol and the proposed methods. **A suggestion would be for the 'participants' and 'study selection' sections to be accompanied by a schematic (figure)** to make it easier for the reader to follow all the described criteria.

We considered this suggestion but felt that the criteria for sample and study selection are clear enough in the manuscript, and that adding a schematic might make it more confusing.

[R2.4] Clarity and degree of methodological detail for replication of the proposed experimental procedures and analysis pipeline.

All procedures are described in sufficient detail to allow for direct replication (only a few minor comments).

[R2.5] Sufficiently clear and detailed description of the methods to prevent undisclosed flexibility in the experimental procedures or analysis pipeline

The authors have provided sufficiently clear and detailed descriptions of the methods and analyses, but a few minor comments could be addressed for increased specificity.

We thank the reviewer for this positive feedback.

[R2.6] Sufficient outcome-neutral conditions for ensuring that the results obtained are able to test the stated hypotheses.

An attention check has been included, but I am recommending that **more details regarding this survey question are provided in the S1 protocol**. The authors also use an inclusion criterion for the recruitment of mTurk participants that should minimise the possibility of dishonest/invalid survey responses.

We have now provided full materials (<https://osf.io/p8n9w/>) for the proposed study, which includes wording for all the questions, demographics, and attention checks. We hope this makes the suitability of the attention check clearer.

Minor comments:

[R2.7] There appears to be an issue with some references (e.g., line 19 p. 3) linking to a Zotero page with this message (“You probably got here by mistake”).

We have now fixed this.

[R2.8] Line 20, p. 3 : Please provide an example or definition of ‘top’ psychology journals (e.g., high impact factor).

We have addressed this in the manuscript.

[R2.9] Line 33, p. 3 : Rephrase to avoid repetition of ‘research practices’ in the same sentence and present abbreviation last not first: e.g. “engaging in questionable research practices, or QRPS (as described in John et al. 2012) that can inflate ..”.

We have addressed this in the manuscript.

[R2.10] Lines 40-54, p. 6: This paragraph could be integrated into the Methods section.

We carefully considered this change, and we see pros and cons of integrating this material into the Method section, but have decided against it.

[R2.11] Pilot B figures : The titles are not very self-explanatory in the current format - as I am guessing similar figures on QRPs may be produced after data collection (e.g., “Figure 3.

HARKing.”). Increased specificity may also be needed for the abbreviations used in the figures- e.g., in Figure 4 you could mention in the note what ‘Rs’ refers to even though it may seem obvious. Correct “fell strongly” in Fig. 4.

We have addressed this in the supplementary materials, where detailed descriptions of the pilots are now located.

[R2.12] Sections: The subsections of the pilots should be consistent (e.g., see “Fraud (n = 114)”). Sample size is not indicated in other subsections while pilot A does not have any subheadings for methods/results. Following the ‘fraud’ subsection the summary is presented with no subheading which makes it seem like they are part of the same section.

This is mainly because Pilot A addresses only one practice while Pilot B addresses eight, so we felt more subheadings would make the text more organized. Either way, we have now moved the pilots to the supplement, which we hope will make the headings in the main manuscript more consistent.

[R2.13] Line 38, p. 20 : Include a footnote to explain what HIT stands for.

We have addressed this in the manuscript.

[R2.14] Lines 48-53, p. 22 : “To err on the side of caution” is not very clear. I might have missed this if it is included in OSF materials, but what is the actual open ended attention/comprehension check question? More details should be provided here for increased reproducibility.

We have now included the full survey instrument on the OSF page (<https://osf.io/p8n9w/>), which we hope will help clarify the comprehension check. We chose to include rather than exclude data from participants when the attention check is not clearly invalid.

[R2.15] Missing data [question]: Is the attention check question enough to catch invalid responses? If the scales default at the middle, can you tell if a response is missed or valid (i.e., do participants get a warning if they haven’t interacted with the question at all)? If missed responses are possible, how will you handle missing data (e.g., excluding participants)?

The questions are written such that if the participant doesn’t move the slider, it will not register an answer, so there is no default answer. Moreover, due to the way the questions are designed, we won’t have any missing data for the main questions, only for the demographic questions, where selecting “prefer not to say” is an option for any personal information. Although no method is 100% guaranteed to catch

inattentive respondents, we have used a similar attention check in the past to make sure participants were at least capable of comprehending the questions being asked and formulate a response, so we feel comfortable saying this is enough to catch most, if not all, invalid responses.

[R2.16] Analyses: Please explain what the planned t-tests are exactly for increased specificity (e.g., two-tailed between-subjects or independent samples t-test).

See comments to the editor in [E.3] about planned analyses.

Recommended revision for background / study aims sections:

[R2.17] The background and study aims paragraphs are very detailed on reasoning/rationale but the core terms presented in the research questions are not explained before-hand: p-hacking, HARKing (RQ1 - Lines 6-12, p. 7).

We have added more explanation about each term before we present the research questions.

[R2.18] QRPs are briefly mentioned and this is a key concept (central to your RQs and methods) that should be appropriately defined. Why do these QRPs matter (apart from the ones you explain in the Study Aims)? For example, how does HARKing relate to the ethical use of participants' data? The importance of replications (RQ2) could also be mentioned.

We have added a discussion of QRPs and clearer definitions of the QRPs included in our study. We have also added more examples of how these practices may be relevant to participants.

I believe this study can have a very high impact which means that it can attract readers from various backgrounds and this is the reason I am recommending this revision.

General comments on S1 protocol content:

[R2.19] Some sentences read as a 'cover letter' and my recommendation would be not to include such observations/statements in the final S1 protocol, or rephrase. Such statements could also be moved to Supplementary Material (e.g. see lines 13-19 on p. 19: "We embarked on a third pilot study that we have since decided not to wait on the results of, as the data collection is out of our control..").

This also applies to lines 35-45, p. 7: a lot of information is presented here that should probably not be part of the final S1 protocol (e.g., "The data collection for Pilot Study C is out of our control.. we have chosen to submit this Stage 1 report..").

We have moved Pilot C to the supplement and added results based on the data we have (more data might be collected by the research team conducting the base study, but this is out of our control). We believe this addresses this point.

[R2.20] The description of the pilot studies is very detailed and it would be great if some of the content could be moved to Supplementary Material (e.g., pilot study A details) but this is only a suggestion, considering the length of the manuscript at Stage 2. If the third pilot study is added at Stage 2 the other pilots could be briefly summarised (key findings and methods development).

We have addressed all of these issues by moving all pilot studies to the supplementary materials. Thank you for this suggestion!

Reviewer 3

THE TEXT BELOW IS INCLUDED IN AN ATTACHED FILE WHICH IS FORMATTED TO BE EASIER TO READ.

The proposed experiment is very well-documented, has been refined through pilot studies, highlights several open practices the authors have taken, and would make a clear, reproducible, and important contribution to the scientific literature. It is among the most well-documented and clear protocols I have read. I recommend that it receives in-principle acceptance after minor revisions.

I have one substantial comment regarding the research questions and one substantial comment regarding the wording of a few questions. **The remainder of my comments are minor and scattered throughout the attached pdf in comment boxes.** I leave it to the authors' judgement of whether or not to respond to some of the minor comments. Below are the specific questions I was asked to explicitly respond to.

Please comment explicitly on each of the following points in your comments to the authors:

The scientific validity of the research question(s)

[R3.1] The two itemized research questions come across as more general than the questions the authors appear to be asking based on a reading of the proposal in its entirety. The itemized research questions ask: What are participants' opinions about (1) QRPs and fraud, and (2) open science practices. The specific questions that participants will be asked, however, center on whether participants support or oppose QRPs, fraud, and open practices. Specifically, related to their own data and the data from studies

they've participated in. If the RQs were mainly interested in 'opinions', then open text boxes would be useful. However, it appears to me that this isn't the main purpose of the study.

Refining the RQs to be more specific could help refine some of the questions or help justify the current wording of the questions. For example, Question 6 asks if participants feel that researchers should or should not share their materials and procedures. This question doesn't ask to weigh the costs of sharing (which some might argue is a lot of time). It's unclear whether the RQs aim to discover what participants want to be done with their data (ideals) or what they think is reasonable to do with their data (practicality), or both. More specificity could clear this up. There are a few comment boxes in the pdf touching on these points.

We have carefully considered the reviewer's points and made attempts to revise the language used to describe our research questions. However, after trying several different alternatives and checking with colleagues to see what their impressions are of the implied goals of different ways of framing the research question, we ended up keeping the research questions more or less the same. We have made a few small changes (removed the word "attitudes" and changed "opinions" to "views"), but overall we must admit that the research questions have not changed much. While we agree with the reviewer that there is some ambiguity in what we mean by "research participants' views" or "preferences", we could not think of a succinct way to be more specific. We don't think that these phrases are misleading, and we hope the interested reader will continue on to our methods and materials to see exactly what we mean.

With respect to asking participants to consider the tradeoffs vs. not explicitly asking them to do so, as we now explain in the manuscript, we opted not to do so because we would like our findings to generalize to other similar participants. Thus, obtaining participants' spontaneous views on these matters is likely the most ecologically valid way to infer how other participants in similar studies likely feel. We have now clarified this throughout the manuscript.

I think the overarching research question is valid and important.

The logic, rationale, and plausibility of the proposed hypotheses

The authors do not propose hypotheses. They do propose two research questions (see previous answer).

The soundness and feasibility of the methodology and analysis pipeline (including statistical power analysis where applicable)

The methodology is sound and feasible. The authors refined it over multiple pilot studies. The main analyses are sound and feasible and supported by a reasonable precision analysis.

We thank the reviewer for this positive feedback.

[R3.2] I recommend the authors include more information about the t-tests they will perform, including their inferential criteria, power, smallest effect size of interest, multiple comparisons, and conclusions they expect to be able to draw based on potential outcomes. Are they testing for differences, equivalence, or both?

We have addressed the issues with our planned analyses (see comments to the editor in [E.3]).

Whether the clarity and degree of methodological detail would be sufficient to replicate exactly the proposed experimental procedures and analysis pipeline

[R3.3] The authors provide all the materials and data from their pilot studies on the osf, as well as appropriate links throughout their proposal. I could not identify where the questionnaire used for pilot C is located. Presumably this file could be shared at this point in time. I also did not locate the questionnaire that will be used for the RR. Although the authors lucidly explain how they changed the questions, it would be useful to see the exact questions as they will be presented to participants.

We have added materials for all pilots as well as the proposed study to our OSF project page (<https://osf.io/bgpyc/>).

Whether the authors provide a sufficiently clear and detailed description of the methods to prevent undisclosed flexibility in the experimental procedures or analysis pipeline

[R3.4] The manuscript is written with good specificity as to prevent undisclosed flexibility. They clearly identify that some collected data will only be used in exploratory analyses. **In the pdf comments**, I identify a few minor tweaks that could be made to increase specificity (reduce the potential for flexibility).

We have made tweaks in response to these helpful suggestions.

Whether the authors have considered sufficient outcome-neutral conditions (e.g. positive controls) for ensuring that the results obtained are able to test the stated hypotheses

[R3.5] The authors do not propose hypotheses. **The questionnaire includes a question on fraud which the authors propose could be used as a positive control. I recommend the authors provide more detail regarding this point.** They also use an **attention/comprehension check**, which I recommend the authors explain in more detail. The methods are sufficient to inform the overarching research question in its current form. However, as stated above, I recommend the authors refine the research question to more closely match what they (appear) to be proposing to answer.

We now realize that our statements about using the responses to the fraud item as a positive control was too vague, and we have removed it.

We have included the full materials which will hopefully clarify the attention check. We have used similar checks for data quality in the past and that seemed to produce the desired result (see also our response to [E.2] and [R2.15] above).

Thank you for the opportunity to review this manuscript. I look forward to reading the results when they come in.

Robert Thibault

Reviewer 4

Comments to the Author(s)

I like the proposed project and think it tackles an interesting and important question. I think the authors make a strong case for why we should know what participants think about our research practices and how this might fall within the requirements of informed consent, broadly defined. And I also believe that the justification for focusing on minimal risk studies with convenience samples is quite sound, with constraints on generalizability. (Surprising results from pilot study A. So many MTurk workers want their data shared for verification and other analyses. Amazing!) My suggestions for improvement/clarification follow:

Methods & Analyses

[R4.1] Questions 3, 5, 6, 7, and 8, which have had at least minor changes made to their wordings weren't presented for me to adequately examine them. My comments below are based on assumptions I made (and which I make clear) regarding the wording of the questions.

We apologize for not including the materials with the submission. We have now uploaded the full set of materials at <https://osf.io/p8n9w/>

[R4.2] Questions 5, Page 21, lines 26-28: I assume the question is “researchers should attempt to replicate a finding before publishing it or trust that the finding is solid and move on to a new project”. I think it has the same problem as in Pilot B (in which some participants gave what seemed to be contradictory answers to the two versions of the question) but now the problem is presented within the question. I think the question can distort responses since both positions can be held at the same time. E.g., I don't think every finding needs to be replicated before being published but I also don't think researchers should trust it is solid and move on. You can publish a finding without trusting that it is solid. And sometimes this can make science move forward, as long as independent replications (both negative and positive) are published. I agree that giving participants context (e.g., that there is a tradeoff) in this case is necessary. My main concern is that in trying to give context within the question the question has become very difficult to answer precisely and thus very difficult to interpret its responses, particularly because the tradeoff isn't clear (at least to me). **The question is a kind of double-barreled question** (the second part not only asks whether authors should publish without replicating but also about whether they should move on after trusting it is solid). I think the authors should make clearer what the tradeoff is, be clear that it is a tradeoff rather than two positions that can be both held simultaneously, and then reword the question. Perhaps even providing the context in a sentence before asking the question. I'm really not sure about this one and it very much depends on what the authors want an answer to. What precisely is the tradeoff that the authors are getting at and is it actually a tradeoff? Maybe the authors want to know, “should researchers make sure their finding is solid before publishing it, or is it ok to publish something you aren't sure is solid yet”? Or is the tradeoff they want to get at better described with a statement such as, “part of establishing whether a finding is solid is by replicating it in another study to see if you get the same results”, and then asking “should researchers replicate their finding in a new study before publishing or is it ok to publish the finding of one study without replicating”?

We have reworded Question 5, about replication, to address this point. The full materials, which include the new wording, can be found at <https://osf.io/p8n9w/>

[R4.3] Question 7, Pages 21-22: I assume the wording of the question is given in the first sentence of the paragraph (i.e., “...where the article reporting the results of the base study should be published: an open access journal vs a paywalled journal”). If the intention is to give a clearer picture of what happens to the funds and who will be accessing the article, then perhaps more explicit wording is required. E.g., “the article reporting the results should be published in an open access journal that gives free access to anyone who wants to read the article vs a for-profit paywalled journal that charges a fee” (note that this is only a suggestion). The concern with the question as presented in Pilot B, that participants might have thought the authors are compensated and that it isn’t the publisher that makes the money, isn’t quite addressed in this case (i.e., where do the funds at the paywalled journal go? Do they pay the author or does it go to funding more research?). There’s still a lot left to assumptions of the reader.

We have carefully considered these points, but believe that clarifying the assumptions even further would add too much text to the question, and might present other issues we haven’t thought of, so we have opted to not change the wording in this case.

[R4.4] Analyses: Given that some of the data are likely to be highly skewed (e.g., Pilot B, QRPs), the means and standard deviations seem less informative to me (although this is mostly offset by presenting the results in more detail, as the authors did for Pilot B and they propose to do for the main study, by presenting percentage of participants finding the practices (un)acceptable). **I recommend also providing the median.** For the main study, the authors propose comparing the two versions of each question (where relevant) by using t-tests but if the data are highly skewed then perhaps a non-parametric test is better. I also wondered why they aren’t proposing a chi-square test comparing percentage of participants finding the practices (un)acceptable, between the two methods of asking each question.

We have updated our planned analyses to address these issues. See comments to the editor in [E.3].

[R4.5] Page 21, lines 18-21, questions 5 and 7. If the worry is that participants might think one end of the numbering scale might be less desirable vs more desirable, then perhaps presenting these labelled options in a nonlinear way might be best. E.g., rather than from left to right (which participants might inadvertently perceive as going from less desirable to more desirable (given that the questions preceding it are presumably going to be presented in this way), the options for questions 5 and 7 might be presented vertically or via a dropdown box, or in whatever way the authors prefer. This is just a suggestion.

We considered this option, but were worried that it might introduce new problems, given that it's an unusual way to present questions, and differs significantly from the answer format for the other questions.

[R4.6] Page 22, question 8: I can't see the wording of the question nor the reasons presented to participants.

We have now uploaded all materials for the proposed study, including the survey with the specific wording of all questions, at <https://osf.io/bgpvc/>.

[R4.7] Page 22: data exclusion attention/comprehension check: what is the question? Also, (I may have missed it but) the authors should be clear **if excluding participants who took part in the pilot studies a and b from taking part in main study.**

Please see our response to [E.2] and [R2.15] above regarding the attention check.

We have added a statement to the supplement explaining how we will address repeated sampling of participants.

[R4.8] Participants: Is there a reason the authors are not including any Prolific participants? If the worry is that MTurk workers are different from prolific workers, then generalizability beyond MTurk will be an issue (and should be stated). In the same way that the authors propose to collect data from student participants at different universities, **I would suggest that online participants from different platforms should not be a problem.** Prolific allows prespecifying that participants be from the US. Of course, I'm not suggesting that there needs to be equal number of MTurk and Prolific participants, but merely that the authors may want to consider also including studies being run on Prolific (just a suggestion).

Thank you for this suggestion. Although the funds we have access to are set aside for use on Amazon Mechanical Turk (i.e., we have already bought credit on MTurk using funds that were going to otherwise expire), we have updated our study inclusion criteria to allow us flexibility to include base studies run on Prolific or other online subject pools when the researchers running the base study will be collecting the data, so that we have a chance to diversify our sample.

Issues of clarity/argumentation in the introduction:

[R4.9] Page 4, lines 29-31, "going against participants' wishes could undermine public trust in science even among non-participants, if they feel we are mistreating participants": I don't follow the logic of the argument that leads to the final sentence. The paragraph is about participants potentially being driven away from participating due to their

preferences being ignored. So how does it follow that the public trust in science would be diminished if they perceived that we mistreat participants? Is ignoring their preferences for data sharing, or using their data to p-hack, going to be perceived as mistreatment? I think **the final sentence of the paragraph requires more reasoning** but it would be a digression from the point of the paper without adding much added value.

We appreciate the reviewer's feedback, but upon careful reflection we feel that the paragraph is as clear as we can make it.

[R4.10] Page 5 lines 20-25: The arguments leading up to this paragraph have been (1) researchers are not necessarily incentivized to produce good replicable work and so this may have created norms and practices that are not optimal for science, and getting participants' views may be one way for checks and balances against this, and (2) participants should be informed about current research practices, potentially as part of informed consent, and thus their views are important because then we can see what their perceptions are. **When the paragraph speaks of blind spots, I found it confusing.** On my second reading I realized the blind spots are with regards to point (1). So maybe something to make this clearer in the paragraph will benefit readers.

We have addressed this issue in the manuscript.

Minor points

[R4.11] Page 5 line 19, the paragraph starts, "to solve this inconsistency". This statement is ambiguous. Is it referring to "mismatch between their expectations and the reality of research"? If so, then the ambiguity can be cleared up with something like, "If we do find that there is an inconsistency between participants' expectations and research practices, scientists have several options".

We have addressed this issue by rewriting the phrase as suggested by the reviewer.

[R4.11] Page 4, lines 26-29: not only might participants stop from participating, but they may continue to participate but lose the motivation to provide high quality data.

Good point! We've added to our paragraph to include this possibility.

[R4.11] Page 17 lines 32-36: although 89% is quite low for finding scientific fraud unacceptable, and may be due to nonserious reporting, it is not too different from some of

the results from Picket and Roche who found 91% reported fraud as unacceptable (Study 2), though their MTurk sample had 96% finding it unacceptable.

Interesting note (Page 18, lines 31-32): Picket and Roche's findings on selective reporting are very consistent with Pilot B results on all QRPs. The finding from Pilot B suggests that over 75% found all QRPs unacceptable; 71% of Picket and Roche's large MTurk sample considered selective reporting unacceptable, though they used the binary yes/no response options.

We thank the reviewer for this valuable information!

[R4.12] Study selection, page 20, lines 31-33: what is the time-frame of the study? Perhaps this should be prespecified.

We are reluctant to specify a timeframe, especially now that we have experienced the (first wave of?) the coronavirus pandemic.

I always sign my reviews,

Farid Anvari

Reviewer 5

Comments to the Author(s)

Full comments are included in the attached document but we respond briefly to the specific RSOS questions here

The scientific validity of the research question(s):

This is Stage 1 Registered Report for a study proposing to characterise English-speaking, psychology research participants' views on 1) select questionable research practices (taken from John et al 2012) and 2) open science practices (data sharing, replication, open methods and open access) online. We feel that the proposed study would be a timely and meaningful contribution to the literature

The logic, rationale, and plausibility of the proposed hypotheses:

[R5.1] The study is primarily descriptive and doesn't offer any specific hypotheses. **However, we ask that the authors add hypotheses regarding the t-tests included in their analysis plan.**

We have modified our planned analyses to be more descriptive (thereby removing t-tests and other inferential statistics) which is more in line with the question we are attempting to answer in this study (see also our response to [E.3]).

The soundness and feasibility of the methodology and analysis pipeline (including statistical power analysis where applicable)

[R5.2] The overall plan appears sound however we would like to **see more description about how the t-tests will be conducted.** The authors should provide a power analysis for these t-tests in addition to the power analysis currently included. We are also concerned that the authors' choice of confidence intervals may not be suitable for multinomial proportions and have suggested an alternative R package that might work better.

We have changed our analyses to include the (much more appropriate) multinomial confidence intervals, and believe this is a better way to communicate the uncertainty inherent in our estimates. Thank you for this very helpful suggestion.

Whether the clarity and degree of methodological detail would be sufficient to replicate exactly the proposed experimental procedures and analysis pipeline

[R5.3] At this point the authors appear to have provided **materials for their pilot studies but not for the Registered Report study.** These should be made available and more information should be provided about the statistical analyses.

We have provided these materials and code for all statistical analyses performed or planned in our OSF page (<https://osf.io/bgpyc/>).

Whether the authors provide a sufficiently clear and detailed description of the methods to prevent undisclosed flexibility in the experimental procedures or analysis pipeline

If the authors provide the requested information about the materials and t-test and confidence interval calculations there will be no room for flexibility in experimental procedures or analyses

Whether the authors have considered sufficient outcome-neutral conditions (e.g. positive controls) for ensuring that the results obtained are able to test the stated hypotheses:

NA

Review Conducted by Hannah Fraser and Daniel Hamilton

Full comments from File “Review of RSOS”

Our understanding of the paper:

This is Stage 1 Registered Report for a study proposing to characterise English-speaking, psychology research participants’ views on 1) select questionable research practices (taken from John et al 2012) and 2) open science practices (data sharing, replication, open methods and open access) online. We feel that the proposed study would be a timely and meaningful contribution to the literature, however, we recommend the following changes which we believe will assist with readability and clarify the details of the Registered Report.

Comments on the structure of the article

Given the non-standard headings in this article we are unsure about which sections of text will be made final in the Stage 1 Registered Report and which sections would be able to be restructured and rewritten. Given this uncertainty we provide comments under the assumption that the majority of what is approved at Stage 1 will need to remain the same at Stage 2.

Major Comments

Background:

[R5.4] We would like to see the authors make a stronger argument for this approach here. Specifically, this would mean **restructuring the background section and including more references to cases where non-experts are involved in the planning of research** (e.g. as stakeholders or as lay people on ethics boards) as well as examples where non-expert opinions can influence policymaking/practices/cultures. Currently, we feel that this section jumps around a bit and was quite difficult to follow. We suggest the following changes to highlight the areas we found disjointed:

We recommend moving the section about non-scientist stakeholders with examples from medicine (starting page 2, line 53) below the section about how this would look in

psychology (page 3, lines 10-30) it feels like it breaks up the argument the authors are making in its current location.

We thank the reviewers for their suggestion but have decided to keep the current structure.

[R5.5] Page 4, line 8 – and perhaps consider whether additional details of how the study will be undertaken should be included in the plan language statement and consent forms of psychological research.

We have added a sentence in the manuscript to address this point.

[R5.6] Consider moving the paragraph starting page 4 line 19 to the end of the introduction, just before the concluding sentence, page 4 line 11.

We thank the reviewers for their suggestion, and have made some edits to the introduction which we hope will also address this point.

[R5.7] Given that RSOS is a multidisciplinary journal, **we would like to see the article written more inclusively for the other fields of science which work with human subjects.** While we can see health-related research has been already been used to contextualise some arguments (Rowhani-Farid 2017, Mello NEJM paper), we would like to see continued drawing on the research from more disciplines for references, reducing the use of psychology-specific jargon (e.g. mTurk, HIT rate etc) and extending the arguments about the utility of this research to include other disciplines. This is not to say that the authors should overstate their results, we would like the authors to retain their statement about the specific population being tested.

We have edited this section and added footnotes and other explanations to make sure the meaning of field-specific words is clearer. We attempted to revise the manuscript to broaden its scope, but given that this is ultimately a study of participants in minimal-risk psychology studies, and we do not want to overclaim, we must admit that we did not find many opportunities to do so.

[R5.8] We also feel that this is an article that participants might want to read, and that the authors should consider **making it as accessible as possible to a non-expert audience** and try not to assume any important knowledge (e.g. what the replicability crisis is and how open science interventions combat it, what the NIH is, what questionable research practices are).

We have made edits to address this point in the introduction. Specifically, we have provided more background about the replication crisis and the proposed reforms, and provided clearer definitions of technical terms.

[R5.9] References: **We are concerned that several of the references in this article are to unstable websites.** For example, we were not able to access the Bastian blog post with the link that was provided. We would like to ensure that someone in 10 years time could access the references exactly as the authors read them which will not be the case for NIH, Open Science badges, patient groups, and AllTrials references. We suggest that the authors either find stable references for these things or (if possible given copyright permissions) take a copy of these resources and add them to the supplementary materials/post them on to OSF.

We tested the links provided and were able to access the Bastian blog as well as the other websites, although clicking directly on the links might have failed because some references were still linked to the reference manager we used, which has now been fixed. Although we agree that it would be important to save as much information as possible, it is unclear what copyright issues there might be related to taking screenshots of these materials and archiving them. We have attempted to provide the most stable link to each of these resources.

[R5.10] We would also like to see the authors **consider referencing the other social science articles about questionable research practices** for completeness [1,2] and bear in mind that other fields are less familiar with this than psychology and given this is a multidisciplinary journal give some basic multidisciplinary references for the replicability crisis e.g. [3-5]

We looked for the references that seem to be related to the numbers included in this comment, but it seems they were not attached to the review. We hope that we have clarified these concepts enough in the introduction to make the article more broadly relevant.

Study Aims:

[R5.11] These were clear and well written. However, we feel that the **study implicitly also aims to understand the impact of how these questions were phrased and we would like to see that acknowledged in the study aims.** Based on the use of t-tests and the text of the background it seems that there is an implied hypothesis that posing the questions

motively will result in a more extreme response from participants. We would like to see it included in the study aims section.

We have rethought our planned analyses to be more consistent with our primary study aim: to describe participant's views about the selected practices. We have detailed the rationale behind including more than one version of some questions, as well as why we don't think the difference between the specific versions we used is interesting (see also our response to [E.3] above).

Scope:

This section is really well done

Pilot Studies:

[R5.12] The use of pilot studies to develop these methods is very thorough and impressive, however we feel that they are currently difficult to digest. We propose that the authors write them up fully as studies within the manuscript, complete with full methods (including for each a table of the questions included similar to Table 1) and the results. We recommend that the description of **how and why things were changed between pilot studies is accord its own separate section**. As it stands, the description of the methods of these sections is confused by the description of what has changed between the different pilot studies.

Following another reviewer's suggestion, we have decided to move the pilot studies to the supplement. We hope you will agree that this makes the manuscript clearer, while allowing us to describe each pilot study in more detail. Furthermore, all materials, data, and analyses for all three pilot studies are available on this project's OSF page (<https://osf.io/bgpyc/>). We believe the combination of all these resources is sufficient to allow readers a more informed understanding of our pilot results and any changes between studies.

[R5.13] We would like the authors to include uncertainty in their figures, ideally using confidence intervals

We have included multinomial confidence intervals around the proportions for our planned analyses, and plan to include these measures of uncertainty in all the figures for the proposed study. However, our pilot studies were meant to test our materials and check whether the study was feasible, rather than provide us with I

estimates of these proportions, so we have not made changes to the pilot study results.

Proposed Registered Report Study method:

[R5.14] We would like to see more clarity about what **incentives are being offered to the participants**. At first we assumed that there would be none, but presumably the mTurk participants will be paid a per question rate. (A friendly sidenote here that one of the reviewers is not familiar with mTurk.)

We have added a statement to the supplement clarifying how MTurk participants who are recruited by the authors will be paid.

[R5.15] We would like to see a table of the final questions – similar to what is included in Table 1 but without results.

We have added a table listing the topics and number of versions for each question, to make it easier for readers to remember which question is about which topic. However, we did not include the full wording of the questions and response options as we felt this would be too much text for a table. However, we have made all materials available in our project page (<https://osf.io/bgpvc/>).

[R5.16] We wonder how the authors will prevent repeat sampling of the same participants who might be involved in multiple base studies.

We have added a statement to the supplement explaining how we will address repeated sampling of participants.

Proposed Registered Report Study Analysis:

More detail is required in the analysis section of this registered report, specifically:

[R5.17] **What type of t tests are these, and exactly what data do they take** (we presume that it will not be broken into the three categories that percentages will be calculated for but this should be clear). **What alpha level are they going to use and are they going to correct for multiple comparisons?**

We have revised our planned analyses and hope you will agree that the new plan addresses this issue and is much more consistent with our objective of describing these distributions rather than testing any specific hypotheses about them.

[R5.18] **Which confidence interval calculation will be conducted.** It appears from the authors code that they plan to use Wald confidence intervals which are known to provide problematic confidence intervals when proportions are close to 0 or 1, which pilot data indicates will be the case (negative open methods and fraud questions). Further, the data described is multinomial so the confidence intervals should be constructed accordingly. The authors should consider the R package DescTools, MultinomCI function.

Power analysis: **The authors should also re-do the power analysis using the new multinomial confidence interval method.** They should also conduct a power analysis for results from the t- test to ensure that the sample size is sufficient for these comparisons.

Thank you for these very helpful suggestions! We were indeed using the more problematic Wald confidence intervals, but have updated both our precision analysis and our planned analyses using the MultinomCI function in DescTools with the Sison-Glaz method. The code for these analyses can be found at <https://osf.io/bgpyc/>

Minor Comments

[R5.19] The authors refer to the NIH guidelines but presumably most psychology research isn't funded by the NIH. The authors should make it clear that other agencies don't have a policy on this if that is the case, or mention their policies.

We appreciate the reviewer's suggestion but we have decided to leave this as is.

[R5.20] In the Stage 2 report we would like to see details of all base studies (i.e. base-study purpose and design, study names, IRB approvals, any published papers/pre-prints, pre-registrations etcetera) and a discussion of how the different elements of these articles might influence participants' responses to the questionnaire.

We agree with the reviewer that having all of this information about the base studies, their results and publications, etc., would be interesting and could lead to fruitful exploratory analyses. However, we cannot commit to providing this information because this would make it difficult for us to be sure that we could find researchers willing to let us use their studies as base studies. Moreover, we have explicitly excluded moderator analyses based on the base study characteristics from our confirmatory plan, and we expect very little meaningful variance among the base studies on the characteristics that are most likely to influence participants' responses (as all studies will be administered online or on a computer, and will be minimal risk). Of course, we will do our best to make all of the

information we have and that all of the information that base study authors are willing to share easily accessible to readers.

[R5.21] The target sample size is very specific and it's unclear whether they will aim to get this exact number of participants or aim to get a larger number knowing that a reasonably high proportion of their data will be removed subject to their exclusion criteria. We would like clarity on the authors' approach to this stopping rule.

We have updated the description of our stopping rule and whether the target sample size is before or after exclusions, which we hope clarifies this point.

[R5.22] It appears that the authors haven't provided access to the final materials for the Registered Report study, only for the pilot studies. The authors should add these

Could the authors clarify what the attention/comprehension check is in this instance?

We have uploaded full materials (see comments to editor in [E.1]) which should clarify this. We apologize for not doing this before the first submission.

[R5.23] We don't think that getting participants' views constitutes checks and balances unless it comes with some kind of regulatory mechanism, the authors should consider reframing this. (lines 47-52)

We agree with the reviewers and have made changes to this section in the introduction which we hope clarifies this point.

[R5.24] Page 22, line 2 – does this mean variance between respondents of different base studies?

We will explore what proportion of variance can be attributed to the base study (between studies) vs. the participant (within study), but given that we will most likely end up with a small sample of studies, this is an exploratory analysis that should be interpreted with caution. We now explain this more clearly in the manuscript.

[R5.25] On sample selection criteria please clarify “and be able to answer our meta-study survey” and describe what a HIT rate is

We have removed this phrasing so it is less confusing.

[R5.26] Have the authors considered capturing other demographics variables previously captured by prior literature? (e.g. political leaning ala Pickett & Roche)

We thank the reviewers for this suggestion but have decided not to include more demographic variables in the study to avoid making it even longer.

Appendix D

21st May 2020

Editor, *Royal Society Open Science*

Dear Editor,

Enclosed is the second revision of our Stage I registered report manuscript entitled “What Do Participants Think of Our Research Practices? An Examination of Behavioral Psychology Participants’ Preferences.” We have now revised the paper to address the reviewers’ remaining concerns. See below for a point-by-point response to the points raised by you and the reviewers.

In summary, we have (1) added the suggested references to our manuscript’s introduction, and (2) compiled a list of potentially unstable references and added Web Archive links for each of them. This document is available in our OSF page (<https://osf.io/26ay8/>) and we have added a footnote pointing readers to it to the manuscript. We also appreciate Reviewer 2’s suggestion regarding the attention check and will incorporate it at Stage 2, as suggested. We appreciate the reviewers’ feedback and hope that the current version is acceptable.

Best Regards,

Julia G. Bottesini, Mijke Rhemtulla, & Simine Vazire

Point-by-point reply to editor and reviewers

Associate Editor Comments to Author (Professor Chris Chambers):

Associate Editor: 1

Comments to the Author:

All five reviewers have now assessed the revised the manuscript and are broadly satisfied. A few minor issues remain to be addressed concerning the attention check (Review 2) and referencing (Review 5). Concerning the point about referencing of websites, the authors note in their previous response that separately archiving the content (e.g. through screenshots) could conflict with copyright law. I wonder if a compromise solution could be to register each of the websites with <http://web.archive.org/> and then link to these URLs in the references?

We have now compiled a list of potentially unstable references and created Web Archive links for each of them. This document is available in our OSF page (<https://osf.io/26ay8/>) and we have added a footnote pointing readers to it to the manuscript. We thought about editing the citation links but decided against it to comply with APA guidelines that require us to indicate the webpage and date where we accessed the information.

Although entries in <http://web.archive.org/> are not necessarily permanent, and can be removed on request from site owners, they are considerably more stable than the regular internet. I will leave this for the authors to consider. Provided the authors are able to respond thoroughly to these remaining points in a minor revision, IPA should be forthcoming without requiring further in-depth Stage 1 review.

Reviewer comments to Author:

Reviewer: 1

Comments to the Author(s)

I'd like to thank the authors for their careful attention to the reviews and think that the information from these studies will be very informative.

We thank the reviewers for this feedback.

Reviewer: 2

Comments to the Author(s)

The scientific validity of the research question(s):

Consistent with the first review, the research questions are scientifically valid.

The logic, rationale, and plausibility of the proposed hypotheses:

The revised introduction provides a very clear rationale for the study. The authors have addressed previous comments and concepts that are important for understanding the rationale and research questions are now defined early on (e.g. QRPs).

The soundness and feasibility of the methodology and analysis pipeline (including statistical power analysis where applicable):

Previous comments on statistical analysis have been addressed. The soundness and feasibility of the methodology have further been established through the pilot studies.

Whether the clarity and degree of methodological detail would be sufficient to replicate exactly the proposed experimental procedures and analysis pipeline:

The authors have adopted a high standard of transparency and methods reproducibility throughout the manuscript. Study materials are also available on OSF (full survey).

Whether the authors provide a sufficiently clear and detailed description of the methods to prevent undisclosed flexibility in the experimental procedures or analysis pipeline:

The authors have provided enough details regarding the methods and analyses (including R code) to prevent undisclosed flexibility at Stage 2.

Whether the authors have considered sufficient outcome-neutral conditions (e.g. positive controls) for ensuring that the results obtained are able to test the stated hypotheses:

The authors have included a proposed robustness check (several Qs have two versions) and an attention check. The only minor comment/suggestion I have regarding the attention check is to make it easier for the reader to figure out what that involves, without having to go through the additional materials on OSF. This can easily be done (possibly at Stage 2) by simply adding a few words [Page 19, Line 52]: "participants have to report what they had for dinner.."; or verbatim "(i.e., What did you have for dinner last night?)".

I have no further comments and I look forward to reading the Stage 2 manuscript!

Loukia Tzavella

We thank the reviewers for this feedback. We plan to incorporate the Reviewer's suggestion at Stage 2, as suggested by the reviewer.

Reviewer: 3

Comments to the Author(s)

The authors have addressed all my comments.

Reviewer: 4

Comments to the Author(s)

This is round 2 of reviews, following revisions and resubmission, of a Stage 1 registered report (RR). I am happy with the revisions made in the paper specifically addressing my previous concerns. I'm also particularly happy about the decision to remove hypothesis testing and report the results descriptively, with the inclusion of confidence bounds. I think the study is ready to go and I'm excited about the results (I think they'll be very interesting and informative for the scientific community). I have no further suggestions and look forward to the Stage 2 review.

My explicit comments on each of the journal review criteria:

The scientific validity of the research question(s)

I believe the research questions are interesting and important for the scientific community and have scientific validity.

The logic, rationale, and plausibility of the proposed hypotheses

There are no hypotheses, though the logic and rationale for the importance of the research questions and the methods are sound.

The soundness and feasibility of the methodology and analysis pipeline (including statistical power analysis where applicable)

The methods and analysis pipeline are sound for addressing the research questions.

Whether the clarity and degree of methodological detail would be sufficient to replicate exactly the proposed experimental procedures and analysis pipeline

Together with the supplement there is enough clarity and methodological detail for an exact replication.

Whether the authors provide a sufficiently clear and detailed description of the methods to prevent undisclosed flexibility in the experimental procedures or analysis pipeline

I see little to no room for undisclosed flexibility.

Whether the authors have considered sufficient outcome-neutral conditions (e.g. positive controls) for ensuring that the results obtained are able to test the stated hypotheses

The authors have taken sufficient steps to obtain high quality data.

We thank the reviewers for this feedback.

Reviewer: 5

Comments to the Author(s)

The authors have done a great job of addressing the reviewer comments. The article flows well now and is more readable for an interdisciplinary audience.

We apologize for not including the references we had asked the authors to include for questionable research practices. We recommend that the authors include reference to:

Agnoli F, Wicherts JM, Veldkamp CLS, Albiero P, Cubelli R. Questionable research practices among Italian research psychologists. PLoS One. 2017;12: 1–17.

doi:10.1371/journal.pone.0172792

Makel MC, Hodges J, Cook BG, Plucker JA. Questionable and open research practices in education research. EdarXiv. 2019; doi:<https://doi.org/10.35542/osf.io/f7srb>

Fox NW, Honeycutt N, Jussim L. How Many Psychologists Use Questionable Research Practices ? Estimating the Population Size of Current QRP Users. 2018; 1–10.

We have now added the suggested references to our manuscript's introduction.

In our previous review of this article we noted that referring to websites is potentially problematic but the authors rebutted this point. We still feel strongly that referencing websites that do not have DOIs is not good practice. We are particularly concerned about the references to company website pages (e.g. FAQ, or 'about' page) which are regularly updated as policies change - If someone looked at these references in the future they might find completely contradictory information and they would have no idea what the sites said when you referenced them. We would prefer it if the authors either a) found reports or articles with DOIs that speak to these issues or b) did not provide references for these statements at all (this is less important for the Bastian blog which may not be available in the future but is unlikely to change in message).

We have now addressed this point. See comments to the editor.

Overall this is a really well written and interesting article and we cant wait to see the results.

Review by Hannah Fraser and Daniel Hamilton

Appendix E

27th December 2021

Editor, *Royal Society Open Science*

Dear Editor,

Enclosed is an invited Stage 2 registered report manuscript to be considered for publication in *RSOS* (the Stage 1 registered report received an In Principle Acceptance). The manuscript is entitled *What Do Participants Think of Our Research Practices? An Examination of Behavioral Psychology Participants' Preferences*.

The archived study data, digital materials, and analysis code can be found at <https://osf.io/bgpvc/>, and this link is also available on page 46 of the Stage 2 manuscript. The approved Stage 1 manuscript can be found at <https://osf.io/re5uf/> and the corresponding time-stamped registration can be found at <https://osf.io/8anxu>. Both links are available on page 46 of the Stage 2 manuscript as well. We confirm that no data for this preregistered study (other than pilot data included at Stage 1) was collected prior to the date of IPA.

The authors confirm that the completed study has been executed and analysed in the manner originally approved. We have not changed any of the text in the Introduction or Method, except we changed “selective reporting of studies” to “selective reporting of studies (filed drawing)” in Table 1, because we used the shorter label (“filed drawing”) in subsequent figures and tables. We included results for all preregistered analyses, and clearly labeled any exploratory analyses as such. Slight deviations from the preregistered recruiting strategies and other unforeseen circumstances are clearly stated in the manuscript.

The authors have no financial conflicts of interest to declare. All authors approved the manuscript and this submission.

Best Regards,

Julia Bottesini, M.A., Mijke Rhemtulla, Ph.D., & Simine Vazire, Ph.D.

Appendix F**ROYAL SOCIETY
OPEN SCIENCE****What Do Participants Think of Our Research Practices? An
Examination of Behavioral Psychology Participants'
Preferences**

Journal:	Royal Society Open Science
Manuscript ID	RSOS-200048.R3
Article Type:	Registered Report - Stage 2
Date Submitted by the Author:	27-Dec-2021
Complete List of Authors:	Bottesini, Julia; University of California Davis, Psychology Rhemtulla, Mijke; University of California Davis, Psychology Vazire, Simine; The University of Melbourne Melbourne School of Psychological Sciences, Psychology; University of California Davis, Psychology
Subject:	psychology < BIOLOGY
Keywords:	Research practices, Open Science, Scientific integrity, Informed consent
Subject Category:	Psychology and cognitive neuroscience

Author-supplied statements

Relevant information will appear here if provided.

Ethics

Does your article include research that required ethical approval or permits?:

Yes

Statement (if applicable):

Permission to perform this study was granted by the University of California Institutional Review Board (IRB), IRB IDs 1423371-2, 1787646-1, and 1744965-1. Permission to perform this study (and accompanying base studies) at other universities was granted by the Sacramento State Institutional Review Board (IRB), IRB ID Cayuse-20-21-240; the Princeton University Institutional Review Board (IRB), IRB ID 13508-04; and the University of Pennsylvania Institutional Review Board (IRB), IRB IDs 844186 and 844870.

Data

It is a condition of publication that data, code and materials supporting your paper are made publicly available. Does your paper present new data?:

Yes

Statement (if applicable):

All data for the pilots is available at the OSF page for this project (<https://osf.io/bgpyc/>). Data for the main study can be found at <https://osf.io/zr29g/>.

Conflict of interest

I/We declare we have no competing interests

Statement (if applicable):

CUST_STATE_CONFLICT :No data available.

27th December 2021

Editor, *Royal Society Open Science*

Dear Editor,

Enclosed is an invited Stage 2 registered report manuscript to be considered
for publication in *RSOS* (the Stage 1 registered report received an In Principle
Acceptance). The manuscript is entitled *What Do Participants Think of Our*
*Research Practices? An Examination of Behavioral Psychology Participants'*
*Preferences*.

The archived study data, digital materials, and analysis code can be found at
<https://osf.io/bgpvc/>, and this link is also available on page 46 of the Stage 2
manuscript. The approved Stage 1 manuscript can be found at
<https://osf.io/re5uf/> and the corresponding time-stamped registration can be
found at <https://osf.io/8anxu>. Both links are available on page 46 of the Stage 2
manuscript as well. We confirm that no data for this preregistered study (other
than pilot data included at Stage 1) was collected prior to the date of IPA.

The authors confirm that the completed study has been executed and analysed
in the manner originally approved. We have not changed any of the text in the
Introduction or Method, except we changed “selective reporting of studies” to
“selective reporting of studies (filedrawing)” in Table 1, because we used the
shorter label (“filedrawing”) in subsequent figures and tables. We included
results for all preregistered analyses, and clearly labeled any exploratory
analyses as such. Slight deviations from the preregistered recruiting strategies
and other unforeseen circumstances are clearly stated in the manuscript.

The authors have no financial conflicts of interest to declare. All authors
approved the manuscript and this submission.

Best Regards,

Julia Bottesini, M.A., Mijke Rhemtulla, Ph.D., & Simine Vazire, Ph.D.

What Do Participants Think of Our Research Practices? An Examination of Behavioral Psychology Participants' Preferences

Julia Bottesini¹, Mijke Rhemtulla¹, & Simine Vazire^{1,2}

¹University of California—Davis; ²University of Melbourne

Abstract

What research practices should be considered acceptable? Historically, scientists have set the standards for what constitutes acceptable research practices. However, there is value in considering non-scientists' perspectives, including research participants'. 1,873 participants from MTurk and university subject pools were surveyed after their participation in one of eight minimal-risk studies. We asked participants how they would feel if common research practices were applied to their data: *p*-hacking/cherry-picking results, selective reporting of studies, Hypothesizing After Results are Known (HARKing), committing fraud, conducting direct replications, sharing data, sharing methods, and open access publishing. An overwhelming majority of psychology research participants think questionable research practices (e.g., *p*-hacking, HARKing) are unacceptable (68.3--81.3%), and were supportive of practices to increase transparency and replicability (71.4--80.1%). A surprising number of participants expressed positive or neutral views toward scientific fraud, raising concerns about the quality of our data. We grapple with this concern and interpret our results in light of the limitations of our study. Despite ambiguity in our results, we argue that there is evidence (from our study and others') that researchers may be violating participants' expectations and should be transparent with participants about how their data will be used.

Keywords: Research practices; Open Science; Scientific integrity; Informed consent

Background

What research practices should be considered acceptable, and who gets to decide? Historically, scientists – and as a group, scientific organizations – have set the standards and have been the main drivers of change in what constitutes acceptable research practices. Perhaps this is warranted. Who better to set the standards than those who know research practices best? It seems reasonable that decisions regarding those practices should be entrusted to scientists themselves. However, there may be value in considering non-scientists' perspectives and preferences, including research participants'.

The replicability crisis in psychology has demonstrated that scientists are not always good at regulating their own practices. For example, a surprisingly high proportion of researchers admit to engaging in questionable research practices, or QRPs (as described in John et al., 2012; see also Agnoli et al., 2017; Fox et al., 2018; Makel et al., 2019). 
[revised manuscript text omitted]

teams will collect data. To find these researchers, we will use the Study Swap website
(osf.io/meetings/studyswap/), social media, and personal contacts.

We will decide which studies to include in our sample on a case-by-case basis. The base studies
must meet the following criteria: (1) a minimal-risk study where all the data are collected in a
single session, either online or on a local computer, (2) study is run in English, and, if it uses an
undergraduate subject pool, it is run at a college or university where English is the primary
language of instruction, (3) participants are recruited from either college/university subject pools
or online platforms and meet our inclusion criteria (see below), (4) feasibility constraints – e.g.,
whether we have the resources to run participants on our end, whether the IRB approval is easy to
obtain, etc.; (5) progress of our sample size goals – e.g., if we have met our goal for student or
online participants, we will stop collecting data from that population; (6) time constraints – we
will be able to complete data collection for the study within the time frame allotted for the
project; (7) sample size – the study will provide a minimum of 50 participants; and (8) base study
materials will be made publicly available.

Sample Selection Criteria

Participants must speak English and be at least 18 years old. They also must qualify for and complete the base study that precedes ours, so our study inclusion criteria will include the inclusion criteria used by each base study to which we will append our meta-study. For example, if one of the studies selects only first-generation college students, or only women, this will also be a criterion to participate in our meta-study for that subsample.

We have funds to collect data on MTurk and resources to collect data from the UC Davis undergraduate subject pool, so data collection conducted by us will come from one of these two populations. For MTurk samples recruited by us, participants need to meet the following criteria: (1) be located in the United States; (2) have a Human Intelligence Task (HIT) approval rate of 90% or higher⁴; and (3) have at least 10 HITs approved. MTurk samples recruited by partner researchers running base studies will follow that team's criteria.

It is also possible that some data will be collected by the base study researchers, and these data could be collected from other colleges' or universities' subject pools, or online platforms other than MTurk, like Prolific (<https://prolific.co/>). In these cases, the selection criteria for participants (beyond the requirement that participants speak English and be at least 18 years old) will be decided by the base study researchers.

Meta-study

⁴ A HIT, or "human intelligence task", is a task available for Amazon Mechanical Turk workers. A workers' HIT approval rate is the proportion of tasks that have been approved by the requester. The authors consider 90% to be a reasonable cutoff to ensure high quality data.

The meta-study will ask participants to consider an anonymized version of the data they have just provided in the base study, and imagine a series of hypothetical situations in which researchers use different research practices on their data. Specifically, we will ask them their opinions on the eight practices shown in Table 1. We have honed the wording of these questions using the data and feedback from our pilot studies, which we describe in detail in the supplementary materials. The full text for the questions in Table 1 can be found at <https://osf.io/p8n9w/>.

Table 1. Description of Survey Questions.

Question Number	Question Topic	Number of Versions
p -hacking or cherry-picking results	2 versions
selective reporting of studies (filedrewing)	2 versions
HARKing	1 version
fraud	1 version
direct replication	1 version
open methods	2 versions
open access publication	1 version
data sharing	2 versions

Note. Each participant will see only one version of each question. See materials for a full description of the question wording, versions, and response options.

Our goal is to ask the questions in a way that is not leading. When we could not find a way to do this while still providing a clear description of the practice (i.e., for Questions 1, 2, and 8 – see table 1 for a list of which questions correspond to which research practice), we wrote two different versions of the question reflecting the tradeoff between providing a fuller but potentially leading description of the practice, and providing a vaguer but less valenced description of the practice. For Question 6, we also created two versions: the “positive” version of the question, asking participants how they would feel if the researchers shared enough details about their materials and procedures for others to conduct a replication study, or the “negative” version,

which asks participants how they would feel if researchers did not share enough details. If the
answers differ by version, which the pilot studies suggest might happen, we will have estimates of
the distribution of responses to these practices for two different, but hopefully reasonable, ways
to ask the same question. In other words, these two versions provide a kind of robustness check
across variations that we hope capture similar phenomena.

For questions 1, 2, 3, and 4, the research practices will be described in simple terms, and
participants will be asked to rate each practice on a 5-point scale with anchors at -2 (“definitely
not acceptable”) through 0 (“Indifferent”) to +2 (“definitely acceptable”).

Question 5 asks participants their opinion about whether researchers should attempt to replicate
a finding before publishing it or simply move on to a new project. With this question, we are
hoping to make the tradeoffs involved in conducting a direct replication (vs. not conducting one)
clear, without leading participants towards one answer or the other. Participants will answer on a
5-point scale with anchors being “strongly prefer that the researchers move on to their next
project”, “slightly prefer that the researchers move on to their next project”, “indifferent”, “slightly
prefer that the researchers replicate the study”, and “strongly prefer that the researchers
replicate the study”.

Questions 6, 7, and 8 ask participants to consider situations where researchers can choose to use
open science practices. For Question 6 (“open methods”), which is about whether researchers
should share their materials and procedures, participants are asked to rate this practice on a 5-
point scale with anchors at -2 (“feel strongly that the researchers should **not** do this”) through 0
(“indifferent”) to +2 (“feel strongly that the researchers **should** do this”). There are two versions of
this question. The positive version describes researchers providing all necessary information for
replication, while the negative version (reverse scored) describes not providing enough
information.

For Question 7, participants are asked whether they have a preference for where the article
reporting the results of the base study should be published: an open access journal vs. a pay-
walled journal. Participants answer on a 5-point scale with anchors being “strongly prefer that it
cost about \$30 to read the article”, “slightly prefer that it cost about \$30 to read the article”,
“indifferent”, “slightly prefer that the article be free to read”, and “strongly prefer that the article
be free to read.” The value of \$30 dollars is typical of several top journals in Psychology. Based on
feedback on our pilot materials, we also added a clarification statement so respondents
understand that the value paid for the article does not go to the authors of the article – a
reasonable but false assumption – but to the publisher.

For Question 8, we will ask two versions of the question: one that explicitly states reasons why a
researcher may or may not want to share their data (“reasons provided”), and one that does not
(“neutral”). The reasons-provided version spells out the main reasons for and against data sharing.
We developed this list of reasons by consulting published work on researchers’ stated reasons for
sharing or not sharing data (Washburn et al., 2018). The neutral version of this question asks
participants to consider potential reasons before answering, and makes it clear that valid reasons
exist both for and against data sharing. Participants will answer on a 5-point scale with anchors at
32 -2 (“feel strongly that the researchers should **not** do this”) through 0 (“indifferent”) to +2 (“feel
strongly that the researchers **should** do this”).

For all of the questions, we will use a 5-point response scale. This has been changed from a 7-point
scale in Pilots B and C, as we believe this better reflects the granularity of judgment that is
reasonable to expect from research participants. Moreover having fewer response options will
give us more precision when estimating the proportion of people who choose each response
option. We have also changed the order and anchors for some of the questions. For questions

where it makes sense to have a negative and positive end of the scale (e.g., “researchers should **not**
do this” vs. “researchers **should** do this”) we kept the numbering (-2 to +2) with anchors at the
ends and midpoint. However, some of the questions represent trade-offs (e.g., whether to publish
open access vs. behind a paywall) which have no clear “positive” or “negative” end. Therefore, we
labeled all 5 points for Questions 5 (direct replication) and 7 (open access publishing) with words
rather than numbers, to avoid inadvertently conveying that one end of the scale is more desirable
than the other (e.g., higher numbers, or positive numbers).

For questions which have two versions, participants will be randomly assigned to answer one or
the other. Random assignment will be independent between questions; e.g., a participant who was
assigned to the neutral version of the data sharing question could be assigned to either the neutral
or the reasons-provided version of the selective reporting question. Furthermore, the order of the
eight questions will be randomized.

Finally, we will ask additional questions for potential exploratory analyses. First, we will ask about
demographics, including gender, race and ethnicity, year of birth, education, proximity to science,
and the number of psychology studies the participant participated in during the previous two
35 weeks. We will also measure trust in psychological science with three statements (“Findings from
36 psychology research are trustworthy,” “I have very little confidence in research findings from
37 psychology” (reverse-scored), and “I trust psychology researchers to do good science”) which
participants are asked to rate on a 7-point scale from (1) “strongly disagree” to (7) “strongly agree.”
We will also ask participants “Have you heard of the replication crisis in psychology?” (Yes/No). If
participants answer yes, we will ask them “Please describe what you have heard about the
replication crisis:” and provide an open-ended text box for their response. Although we do not
have planned analyses that use these additional questions, they will be collected to allow for
exploratory analyses both by the authors and others who wish to reuse the data.

Data Exclusion Criteria

The survey will include one open ended attention/comprehension check. These answers will be coded by an independent coder, who will be blind to how they relate to the rest of the data, as “appropriate,” “inappropriate,” and “unclear.” Only “inappropriate” answers will be excluded.

Results

Sample

The data were collected between January and October of 2021, yielding a total of 1,990 observations before exclusions from 8 different base studies. After performing the preregistered exclusions, we obtained a final sample of 1,873 participants – 40% from participants in Amazon Mechanical Turk studies (5 studies) and 60% from university subject pool study participants (3 studies across 4 subject pools) – with the breakdown described in Table 2.

57.9 % of participants described themselves as female, 40.4% described themselves as male, 0.8% self-identified as non-binary or a third gender, 0.3% preferred to self-describe, and 0.5% preferred not to report their gender. The median year of birth for participants was 1999 (IQR = 14; range = 1946-2003). Participants could select multiple race and ethnicity categories; 51.8% identified as white, 29.2% as Asian, 13.5% as Hispanic or Latino or Chicano or Puerto Rican, 8.0% as Black or African American, 1.6% Middle Eastern or North African, 0.8% American Indian or Alaska Native, 0.7% Native Hawaiian or Pacific Islander, and 1.4% said they had another identity; 7.5% of participants declined to self-identify on race and ethnicity.

Table 2. Sample size, population, and short description of each base study.

Base Study	Sample size			Population	Study description
	Before exclusions	After preregistered exclusions only	After non-preregistered (strict) exclusions		
BS01	500	499	437	Sacramento State University and University of California, Davis Subject Pools	A study about individuals' reactions to marginalized individuals in positions of power
BS02	390	389	363	University of Pennsylvania and University of California, Davis Subject Pools	A study exploring the reasons that people overassess experts' abilities.
BS03	237	237	227	Princeton University Subject Pool	A study about friendship formation and related attitudes
BS04	162	145	93	Amazon Mechanical Turk Workers	A study about interviews in false confessions documentaries and how they influence laypeople's perceptions of a confession
BS05	252	201	115	Amazon Mechanical Turk Workers	A study about perspective taking of climate refugees
BS06	106	100	86	Amazon Mechanical Turk Workers	A study about the relationships between Dark Personality, Self-Control and Aggression.
BS07	130	123	99	Amazon Mechanical Turk Workers	A study testing interindividual variability in free- and cued-recall memory performance
BS08	213	179	117	Amazon Mechanical Turk Workers	A study about people's perceptions of the most moral, least moral, and morally average people they personally know.

**Deviations from Planned Recruiting Strategy.** In the base-study recruiting phase, we
communicated with several potential base studies, and received responses from many researchers
willing to collaborate. Those not mentioned here did not meet our inclusion criteria, or the
collaborating researcher later decided against following through for a variety of reasons, and we
never reached the data collection stage with these studies. The one exception to this was a study
for which we did not have enough information to realize it did not meet our inclusion criteria until
after data collection was completed, so although we do have the participants' data for this other
study, we are not including it or its data here. One other slight deviation from our recruiting plan
was that BS03 did not have an initial agreed-upon sample size, but an end date (October 31st)
when the collaborating researchers had preregistered to check whether they had enough data to
perform their analyses; this served as the stopping rule for our part of the study. Finally, some
MTurk studies ended up with a few more observations than we aimed to collect, and BS02 had 10
fewer observations than agreed due to reaching the end of their semester.

*Analyses*

Our primary analyses examine the distribution of participants' responses, which we examined
using descriptive statistics presented in Tables 3 and 4. We also present the corresponding
visualizations of the distributions of responses in Figures 1 and 2. We first report the descriptives
using the preregistered exclusion criteria, starting with results for the combined samples (Table 4
and Figure 1), then results for each question version separately (for questions that had more than
one version; Table 5 and Figure 2). We also report the exploratory analyses we outlined in the
stage 1 manuscript. Then, because we found certain patterns of results suspicious, we deviate
from the preregistered analysis plan and repeat some of the analyses with stricter exclusion

[revised manuscript text omitted]
/ filedrawing		Neutral (%) (Mdn = -1, IQR = 2) n = 950	Motive (%) (Mdn = -2, IQR = 2) n = 923
	Not acceptable	66.8 [63.9, 69.9]	71.6 [68.8, 74.6]
	Indifferent	9.37 [6.42, 12.5]	6.18 [3.36, 9.18]
	Acceptable	23.8 [20.8, 26.9]	22.2 [19.4, 25.2]
Question 6: open methods		Rs use open methods (%) (Mdn = 2, IQR = 1) n = 907	Rs use “closed” methods (%) (Mdn = 1, IQR = 2) 
[revised manuscript text omitted]

**Author Contributions**

JB and SV developed the study idea, design, and materials. JB and MR developed and
wrote code for the planned analyses. JB ran pilot study B and coordinated with colleagues
who ran pilot studies A and C. JB performed all pilot data analyses. JB did most of the data
collection and coordinated with colleagues who did the rest of the data collection at their
institutions. JB performed all stage 2 data analyses. JB drafted most of the first draft
manuscript, SV drafted some parts. JB and SV made extensive revisions to the
manuscript. All authors made minor edits and approved the final version.

**CRedit taxonomy:**

45 J.B.: Conceptualization, Data curation, Formal analysis, Investigation, Methodology,
Project administration, Resources, Software, Visualization, Writing - original draft, and
Writing - review & editing.

52 M.R.: Methodology, Supervision, and Writing - review & editing.

S.V.: Conceptualization, Investigation, Methodology, Supervision, Writing - original draft,
and Writing - review & editing.

Competing Interests

We have no competing interests.

Funding

Funding for this study is provided by university research funds to Simine Vazire and Mijke Rhemtulla.

Acknowledgments

We thank Hale Forster, Oliver Clark, Jessie Sun, Gerit Pfuhl, Eric Y. Mah, D. Stephen Lindsay, Yeji Park, Kate M. Turetsky, Kevin Reinert, Samuel H. Borislow, Jasmin Fernandez Castillo, Greg M. Kim-Ju, Jeremy R. Becker, Kate Hussey, and Fabiana Alceste for agreeing to provide us with base studies. We also thank Hale Forster and Oliver Clark for running data collection for Pilots A and C; Jessie Sun, Yeji Park, Gerit Pfuhl, Jasmin Fernandez Castillo, Samuel H. Borislow, and Jack Friedrich for running data collection for parts of the main study; and Beth Clarke for comments on the manuscript.

References

Agnoli, F., Wicherts, J. M., Veldkamp, C. L., Albiero, P., & Cubelli, R. (2017). Questionable research
practices among Italian research psychologists. *PloS one*, 12(3).
- AllTrials. (n.d.). About AllTrials. Retrieved June 13, 2019, from AllTrials website:
<https://www.alltrials.net/find-out-more/all-trials/>
- Bastian, H. (2017, August 29). Bias in Open Science Advocacy: The Case of Article Badges for Data
Sharing. Retrieved November 24, 2019, from Absolutely Maybe website:
[https://blogs.plos.org/absolutely-maybe/2017/08/29/bias-in-open-science-advocacy-the-
case-of-article-badges-for-data-sharing/](https://blogs.plos.org/absolutely-maybe/2017/08/29/bias-in-open-science-advocacy-the-
19 case-of-article-badges-for-data-sharing/)
- Cummings, J. A., Zagrodny, J. M., & Day, T. E. (2015). Impact of Open Data Policies on Consent to
Participate in Human Subjects Research: Discrepancies between Participant Action and
Reported Concerns. *PLoS ONE*, 10(5). <https://doi.org/10.1371/journal.pone.0125208>
- Fox, N. W., Honeycutt, N., & Jussim, L. (2018, August 14). How Many Psychologists Use
Questionable Research Practices? Estimating the Population Size of Current QRP Users.
<https://doi.org/10.31234/osf.io/3v7hx>
- Frequently Asked Questions about the NIH Public Access Policy | publicaccess.nih.gov. (n.d.).
Retrieved June 13, 2019, from <https://publicaccess.nih.gov/faq.htm#753>
- John, L. K., Loewenstein, G., & Prelec, D. (2012). Measuring the Prevalence of Questionable
Research Practices With Incentives for Truth Telling. *Psychological Science*, 23(5), 524–532.
<https://doi.org/10.1177/0956797611430953>
- Kidwell, M. C., Lazarević, L. B., Baranski, E., Hardwicke, T. E., Piechowski, S., Falkenberg, L.-S., ...
Nosek, B. A. (2016). Badges to Acknowledge Open Practices: A Simple, Low-Cost, Effective
Method for Increasing Transparency. *PLOS Biology*, 14(5), e1002456.
<https://doi.org/10.1371/journal.pbio.1002456>
- MacInnis, B., Krosnick, J. A., Ho, A. S., & Cho, M. J. (2018). The accuracy of measurements

with probability and nonprobability survey samples: Replication and extension. *Public*
*Opinion Quarterly*, 82(4), 707-744. <https://doi.org/10.1093/poq/nfy038>

Makel, M. C., Hodges, J., Cook, B. G., & Plucker, J. (2019, October 31). Questionable and Open
Research Practices in Education Research. <https://doi.org/10.35542/osf.io/f7srb>

McSweeney, B., Allegretti, J. R., Fischer, M., Monaghan, T., Mullish, B. H., Petrof, E. O., ... Kao, D. H.
(n.d.). Potential Motivators and Deterrents for Stool Donors: A Multicenter Study.

Retrieved February 14, 2019, from

<https://ep70.eventpilot.us/web/page.php?page=IntHtml&project=DDW18&id=2907807>

Mello, M. M., Lieou, V., & Goodman, S. N. (2018). Clinical Trial Participants' Views of the Risks and
Benefits of Data Sharing. *New England Journal of Medicine*, 378(23), 2202-2211.

<https://doi.org/10.1056/NEJMsa1713258>

Motyl, M., Demos, A. P., Carsel, T. S., Hanson, B. E., Melton, Z. J., Mueller, A. B., ... Skitka, L. J.

(2017). The state of social and personality science: Rotten to the core, not so bad, getting
better, or getting worse? *Journal of Personality and Social Psychology*, 113(1), 34-58.

<https://doi.org/10.1037/pspa0000084>

Open Science Badges. (n.d.). Retrieved June 6, 2019, from [https://cos.io/our-services/open-](https://cos.io/our-services/open-science-badges/)
[science-badges/](https://cos.io/our-services/open-science-badges/)

Patient Groups, Industry Seek Changes to Rare Disease Drug Guidance. (n.d.). Retrieved June 13,
2019, from [https://www.raps.org/news-and-articles/news-articles/2019/4/patient-](https://www.raps.org/news-and-articles/news-articles/2019/4/patient-groups-industry-seek-changes-to-rare-dise)
[groups-industry-seek-changes-to-rare-dise](https://www.raps.org/news-and-articles/news-articles/2019/4/patient-groups-industry-seek-changes-to-rare-dise)

Pickett, J. T., & Roche, S. P. (2018). Questionable, Objectionable or Criminal? Public Opinion on
Data Fraud and Selective Reporting in Science. *Science and Engineering Ethics*, 24(1), 151-
171. <https://doi.org/10.1007/s11948-017-9886-2>

Protection of Human Subjects. , Pub. L. No. 45, § 46, C.F.R. (2009).

Rowhani-Farid, A., Allen, M., & Barnett, A. G. (2017). What incentives increase data sharing in
health and medical research? A systematic review. *Research Integrity and Peer Review*, 2(1),

4. <https://doi.org/10.1186/s41073-017-0028-9>

Sanderson, S. C., Linderman, M. D., Suckiel, S. A., Diaz, G. A., Zinberg, R. E., Ferryman, K., ... Schadt,
E. E. (2016). Motivations, concerns and preferences of personal genome sequencing
research participants: Baseline findings from the HealthSeq project. *European Journal of*
*Human Genetics*, 24(1), 14–20. <https://doi.org/10.1038/ejhg.2015.118>

Simmons, J. P., Nelson, L. D., & Simonsohn, U. (2011). False-Positive Psychology: Undisclosed
Flexibility in Data Collection and Analysis Allows Presenting Anything as Significant.
*Psychological Science*, 22(11), 1359–1366. <https://doi.org/10.1177/0956797611417632>

Smaldino, P. E., & McElreath, R. (2016). The natural selection of bad science. *Royal Society Open*
*Science*, 3(9), 160384. <https://doi.org/10.1098/rsos.160384>

Trinidad, S. B., Fullerton, S. M., Ludman, E. J., Jarvik, G. P., Larson, E. B., & Burke, W. (2011).
Research Practice and Participant Preferences: The Growing Gulf. *Science*, 331(6015),
287–288. <https://doi.org/10.1126/science.1199000>

TurkPrime. (n.d.). After the Bot Scare: Understanding What's Been Happening with Data
Collection on MTurk and How to Stop it. Retrieved May 12, 2019, from
[https://blog.turkprime.com/after-the-bot-scare-understanding-whats-been-happening-](https://blog.turkprime.com/after-the-bot-scare-understanding-whats-been-happening-with-data-collection-on-mturk-and-how-to-stop-it)
[with-data-collection-on-mturk-and-how-to-stop-it](https://blog.turkprime.com/after-the-bot-scare-understanding-whats-been-happening-with-data-collection-on-mturk-and-how-to-stop-it)

Washburn, A. N., Hanson, B. E., Motyl, M., Skitka, L. J., Yantis, C., Wong, K. M., ... Carsel, T. S. (2018).
Why Do Some Psychology Researchers Resist Adopting Proposed Reforms to Research
Practices? A Description of Researchers' Rationales. *Advances in Methods and Practices in*
*Psychological Science*, 1(2), 166–173. <https://doi.org/10.1177/2515245918757427>

Figure 1. Distribution of participants' answers for each question. For the top four panels, negative numbers indicate that participants found the practice unacceptable while positive numbers indicate they found the practice acceptable. For the bottom four panels, higher numbers indicate more support for the practice. N = 1,873. See also Table 4.

888x1110mm (72 x 72 DPI)

Figure 2. Distribution of participants' answers for each question, by question version, for the four questions with two versions. For the top two panels, negative numbers indicate that participants found the practice unacceptable while positive numbers indicate they found the practice acceptable. For the bottom two panels, higher numbers indicate more support for the practice. For p-hacking and filedrawing, the neutral version described the behavior only (i.e., researchers "only reported some of the results" or "did not report all of the studies they ran"), while the motive version implied motivated reasons behind the selective reporting of results or studies (e.g., "only reported the results/studies that came out the way they predicted"). The positive and negative versions of the open methods question were phrased as "[researchers] provided [vs. did not provide] a lot of details about how they did the study. Therefore, other researchers could [vs. could not] easily conduct a replication...". The neutral data sharing question asked whether participants thought "researchers should share the dataset when they publish their results" while the reasons version asked the same but provided some reasons why researchers may or may not want to share their data (e.g., concerns about scooping or making it possible for others to verify their work). See Table 5 for more detailed results and sample sizes.

688x841mm (95 x 95 DPI)

Figure 3. Distribution of university subject pool and MTurk participants' responses for all 8 questions, after non-preregistered, strict exclusions. For the top four panels, negative numbers indicate that participants found the practice unacceptable while positive numbers indicate they found the practice acceptable. For the bottom four panels, higher numbers indicate more support for the practice. N = 1,537. See also Table 7.

888x1110mm (72 x 72 DPI)

Figure 4. Distribution of participants' answers for each question with non-preregistered, strict exclusions. For the top four panels, negative numbers indicate that participants found the practice unacceptable while positive numbers indicate they found the practice acceptable. For the bottom four panels, higher numbers indicate more support for the practice. See Table 9 for additional results and sample sizes.

888x1110mm (72 x 72 DPI)

Appendix G

2 February, 2022

Editor, *Royal Society Open Science*

Dear Editor,

Enclosed is a revision of our Stage 2 registered report manuscript entitled “What Do Participants Think of Our Research Practices? An Examination of Behavioral Psychology Participants’ Preferences.” We would like to thank you and the reviewers for your detailed feedback. We have made minor revisions to the paper in light of this feedback.

Broadly, our changes consist of:

- 1) Replacing Figure 4 (distribution of responses after strict/post-hoc exclusions) with a similar figure which shows those distributions overlaid on the original distributions, with preregistered exclusions. We believe this will allow the reader to more easily compare the two sets of results, an excellent point which was brought up by more than one reviewer.
- 2) Editing the abstract to include more detail (as much as we could within the word limit), as suggested by Reviewer 3.
- 3) Expanding the first few paragraphs of the Analyses section to more clearly warn the reader that non-preregistered analyses parallel to the main analyses are presented later in the manuscript with stricter, but post-hoc, exclusion criteria to address potential data quality issues. We hope this will clarify where the reader can go to find these potentially more informative results, if they so desire, while still keeping all exploratory analyses in a single, clearly labeled section of the manuscript.
- 4) Making miscellaneous small edits for clarity.

We believe this addresses most of the concerns brought up by the reviewers. We could not incorporate all of the reviewers’ excellent suggestions due to our desire to stick to the Stage 1 plan, and keep the focus of the paper on our central research questions and pre-registered plan. However, if you feel that there are changes we should make but did not make, we would be more than happy to take your advice. One suggestion we were not sure whether to take, for example, was the suggestion to change the future tense in the introduction and method sections to past tense. We were not sure whether it is better to stick to the language used in the Stage 1 manuscript or update the tense, and we would be happy to go with whatever you recommend.

Thank you again for your support during all of this process — this was a great experience!

Best Regards,

Julia G. Bottesini, Mijke Rhemtulla, & Simine Vazire